# iPSC-derived ITGA6-positive cells restore aqueous humor outflow in glaucoma eyes

Pengchao Feng [1,17], Chen Yu [2,17], Xiaoyan Zhang [1,17], Bin Xu [3,17], Shen Wu [4,5,6,17], Chen Xin [5], Sejiro Littleton [2,7], Jie Kang [3], Xiangji Wang [1], Shaoshuai Liang [8], Susu Chen [1], Wenyan Wang [1], Yanan Wang [1], Zhixin Yuan [9], Gaiping Xi [1], Xinhui Xing [1], Xinyu Ge [1], Zhishang Chang [10], Jinshan Tan [11], Jingxue Zhang [4,5,6], Bingqiang Zhang [8], Qilong Cao [9,12], Wenhua Xu [13] ✉, Markus H. Kuehn [14,15], Ningli Wang [4,5,6,16] ✉ & Wei Zhu [1] ✉

Decreased trabecular meshwork (TM) cellularity is a critical pathogenic cause of primary open-angle glaucoma, yet therapies to regenerate the decellularized TM are very limited. Induced pluripotent stem cell-derived TM-like cells (iPSC-TM) can efficiently restore aqueous humor outflow. Here, we conducted a multi-modal RNA sequencing analysis to characterize the molecular mechanisms underlying TM regeneration. Our clustering analysis identified a group of iPSC-derived alpha6 integrin-positive (iPSC-ITGA6+) cells with a distinct transcriptome that wasn't observed in primary TM (pTM) cells. These iPSC-ITGA6+ cells not only stimulate pTM proliferation but also facilitate the repopulation of the TM and Schlemm's canal in glaucoma, with a much higher efficiency than other iPSC-TM subtypes. Interaction with iPSC-ITGA6+ cells is characterized by the proliferation and rejuvenation of endogenous pTM cells, primarily through the transcription of long non-coding RNA nuclear paraspeckle assembly transcript1 and the abundance of paraspeckles within iPSC-ITGA6+ cells. Enhancing paraspeckle assembly by MEN β-associated RNA promotes the rejuvenation and proliferation of pTM, suggesting a novel and promising approach for TM regeneration.

Glaucoma is the leading cause of irreversible blindness worldwide[1]. Elevated intraocular pressure (IOP) due to the imbalance of aqueous humor (AH) production and outflow is a major risk factor for developing glaucoma[2]. Controlling IOP remains the mainstay of glaucoma therapy[3]. In most primary open-angle glaucoma (POAG) cases, AH production remains normal, while increased resistance to AH outflow is the pathological phenotype[4]. Notably, 80–90% of the AH outflow resistance is generated from the trabecular meshwork (TM) and Schlemm's canal (SC), which together form the tissues of the conventional AH outflow pathway[5]. Increasing evidence has demonstrated that reduced TM cellularity with aging is a critical pathogenic cause of POAG[6]. However, therapies to reverse the loss of TM cells in POAG are very limited[7].

Advances in stem cell technology have allowed the generation of an autologous stem cell type, known as induced pluripotent stem cells (iPSCs), through reprogramming of somatic cells[8]. The similarities between iPSCs and embryonic stem cells make iPSCs an attractive cell type for treating various degenerative diseases, including POAG. The culture of iPSCs in media conditioned by human TM cells has become a routine approach to generating TM-like cells[9–11]. These iPSC-derived TM (iPSC-TM) cells have been shown to have the ability to rescue glaucomatous phenotypes in many glaucoma models, including Tg-MYOC[Y437H] mice and sGCα1-deficient mice[12–14]. In these animals, transplantation of mouse iPSC-TM reverses the loss of TM cells. In ex vivo human eyes with either age-related decline in TM cellularity or experimentally damaged TM, transplantation of human iPSC-TM cells

also restores dysfunctional TM[10,15]. These advancements suggest that iPSC-based therapy may be a feasible approach to restore TM function to control IOP in POAG. However, the molecular mechanisms leading to TM regeneration remain largely unknown, which hinders translation toward the use of iPSCs in glaucoma therapy. In this study, we aimed to answer two important questions: (i) Are there subpopulations of iPSC-TM cells that are specifically involved in regenerating damaged tissues? and (ii) Which tissues of the conventional outflow pathway benefit from iPSC-based therapy?

Single-cell RNA-sequencing (scRNA-seq) and single-nucleus RNA sequencing (snRNA-seq) have become powerful approaches to uncover cellular mechanisms. In the conventional outflow pathway, this technique has enabled van Zyl et al.[16,17], Patel et al.[18], Thomson et al.[19], Hamel et al.[20], Balasubramanian et al.[21], and Wu et al.[22] to identify distinct cell types, including the expected myofibroblast- and fibroblast-like TM cells, SC endothelial cells, as well as some unexpected cell types, such as Schwann cells and macrophages. In addition, these studies mapped the expression of glaucoma-related genes to specific cell types[20,23,24]. For example, *myocilin* is expressed by myofibroblast and fibroblast-like TM cells, but also by smooth muscle cells. These studies have advanced our knowledge of the diversity of cell types in conventional outflow tissues and helped to identify those associated with glaucoma.

Herein, we employed scRNA-seq and bulk RNA sequencing to address whether specific subpopulations of iPSC-TM cells contribute to TM regeneration and how they achieve regeneration. Through comparison with primary TM (pTM) cells, we first identified a cluster of iPSC-TM cells with a distinct transcriptional pattern associated with integrin signaling and retention of pluripotency. Subsequent functional tests revealed this cluster, marked by *ITGA6* expression, as a primary stimulus for the proliferation and rejuvenation of endogenous TM cells. Moreover, we mapped the portions of conventional outflow tissues that benefit from iPSC-derived ITGA6+ cells by histological analysis. Finally, we identified the molecular mechanisms underlying the regenerative potential of this specific cluster of iPSC-TM. Together, these data provide crucial information about the molecular events initiated by transplanting iPSC-TM cells and represent an important step toward the translation of this regenerative approach to clinical practice.

## Results
### Characterization of cultured human TM cells at the single-cell level
To comprehensively profile the transcriptome of cultured human TM cells, we performed scRNA-seq of human pTM cells, iPSC-TM cells induced by the conditioned medium method (iPSC-TM CM)[12], and iPSC-TM generated using the recombinant cytokine approach (iPSC-TM RC)[25]. The pTM cells were obtained by isolating cells from the TM of a human donor and were characterized by immunohistochemical staining for TM markers, testing for dexamethasone-inducible myocilin secretion, and formation of cross-linked actin network (CLAN) (Supplementary Fig. 1). We analyzed a total of 7,444 cells of high quality, including 2795 pTM cells, 2322 iPSC-TM CM cells, and 2327 iPSC-TM RC cells (Fig. 1a–c). To determine whether subpopulations exist within the cell populations, we performed clustering analysis and identified six, four, and three clusters in pTM, iPSC-TM CM, and iPSC-TM RC, respectively, indicating higher heterogeneity among pTM cells when compared to iPSC-TM cells (Fig. 1a–c).

To understand the correlation of these cultured TM cell types to uncultured cells in the human TM, we compared our findings to those reported in the recently published scRNA-seq datasets of human outflow tissues[16–18]. These studies reported very distinct cluster numbers and annotations, likely due to different analysis parameters applied by the investigators. In our integrated analysis, we applied uniform parameters resulting in similar clusters across both studies (Fig. 1d and

Supplementary Fig. 2a-b), and included TM1 (Beam A/fibroblast-like), TM2 (Beam B/myofibroblast-like), fibroblast, JCT, vascular endothelium, and lymphatic endothelium/SC. Moreover, we created signature gene modules using the markers identified in the integrated dataset and calculated the corresponding module scores of cultured TM clusters. We observed that pTM clusters exhibited high similarity to TM1, TM2, and JCT cells from human tissues. iPSC-TM cells, particularly iPSC-TM RC, showed more discrepancies compared to tissues obtained in situ, but still retained substantial transcriptional equivalence (Fig. 1e).

To validate the similarity of pTM with cells within the human outflow tissues, we stained human donor tissues with markers of pTM clusters using IHC and FISH (Supplementary Fig. 3). CA12 and TGFBR2, which are expressed by cluster pTM1, are robustly expressed by all TM cells in situ. We also detected that collagen IV and VCAM1 are prominent biomarkers expressed throughout the entire TM and SC in situ. In contrast, CADM3 and FBLN2, representative of differentially expressed genes (DEGs) in cluster pTM2, are primarily expressed by cells located in uveal TM tissue. CRYAB in cluster pTM6 is localized in both TM and SC, but shows a higher expression near the inner wall of SC (Supplementary Fig. 3). Conversely, we did not detect DEGs representative of cluster pTM3 in the human tissues by IHC analysis (Supplementary Fig. 3). This cluster appears to represent actively proliferating cells that are expected to be absent in vivo. Characterization of pTM5 remains challenging as its typical DEGs cannot be readily classified. These include *ITGA8*, primarily expressed in the inner wall of Schlemm's canal[26], *RGS5*, a biomarker of pericytes, and *COL6A1/2*, robustly expressed in TM and fibroblasts. However, it is possible that pTM5 represents an artifact of in vitro cell culture, such as senescent pTM cells or those in the early stages of de-differentiation. Together, these data demonstrate the heterogeneity of cultured human TM cells and their relationship to TM cells, fibroblasts, and Schlemm's canal endothelial cells in situ.

### iPSC-ITGA6+ cells promote proliferation of TM in vitro
Previously we reported that iPSC-TM CM and iPSC-TM RC stimulate the proliferation of pTM cells in vitro[15,25]. We hypothesized that this is due to a subpopulation of cells that are shared by both iPSC-TM cell types, but are not present in pTM cells, since these do not exhibit pro-proliferative effects on other pTM. Here, we first conducted an integrated transcriptional analysis of iPSC-TM and pTM cells with low resolution (Fig. 1f, g and Supplementary Fig. 2a, b) to exclude clusters present only in either iPSC-TM CM or RC (cluster 1 in iPSC-TM RC; cluster 3 in iPSC-TM CM). After focusing on the remaining shared cluster (Fig. 1f, red frame), we observed a marked difference between iPSC-TM (CM and RC) and pTM. When high-resolution clustering settings were applied, this shared cluster was further divided into two major subclusters and one minor subcluster. One of the major subclusters is exclusively present in iPSC-TM CM and RC, but absent in pTM (Fig. 1h). This subcluster is characterized by expression of genes in the integrin family, including ITGA2, ITGA3, ITGA5, and ITGA6 (Fig. 1i). Pathway enrichment analysis further showed that this iPSC-related subcluster is enriched with pathways related to integrin cell surface interactions, extracellular matrix organization, and transcriptional regulation of pluripotent stem cells (Fig. 1j). In contrast, the pTM-related subcluster is enriched with genes related to the pathway of non-integrin membrane-ECM interaction (Fig. 1j).

Our scRNA-seq data showed that ITGA6 is specifically upregulated in iPSC-TM CM and iPSC-TM RC (Fig. 2a). We were further able to confirm by FACS that the protein expression of integrin alpha 6 is consistently observed in iPSC-TM induced by either method, but that the protein level is very low in pTM cells (Fig. 2b). Thus, we considered it likely that this iPSC-TM specific subtype of cells induces pTM proliferation and thereby mediates iPSC-based TM regeneration. In order to confirm this hypothesis, we isolated

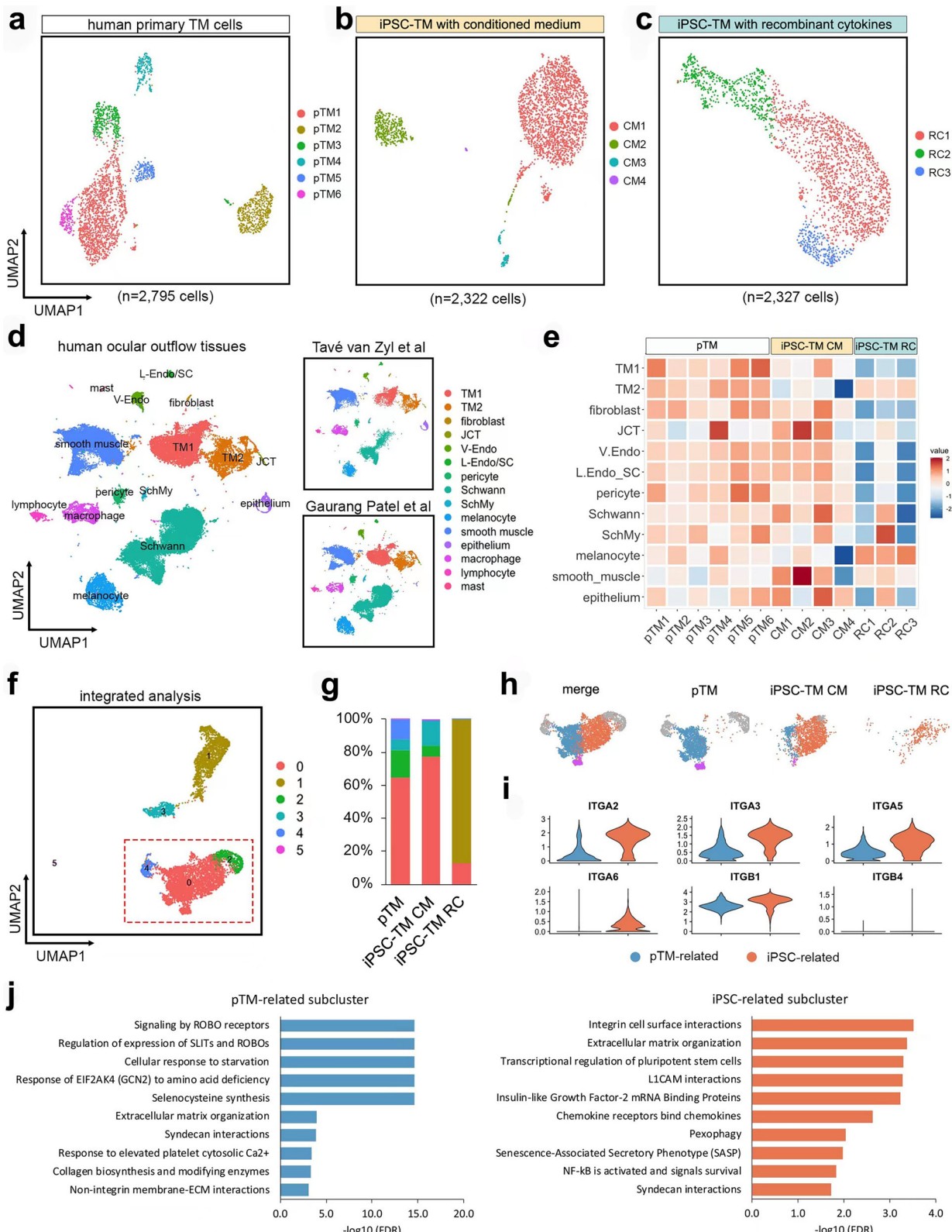

ITGA6+ cells from differentiated iPSCs using ITGA6 magnetic beads (Fig. 2c)[27]. We then analyzed the isolated cells by flow cytometry (Fig. 2d) and RT-PCR (Fig. 2e) to verify the expressions of DEGs of ITGA6+ cells, *STC1, NEAT1*, and *IER3* (Fig. 2f). Next, EGFP labeling was induced in pTM (EGFP-pTM) by lentiviral infection, and these were co-cultured with iPSC-TM CM preparations containing either 8.1% ITGA6-positive cells (ITGA6-) or 31.7% ITGA6-positive cells (ITGA6+;

Fig. 2g-i). EGFP-pTM co-cultured with the same number of unlabeled pTM cells were used as a negative control. Only ITGA6+ cell preparations significantly increased the number of EGFP-pTM cells, while ITGA6- iPSC-TM did not cause an increase in pTM proliferation when compared to the negative control (Fig. 2i). These findings were confirmed by using CellTracker Red CMTPX as an alternative method to label pTM (Supplementary Fig. 4a), and were repeated

**Fig. 1 | Integrated single-cell analysis of pTM and iPSC-TM. a–c** UMAP plots showing clusters in human primary TM cells (pTM), iPSC-TM induced with conditioned medium (iPSC-TM CM), and with recombinant cytokines (iPSC-TM RC). **d** UMAP plots (in different spaces) depicting the integrated analysis of published scRNA-seq datasets of human conventional outflow tissues[16,18]. Individual dataset plots are shown on the right. V-Endo: vascular endothelium; L-Endo: lymphatic endothelium; SchMy: myelinating Schwann cells. **e** Heatmap showing the normalized module scores of the clusters identified in cultured human TM cells. The top 500 marker genes from each non-immune cell cluster shown in (**d**) were used to generate each module. **f** Integrated cluster analysis at the low resolution of cultured human TM cells shown in the UMAP plot. The box outlined in red dashes indicates the shared cluster 0. **g** Cluster composition in pTM, iPSC-TM CM, and iPSC-TM RC. (**h**) Integrated cluster analysis with high resolution showing subclusters in the shared cluster shown in UMAP plots (1f). Blue: pTM-related subcluster; Orange: iPSC-TM-related subcluster; Purple: minor subcluster. **i** Violin plots showing the expression of integrin genes in the two major subclusters. **j** Bar plots showing enriched non-immune reactome pathways in pTM- and iPSC-TM-related subclusters.

using iPSC-TM RC (Supplementary Fig. 4b) and iPSC-TM cells derived from a second donor (Supplementary Fig. 4c-d).

## iPSC- ITGA6⁺cells restore aqueous humor outflow in glaucomatous mice

To determine whether iPSC-ITGA6⁺ cells are capable of iPSC-based tissue regeneration in vivo, we determined IOP and AH outflow in two mouse glaucoma models using magnetically labeled iPSC-TM. Consistent with our previous observations[28], labeling with the nanoparticles did not change iPSC-TM cell morphology (Fig. 3a, b). In our first experiment, we induced IOP elevation through injections of Ad5-MYOC$^{Y437H}$-EGFP[29,30]. We then injected preparations of nanoparticle-labeled iPSC-TM containing either 22.8% or 40.6% iPSC-ITGA6⁺ cells into the anterior chamber and steered these toward the TM using a ring magnet. Control animals received injections of an equal amount of PBS. IOP was monitored every week until the end of the experiment, 45 days after transplantation (Fig. 3c, d).

Consistent with previous reports[30], injection of Ad5-MYOC$^{Y437H}$-EGFP ($n = 8$) significantly increased IOP when compared to eyes that received injections of adenovirus expressing only EGFP (Ad5-EGFP) (Day 25: 17.8 mmHg *vs.* 14.1 mmHg, $P = 0.0001$, $n = 8$) or PBS (Day 25: 17.8 mmHg *vs.* 14.6 mmHg, $P = 0.0002$, $n = 10$). Eyes that received preparations containing 40.6% iPSC-ITGA6⁺ cells displayed a 44.4% decrease in IOP 24 days after transplantation (day 49), which is a significantly stronger effect than that observed in eyes that received preparations containing only 22.8% iPSC-ITGA6⁺ cells (Fig. 3c, d; $P = 0.0017$). However, IOP continued to decrease in eyes that received preparations containing 22.8% iPSC-ITGA6⁺ cells, and 38 days after transplantation (day 63) these eyes displayed IOPs that were similar to both WT and eyes that had received 44.4% ITGA6⁺ cells. Normal IOP was maintained in all treated eyes until the end of the experiment (day 70). A significant reduction in IOP was not observed following the injection of PBS.

We further confirmed the therapeutic effect of iPSC-ITGA6⁺ cells in transgenic mice (Tg-MYOC$^{Y437H}$), the most appropriate model to examine the pathogenesis of POAG (Fig. 3e, f). Here, 10-month-old animals with IOPs over 19.2 mmHg were chosen and received nanoparticle-labeled iPSC-TM containing either 3.9% or 49.7% ITGA6⁺ cells (Supplementary Fig. 5a, b). Eight weeks after transplantation, a significant decrease in IOP was found in Tg-MYOC$^{Y437H}$ mice receiving 49.7% ITGA6⁺ cells when compared to mice receiving 3.9% ITGA6⁺ cells ($P = 0.011$) (Fig. 3e). The outflow facility ($C_r$) was determined in enucleated eyes (Supplementary Fig. 6)[31]. We found that $C_r$ in Tg-MYOC$^{Y437H}$ treated with 49.7% ITGA6⁺ cells was 18.9 nl/min/mmHg, significantly higher than that observed in mice treated with 3.9% ITGA6⁺ cells (9.9 nl/min/mmHg, $P = 0.04$; Fig. 3f).

Taken together, these data demonstrate that iPSC-TM contain a subtype of cells that is characterized by the expression of ITGA6⁺ and is capable of inducing pTM proliferation in vitro and reduction of IOP in vivo. This suggests that this subclass of cells, which we will refer to as iPSC-ITGA6⁺ cells from here on, is crucial for functional recovery of glaucomatous TM.

## Molecular characterization of iPSC-ITGA6⁺ cells

In order to further characterize the molecular profile of iPSC-ITGA6⁺ cells we examined global transcriptional differences between ITGA6⁺ cells (33.0% ITGA6⁺ cells), ITGA6⁻ cells (14.0% ITGA6⁺ cells), and pTM and confirmed these with RT-PCR (Fig. 4a). Our results indicate significantly higher expression of TM biomarker, *AQP1*, in ITGA6⁺ cells when compared to ITGA6⁻ cells. However, we also observed elevated expression of the pluripotency markers *ABCG2, SOX2, NANOG*, and *NOTCH1* in iPSC-ITGA6⁺ cells, whereas NES and POU5F1 (Oct4) were expressed at lower levels. These data suggest that iPSC-ITGA6⁺ cells retain a higher degree of pluripotency and may thus represent a less defined stage during the induction of the TM cellular phenotype.

Consistent with this notion, iPSC-ITGA6⁺ cells are small and round, morphologically resembling iPSCs, whereas iPSC-ITGA6⁻ cells are spindle-shaped, further supporting the idea that iPSC-ITGA6⁺ cells retain characteristics of pluripotent stem cells[32]. Further, gene ontology analysis also indicates an enrichment of genes in iPSC-ITGA6⁺ cells that are associated with developmental processes of numerous tissues, including muscle, lung, glia, and neurons (Fig. 4b, c). However, iPSC-ITGA6⁺ cells are also enriched in genes involved in cell differentiation, adhesion, and proliferation (Fig. 4c) as well as those that are associated more directly with TM function. These include insulin signaling, which appears to play a role in the maintenance of IOP[33,34], and pathways regulating TGFβ production, a crucial factor mediating the production of extracellular matrix constituents by TM cells[35,36]. Further, analysis of DEGs identified *ILA1, SERPINE1, VAV3, COL8A1*, and *TYRP1*, which are all implicated in the development of POAG (Fig. 4d, e).

## iPSC-ITGA6⁺cells rescue cell loss of TM and SC

We are particularly interested in identifying which cell types involved in the conventional AH outflow benefit from iPSC-ITGA6⁺ cell transplantation. To explore this, we first evaluated the ligand-receptor interactions between iPSC-TM cells and aqueous outflow tissues using CellPhoneDB (https://www.cellphonedb.org/). Our analysis revealed 2182 and 2160 ligands or receptors in iPSC-TM and pTM, respectively, that could interact with receptors or ligands of major cell types in the outflow tissues, including TM1, TM2, fibroblasts, JCT, and lymphatic endothelium/SC (Supplementary Data 1 and 2). Among integrins, the interaction between the integrin alpha 6 and beta 1 complex and LAMC1 was detected only in iPSC-TM cells, but not in pTM (Fig. 5a). Based on their similarity to fibroblasts and endothelial cells, iPSC-ITGA6⁺ cells were characterized as a distinct cell type from both pTM and iPSC-ITGA6⁻ cells (Fig. 5b). Among the ligand-expressing cell types, we found that fibroblasts and SC cells are the primary sources expressing LAMC1 in the outflow tissues (Fig. 5c).

Based on this finding, we determined the cellularity of both TM and SC in WT, WT-EGFP, and Ad5-MYOC$^{Y437H}$-EGFP treated eyes 45 days after iPSC-TM transplantation of 22.8%, and 40.6% iPSC-ITGA6⁺ cells or PBS (Supplementary Fig. 7a). As previously reported in Tg-MYOC$^{Y437H}$ mice[12], injection of Ad5.MYOC$^{Y437H}$-EGFP results in significantly decreased TM cellularity compared to WT ($P = 0.001$) or WT-EGFP ($P = 0.0005$) (Fig. 5d). This loss in TM cellularity was fully restored by transplanting 40.6% ITGA6⁺ cells (24.7 *vs.* 16.8 cells/section, $P < 0.0001$), but not by transplanting 22.8% ITGA6⁺ cells (18.5 and *vs.* 16.8 cells/section, $P = 0.69$). We also observed that Ad5-MYOC$^{Y437H}$-EGFP transfection induced loss of cells along the inner wall of Schlemm's canal, though this was not statistically significant (15.4 *vs.* 18.8 cells/section, $P = 0.18$; Fig. 5e). These losses were significantly

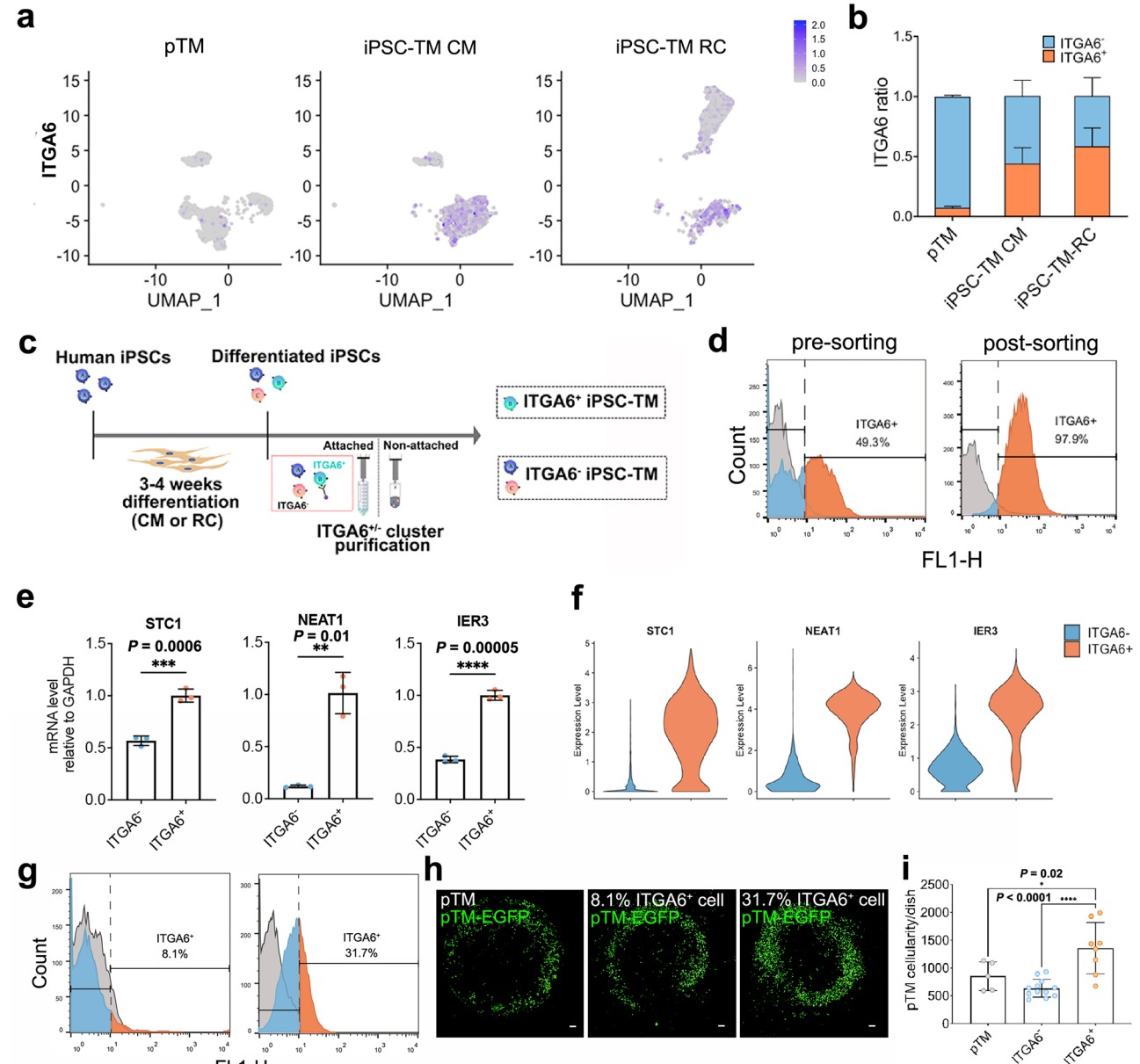

**Fig. 2 | Function of iPSC-ITGA6+ cells in vitro. a** Feature plots showing ITGA6 expression in pTM, iPSC-TM CM, and iPSC-TM RC. **b** Flow cytometry analysis of ITGA6-positive ratios in pTM (0.06 ± 0.02; *n* = 3 independent experiments), iPSC-TM CM (0.44 ± 0.14; *n* = 8 independent experiments) and iPSC-TM RC (0.58 ± 0.16; *n* = 2 independent experiments). Orange: ITGA6+ fraction; Blue: ITGA6⁻ fraction. **c** Schematic illustration of ITGA6+ iPSC-TM purification using a magnetic bead approach. **d** Flow cytometry histograms showing ITGA6+ proportions in two samples of iPSC-TM CM, with ITGA6+ contributions of 49.3% and 97.9%, respectively. **e** RT-PCR results showing higher expressions of *STC1* (0.6 ± 0.05 vs. 1.0 ± 0.06), *NEAT1* (0.1 ± 0.01 vs. 1.0 ± 0.20), and *IER3* (0.38 ± 0.03 vs. 1.0 ± 0.05) in iPSC-TM CM containing 97.9% ITGA6+ cells compared to iPSC-TM CM containing 49.3% ITGA6+

cells. \*\**P* < 0.01, \*\*\**P* < 0.001, \*\*\*\**P* < 0.0001 by two-tailed t test. *n* = 3 technical repeats. **f** Violin plot showing the expression of *STC1*, *NEAT1*, and *IER3* in ITGA6⁻ (blue) and ITGA6+ (orange) iPSC-TM cells. **g** Flow cytometry histograms showing ITGA6+ proportions in two iPSC-TM CM samples with ITGA6+ contribution of 8.1% and 31.7%, respectively. **h** Representative images of EGFP-pTM after 48 h of co-culture with the iPSC-TM CM samples. Co-culture with additional pTM cells was used as a control. Scale bars, 100 μm. **i** Quantification of EGFP-pTM cellularity following co-culture with pTM (853.2 ± 259.1 cells/dish; *n* = 5 technical repeats), ITGA6⁻ iPSC-TM (561.9 ± 113.8 cells/dish; *n* = 12 technical repeats), or ITGA6+ iPSC-TM (1453.7 ± 399.4 cells/dish; *n* = 8 technical repeats). \**P* < 0.05, \*\*\*\**P* < 0.0001 by one-way ANOVA with Tukey's post-hoc test.

reversed by transplanting cell preparations containing 40.6% ITGA6+ cells and, to a lesser extent, by those containing 22.8% ITGA6+ cells (40.6% ITGA6+ cells: 21.9 *vs.* 15.4 cells/section, *P* < 0.0001; 22.8% ITGA6+ cells: 18.6 *vs.* 15.4 cells/section, *P* = 0.032). Staining with MAB1281, a general marker of human cells used here to identify human iPSC-derived cells in the mouse eye, showed that only a limited amount of iPSC-TM cells remained in these mouse eyes 45 days after transplantation (Supplementary Fig. 7b and Fig. 5f), consistent with our previous findings[12,30].

Similar results were observed in the TM and SC in the eyes of Tg-MYOC^Y437H mice (Supplementary Fig. 7c and Fig. 5g). A significant increase in cell density in both TM and SC density was observed following transplantation of fractions containing 49.7% ITGA6+ cells (TM: 37.6 *vs.* 17.5 cells/section, *P* = 0.018; SC: 22.6 *vs.* 10.7 cells/section, *P* = 0.002). Staining with STEM121, a second marker of human cells, also indicated the presence of a small number of iPSC-TM cells 8 weeks after transplantation. Together, these findings indicate that iPSC-ITGA6+ cells exert pro-proliferative effects on both TM and SC cells and

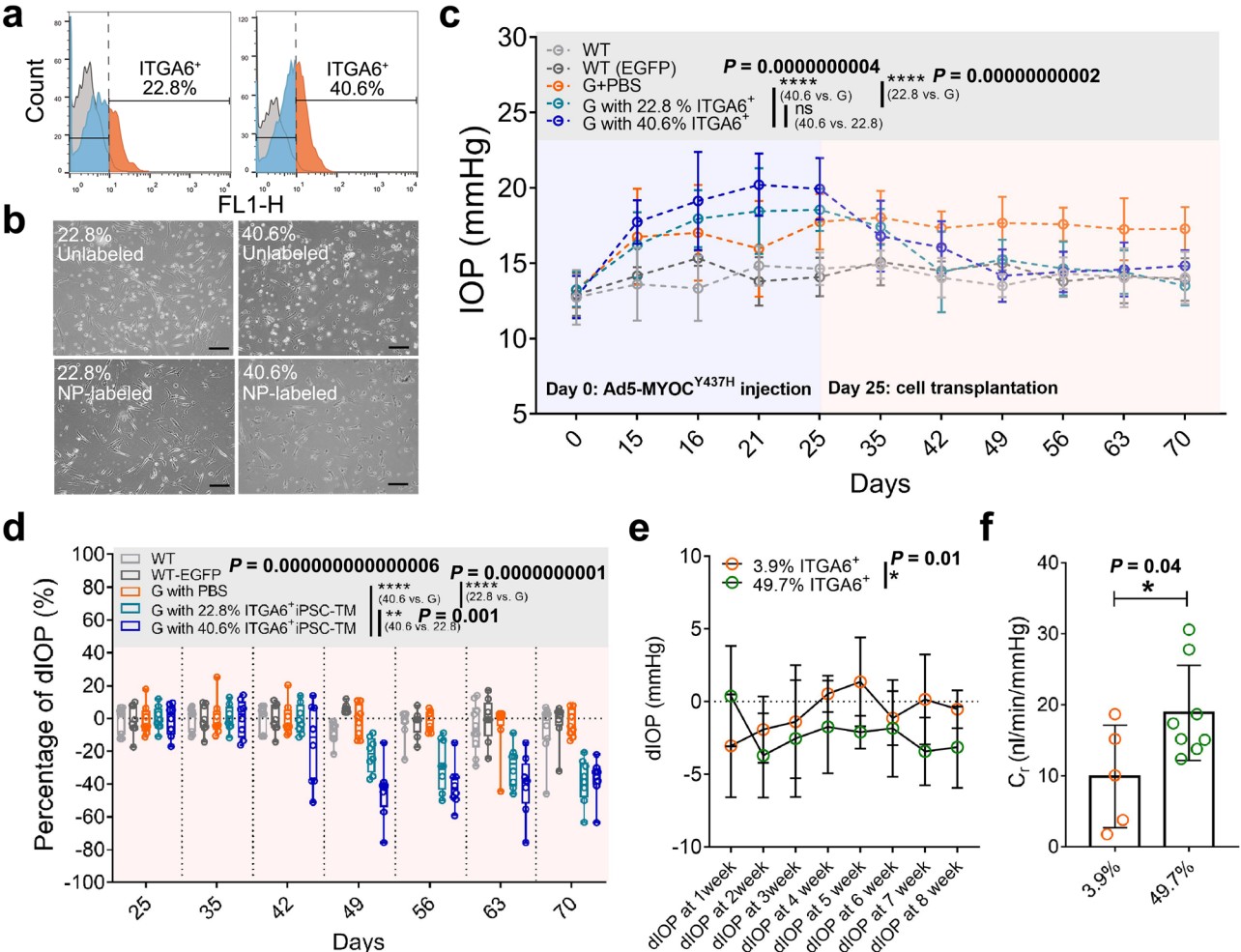

**Fig. 3 | In vivo function of iPSC-ITGA6⁺ cells in mouse models of glaucoma.**
**a** Flow cytometry histograms showing the proportion of ITGA6⁺ cells in three iPSC-TM CM samples. **b** Representative images of iPSC-TM morphology before and after purification, as well as after nanoparticle (NP) labeling. **c** IOPs in different groups of mice: naïve mice (light gray; $n = 10$ eyes), naïve mice following intracameral injection of Ad5-EGFP (dark gray; WT-EGFP; $n = 8$ eyes), glaucoma mice receiving injections of Ad5-MYOC[Y437H]-EGFP and sham (PBS) treatment (orange; $n = 8$ eyes); glaucoma mice receiving injections of Ad5-MYOC[Y437H]-EGFP and iPSC-TM containing 22.8% ITGA6⁺ cells (aquamarine; $n = 8$ eyes), and glaucoma mice receiving injections of Ad5-MYOC[Y437H]-EGFP and iPSC-TM containing 40.6% ITGA6⁺ cells (blue; $n = 8$ eyes). The blue/orange-shaded background indicates the period before/after iPSC-TM transplantation. ****$P < 0.0001$ by two-way ANOVA with Tukey's post-hoc test. **d** Percentage decrease in IOP (dIOP) in each group after cell

transplantation, corresponding to the orange-shaded period in (**c**). dIOP was calculated based on IOP measurements on day 25. The box's bounds and center are defined by the 25th, 50th, and 75th percentiles, and the ends of the whiskers represent the maximum and minimum values of the data. Eyes that received 40.6% ITGA6⁺ iPSC-TM exhibited a more significant decrease in IOP than those that received 22.8% ITGA6⁺ iPSC-TM. *$P < 0.01$, ****$P < 0.0001$ by two-way ANOVA with Tukey's post-hoc test. **e**, **f** dIOP (left, *$P < 0.05$ by two-way ANOVA with Tukey's post-hoc test) and outflow facility ($C_r$, right, 9.9 ± 7.2 vs. 18.9 ± 7.2 nl/min/mmHg, *$P < 0.05$ by two-tailed Student's t test) in Tg-MYOC[Y437H] receiving iPSC-TM CM at ITGA6⁺ contribution of either 3.9% (orange; $n = 5$) or 49.7% (green; $n = 8$). Significant IOP reduction and increased $C_r$ were observed in Tg-MYOC[Y437H] mice receiving 49.7% ITGA6⁺ cells compared to those receiving 3.9% ITGA6⁺ cells. Each data point represents an individual dIOP or $C_r$ measurement. Scale bars, 100 μm.

support our hypothesis that the restorative effects of iPSC-TM transplantation are largely due to re-cellularization of the tissue through proliferation of endogenous cells.

### Transcriptomic changes in pTM after co-culture

It is conceivable that the observed effect of iPSC-TM on pTM proliferation is the result of cell fusion between both cell types. However, exposure to lysophosphatidylcholine (LPC), a potent inhibitor of hemifusion and subsequent cell fusion[37], failed to impede the iPSC-TM's ability to stimulate pTM proliferation in co-culture (Fig. 6a). Moreover, LPC treatment did not alter the expression ratio of the sex-determining region Y gene (*SRY*), a specific Y chromosome gene, to *ACTB* in female pTM co-cultured with male iPSC-TM (Fig. 6a). These results are consistent with our previous observations in vivo and argue against cell fusion as a crucial mechanism[15].

To investigate possible mechanisms involved in iPSC-ITGA6⁺ cell-mediated pTM proliferation, we conducted bulk RNA-seq analysis of iPSCs, iPSC-ITGA6⁺ cells (33.0% ITGA6⁺ cells), ITGA6⁻ iPSC-TM (14.0% ITGA6⁺ cells), and pTM from donors 6 and 9 following co-culture (treated) with iPSC-ITGA6⁺ cells. Hierarchical clustering of the data resulted in the formation of three main clusters: one comprising pTM, the second comprising ITGA6⁻ iPSC-TM and co-cultured pTM, and the third containing iPSC and iPSC-ITGA6⁺ cells (Fig. 6b, c). Pathway analysis of these data revealed an enrichment of DEGs related to cell fate and developmental processes in co-cultured pTM (Fig. 6d, e, S8, and Supplementary Data 3), suggesting partial reprogramming or rejuvenation of pTM[38]. To gather further support for this hypothesis, we examined expression changes of individual genes associated with cellular rejuvenation in our RNA-seq data and confirmed these by RT-PCR (Fig. 6f). In co-cultured pTM increased

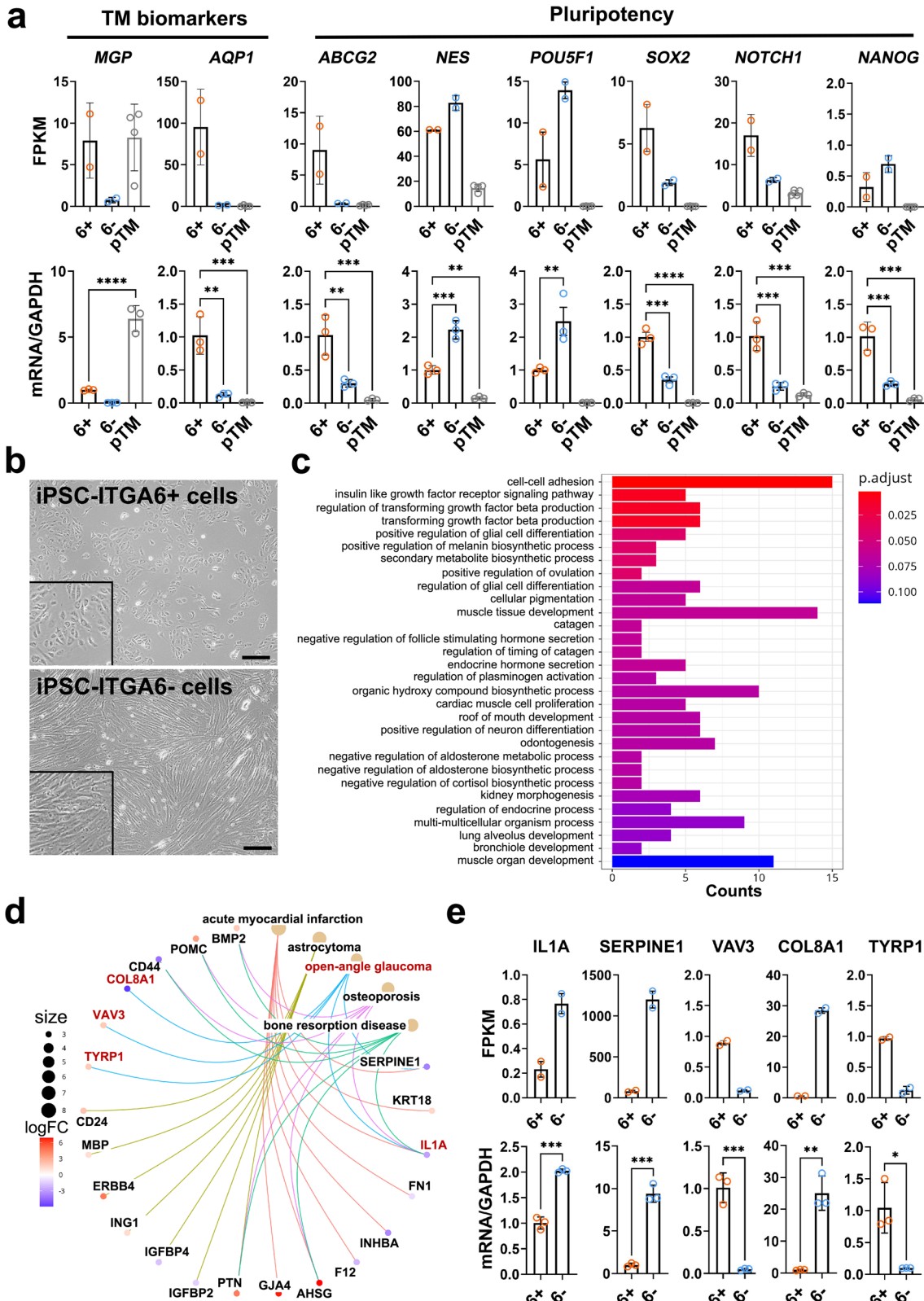

expression was observed for the stem cell biomarkers *NES*, *NOTCH1*, *POU5F1*, *SOX2*, and *NANOG* as well as the proliferation markers *MKI67* and *NEAT1*, supporting the notion that exposure to iPSC-ITGA6[+] cells causes cellular rejuvenation and proliferation in pTM. Expression of *AQP1*, a biomarker for TM cells also increased, but that of *MGP* was reduced, suggesting that this protein may be expressed less robustly in developing TM cells.

### NEAT1 participates in iPSC-ITGA6[+] TM function

In order to further unravel the molecular mechanisms underlying the pro-proliferative effect of iPSC-ITGA6[+] cells, we re-analyzed our scRNA-seq data to identify DEGs with fold changes > 2 in ITGA6[+] cells when compared to ITGA6-negative populations. These included 113 DEGs in iPSC-TM CM and 138 DEGs in iPSC-TM RC. Among these, *KLF6*, *ITGB1*, *NEAT1*, *COL1A2*, *COL1A1*, *MMP14*, *CTGF*, and *FN1* were identified in both

**Fig. 4 | Characterization of iPSC-ITGA6⁺ cells. a** FPKM values (upper panel; Supplementary Data 5) and RT-PCR results (lower panel; Supplementary Data 5) indicating higher expressions of *AQP1*, a biomarker for the TM, and several stem cell biomarkers, including *ABCG2, SOX2, NOTCH1*, and *NANOG*, in iPSC-ITGA6⁺ cells (orange) compared to ITGA6⁻ cells (blue) and pTM (gray). **\*\*P* < 0.01, \*\*\*P* < 0.001, \*\*\*\*P* < 0.0001 by one-way ANOVA with Tukey's post-hoc test. 6⁺: iPSC-ITGA6⁺ cells; 6⁻: iPSC-ITGA6⁻ cells. *n* = 3 technical repeats. **b** Representative morphology of iPSC-ITGA6⁺ and iPSC-ITGA6⁻ cells. Higher magnification views are shown in each image. Five experiments were repeated with similar observations. Scale bars, 100 μm. **c** Gene ontology (GO) analysis showing enrichment of genes involved in cell differentiation, proliferation, and some metabolic processes in iPSC-ITGA6⁺ cells. *P* values were calculated by the one-sided Hypergeometric test with the Benjamini-Hochberg correction. **d** Disease ontology (DO) analysis indicating five DEGs (red), *IL1A, SERPINE1, VAV3, COL8A1*, and *TYRP1*, that are highly associated with the pathogenesis of open-angle glaucoma (red). **e** FPKM values (upper panel; Supplementary Data 5) and RT-PCR results (lower panel; Supplementary Data 5) showing the expression of these five DEGs in iPSC-ITGA6⁺ cells (orange) and ITGA6⁻ cells (blue). *\*P* < 0.05, *\*\*P* < 0.0¹ *\*\*\*P* < 0.001 by two-tailed Student's *t* test. *n* = 3 technical repeats.

gene lists (Supplementary Fig. 9a), implicating them as candidates responsible for ITGA6⁺ cell-mediated cell division. NEAT1 was of particular interest since it is a major component of paraspeckles, spheroidal structures within the nucleus associated with phase separation and cell fate determination, and consequently is functionally associated with cell proliferation[39]. IHC analysis showed that NONO, a nuclear protein also involved in paraspeckle function, is more abundant in iPSC-ITGA6⁺ cells than in iPSC-derived ITGA6⁻ cells or pTM (Fig. 7a). Quantification analysis further supports these IHC observations, demonstrating a significantly higher amount of paraspeckles in iPSC-ITGA6⁺ cells (*P* < 0.0001; Fig. 7b, c).

In order to directly demonstrate a function of NEAT1 for iPSC-TM-induced pTM cell proliferation, we reduced NEAT1 expression in iPSC-TM using siRNA. In cultured iPSC-TM cells, a reduction of NEAT1 expression by 68.6% (Fig. 7d) not only impaired the expression of the important paraspeckle components NONO and SFPQ, but also significantly decreased the expression of ITGA6 and Ki-67, a proliferative biomarker (Fig. 7e). Co-culture of pTM with iPSC-TM treated by NEAT1 siRNA resulted in a 94.2% decrease in pTM proliferation when compared to those treated with control siRNA (*P* < 0.01, Fig. 7f). In contrast, *ITGA6* knockdown in iPSC-TM had only a minor effect on stimulation of pTM proliferation, likely due to anchoring deficiencies (Supplementary Fig. 9b–d). These results reveal that NEAT1 rather than ITGA6 is crucial for iPSC-ITGA6⁺ cell stimulated pTM proliferation in vitro.

We subsequently transplanted NEAT1-deficient iPSC-TM cells into the eyes of 2-month-old C57BL/6 mice to evaluate in vivo whether NEAT1 is required for iPSC-TM mediated pTM proliferation. Compared to eyes treated with normal iPSC-TM, we observed significantly lower TM and SC cellularity in eyes treated with NEAT1-deficient iPSC-TM (siRNA control vs. siRNA NEAT1: 43.1 vs. 37.4 endogenous cells/section, *P* = 0.030; *n* = 47–78 sections of 3 eyes in each group; Fig. 7g and S9E). The findings indicate that iPSC-TM without high levels of NEAT1 fail to induce proliferation in TM and SC.

In the above results, reduced regenerative responses could be due to a failure of NEAT1-deficient iPSC-TM cells to survive. To determine if this is the case, we compared pTM cellularity with the corresponding iPSC-TM cellularity in each co-culture sample, thereby minimizing any influence from iPSC-TM cellularity. As shown in Fig. 7h, the findings were consistent with the changes observed in pTM cellularity (Fig. 7f). In vivo we quantified the abundance of iPSC-TM implanted in the TM 2 days after transplantation of iPSC-TM treated with either NEAT1 siRNA or a siRNA control. Our findings demonstrate that siRNA-mediated reduction of NEAT1 expression does not significantly decrease the ability of iPSC-TM to integrate into the mouse TM (siRNA control vs. siRNA NEAT1: 33.1 vs. 32.3 iPSC-TM/section, *n* = 48–67 sections of 3 eyes in each group, *P* = 0.056; Fig. 7i). These findings demonstrate that NEAT1 expression in iPSC-TM is critically important for increased proliferation in pTM.

**Molecular mechanism of NEAT1-mediated iPSC-TM function**
To gain a better understanding of how NEAT1 specifically influences the function of iPSC-TM, we examined the trends in the pro-proliferative effects, paraspeckle quantity, and NEAT1 expression during TM development. Consistent with our previous findings, only iPSC-TM exhibits pro-proliferative capacity and a high quantity of paraspeckles (Fig. 8a). Notably, we found that *NEAT1_2*, the scaffold for paraspeckles, is robustly transcribed once iPSCs begin to differentiate and remains abundant in both iPSC-TM and pTM cells (Fig. 8b).

In addition to the expression level of *NEAT1_2*, another key factor influencing paraspeckle assembly is the stabilization of 3' RNA triple helix structure of *NEAT1_2*, which occurs through the cleavage of MEN β-associated RNA (menRNA). This small menRNA (59 nt) is thought to serve as a messenger to indicate NEAT1 promoter activity. However, the expression of menRNA during TM development showed an opposite trend compared to *NEAT1_2* (Fig. 8c).

These results suggest that the limited presence of menRNA in pTM cells may obscure NEAT1 promoter activity and inhibit paraspeckle assembly. To investigate this, we overexpressed menRNA in pTM cells and found that this overexpression significantly enhanced the formation of paraspeckles after 12 h (Fig. 8d, e). Additionally, we observed that the overexpressed menRNA, labeled with green fluorescence, was present in neighboring pTM cells that had been pre-stained with CellTracker Red (Fig. 8f and Supplementary Fig. 10). These findings not only clarify how menRNA induces a high presence of paraspeckles in iPSC-ITGA6⁺ cells but also illustrate how NEAT1 expression in iPSC-ITGA6⁺ cells has transcellular effects on adjacent pTM cells. Furthermore, overexpressing menRNA was found to directly rejuvenate pTM cells and stimulate their proliferation (Fig. 8g, h), suggesting potential alternative approaches for regenerating the trabecular meshwork.

## Discussion
Stem cell-based therapy represents a promising new approach to regenerating damaged TM in glaucoma. Our studies are focused on the use of iPSCs, a stem cell type that can be derived from autologous material, thereby lowering the risk of immune rejection in the recipients and avoiding ethical issues associated with other types of stem cells. As our previous studies have shown, iPSC-TM stimulates the proliferation of endogenous TM cells, leading to repopulation of the TM and restoration of AH outflow in glaucoma models. These findings demonstrate that even though endogenous TM cells do not typically proliferate in situ[40], they retain the ability to do so[12,15]. However, the mechanism through which iPSC-TM induces TM cell proliferation has remained elusive. Furthermore, it has remained unclear which subpopulation of TM cells benefits from iPSC-TM transplantation and how well iPSC-TM resembles these TM cell populations compositionally.

In our initial analysis, we determined the transcription patterns of human primary TM cell cultures using single-cell RNA sequencing. Primary TM cells could be grouped into six clusters, four of which corresponded well to those reported for uncultured TM cells isolated directly from donor eyes. However, TM cultures also contain cell types not observed in vivo, including proliferating cells (pTM3) and those that appear to have partially lost TM cell characteristics (pTM5). This is not unexpected because pTM cells grow readily in cell culture but are thought not to divide in the eye[40]. Cell behavior is also significantly influenced by the culture environment[41], and many cultured cells resemble those in vivo less closely as time in culture increases.

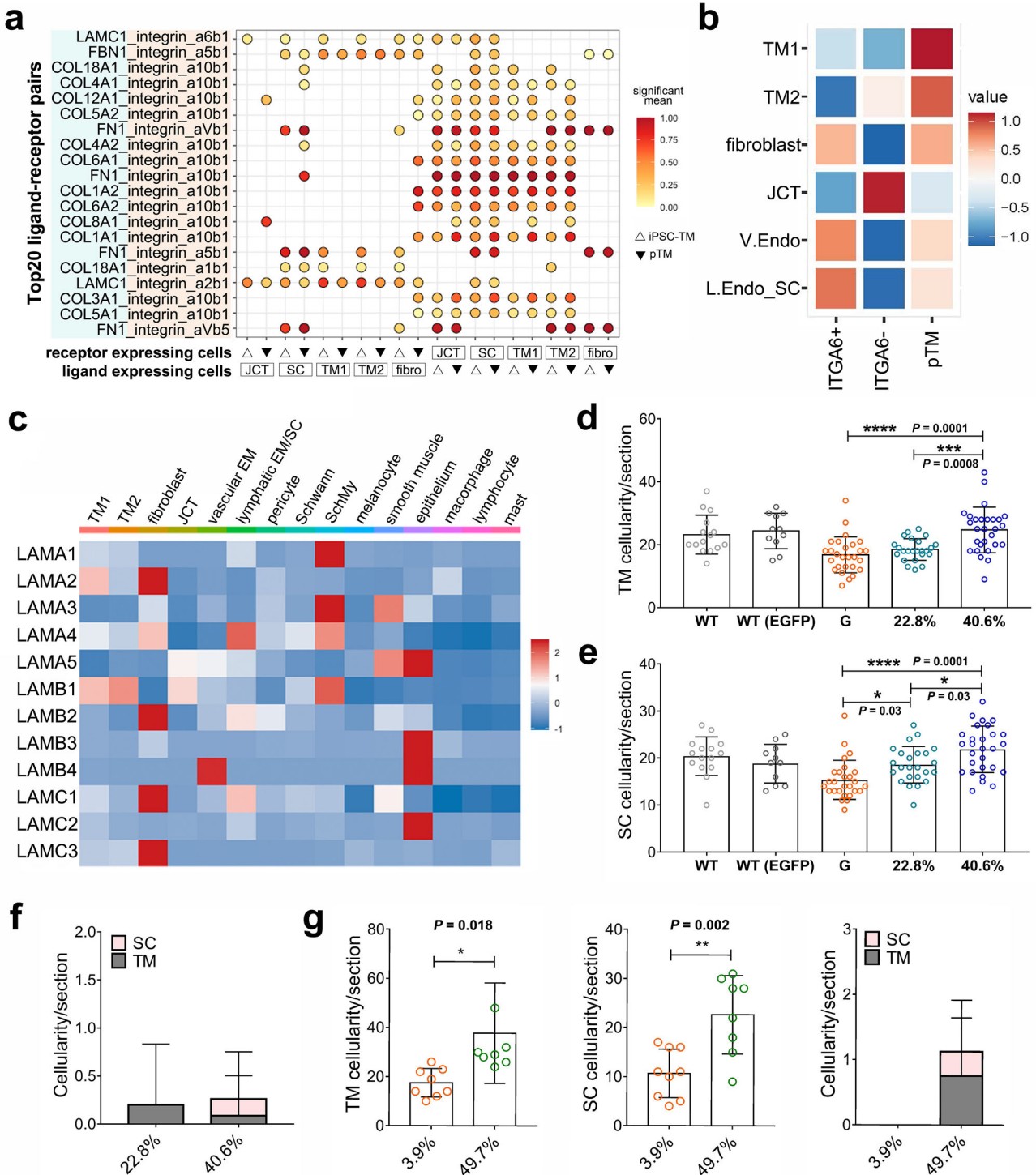

**Fig. 5 | Regeneration of TM and SC by iPSC-ITGA6+ cells. a** Dot plot showing significant integrin related ligand-receptor pairs between cultured human TM cells and major cell types in human conventional outflow tissues, including TM1, TM2, fibroblasts, JCT, and SC. **b** Heatmap displaying normalized module scores of clusters identified in ITGA6+ iPSC-TM, ITGA6- iPSC-TM, and pTM. **c** Heatmap showing normalized expression of LAMININ family genes across cell types of the conventional outflow tissues. **d, e** Quantification of cellularity in the TM and SC, indicating enhanced regenerative capacity of iPSC-TM with increasing proportions of ITGA6+ cells. *P < 0.05, ***P < 0.001, ****P < 0.0001 by one-way ANOVA with Tukey's post-hoc test. In WT (n = 10 eyes), WT-EGFP (n = 8 eyes), and glaucomatous eyes receiving PBS (n = 8 eyes), 22.8% (n = 8 eyes) and 40.6% (n = 8 eyes) iPSC-ITGA6+ cells, TM cellularity/section: 23.2 ± 6.2, 24.4 ± 5.7, 16.8 ± 5.7, 18.5 ± 3.4, and

24.7 ± 7.2; SC cellularity/section: 20.4 ± 4.1; 18.8 ± 4.1; 15.4 ± 4.2, 18.6 ± 3.9, and 21.9 ± 4.9; n = 15, 11, 27, 24, and 28 sections/group. **f** Quantification of MAB1281+ cellularity in the TM (gray) and SC (pink). MAB1281+ cellularity in the TM: 0.2 ± 0.6 and 0.09 ± 0.4; MAB1281+ cellularity in the SC: 0.0 ± 0.0 and 0.2 ± 0.5; n = 10 and 23 sections in glaucomatous eyes receiving 22.8% and 40.6% iPSC-ITGA6+ cells. **g** Quantification of cellularity in the TM (left), SC (middle), and MAB1281+ cellularity (right). *P < 0.05, **P < 0.01 by two-tailed Student's t test. In Tg-MYOC^Y437H mice receiving 3.9% (n = 5 eyes) and 49.4% (n = 8 eyes) iPSC-ITGA6+ cells, TM cellularity: 17.5 ± 5.8 and 37.8 ± 20.5; SC cellularity: 10.7 ± 4.9 and 22.6 ± 8.0; MAB1281+ cellularity in the TM: 0.0 ± 0.0 and 0.8 ± 1.2; and MAB1281+ cellularity in the SC: 0.0 ± 0.0 and 0.4 ± 0.5; n = 9 and 8 sections/group. One outlier has been excluded when quantifying TM cellularity in mice that received 3.9% iPSC-ITGA6+ cells.

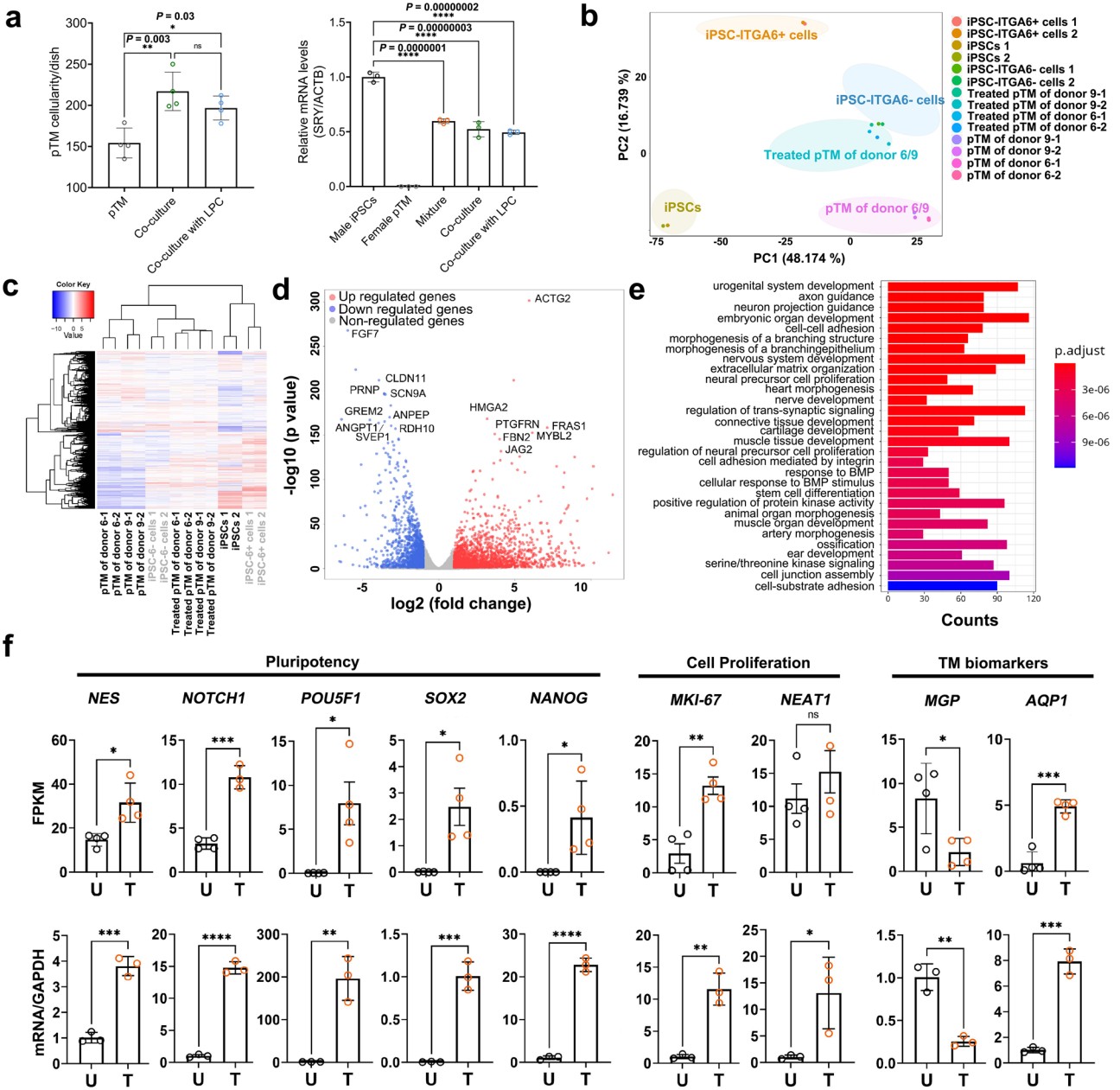

**Fig. 6 | Transcriptomic changes in co-cultured pTM. a** The capacity of iPSC-ITGA6+ cells to stimulate pTM proliferation (left panel; pTM/dish: 154.3 ± 18.2, 217.0 ± 23.4, and 196.8 ± 14.5 in pTM, co-cultures before and after LPC treatment) and the relative expression of *SRY* to *ACTB* in co-cultured samples (right panel) following LPC treatment. *n* = 4 technical repeats. In the right panel, pure iPSCs from donor U1 (male) and pTM of donor 6 (female) were used as positive and negative controls, respectively. The mixture contains male iPSC-TM and female pTM at a 2:1 ratio. Co-cultured samples treated with or without LPC (150 μM) for 48 h are shown. Fold changes in male iPSCs, female pTM, mixture, and co-culture before and after LPC treatment: 1.0 ± 0.04, 0.0 ± 0.0, 0.6 ± 0.02, 0.5 ± 0.07, and 0.5 ± 0.02. *n* = 3 technical repeats. *P < 0.05, **P < 0.01, ****P < 0.0001 by one-way ANOVA. **b** Principal component analysis (PCA) showing distinct transcriptomic profiles of iPSCs, iPSC-ITGA6+ and iPSC-ITGA6- cells, and pTM from donors 6 and 9 with or without co-culture for 7 days. Each group of samples contains two technique repeats. **c** Hierarchical clustering of DEGs in the same samples, consistent with the PCA results in panel B. **d** Volcano plot displaying DEGs in co-cultured pTM from donor 6, including genes involved in cell fate determination and proliferative regulation (Supplementary Data 3). *P* values were calculated by the two-sided Wald test with the Benjamini–Hochberg correction. **e** Enriched biological processes in co-cultured pTM from donor 6, including nervous system development, neural precursor proliferation, integrin-mediated cell adhesion, and cell junction assembly. *P* values were calculated by the one-sided Hypergeometric test with the Benjamini-Hochberg correction. **f** FPKM values (upper panel; *n* = 2 human donors examined over 2 independent experiments; Supplementary Data 5) and RT-PCR results (lower panel; *n* = 3 technical repeats; Supplementary Data 5) showing higher expressions of stem cell markers (*NES, NOTCH1, POU5F1, SOX2,* and *NANOG*), proliferation markers (*MKI-67* and *NEAT1*), and TM markers (*MGP* and *AQP1*) in pTM after co-culture. *P < 0.05, **P < 0.01, ***P < 0.001, ****P < 0.0001 by two-tailed Student's *t* test. U: untreated pTM; T: treated pTM.

Consistent with the reported clusters in conventional outflow tissue, we found that the inner layers of the TM mainly consist of pTM2 (fibroblast-like cells), while the outer regions are primarily covered with pTM1 and 6 (TM1-like cells), pTM4 (JCT-like cells) and pTM5 (mixture), some of which are endothelial cells. Of note, the distinct biomarkers for these clusters have been identified, including *FABP4, DCN, PDGFRA,* and *BMP5* for TM1-like cells, *ANGPTL7, CHI3L1, RSPO4,* and *FMOD* for JCT-like cells, and *POSTN, TFF3, MMRN1,* and *FLT4* for L-

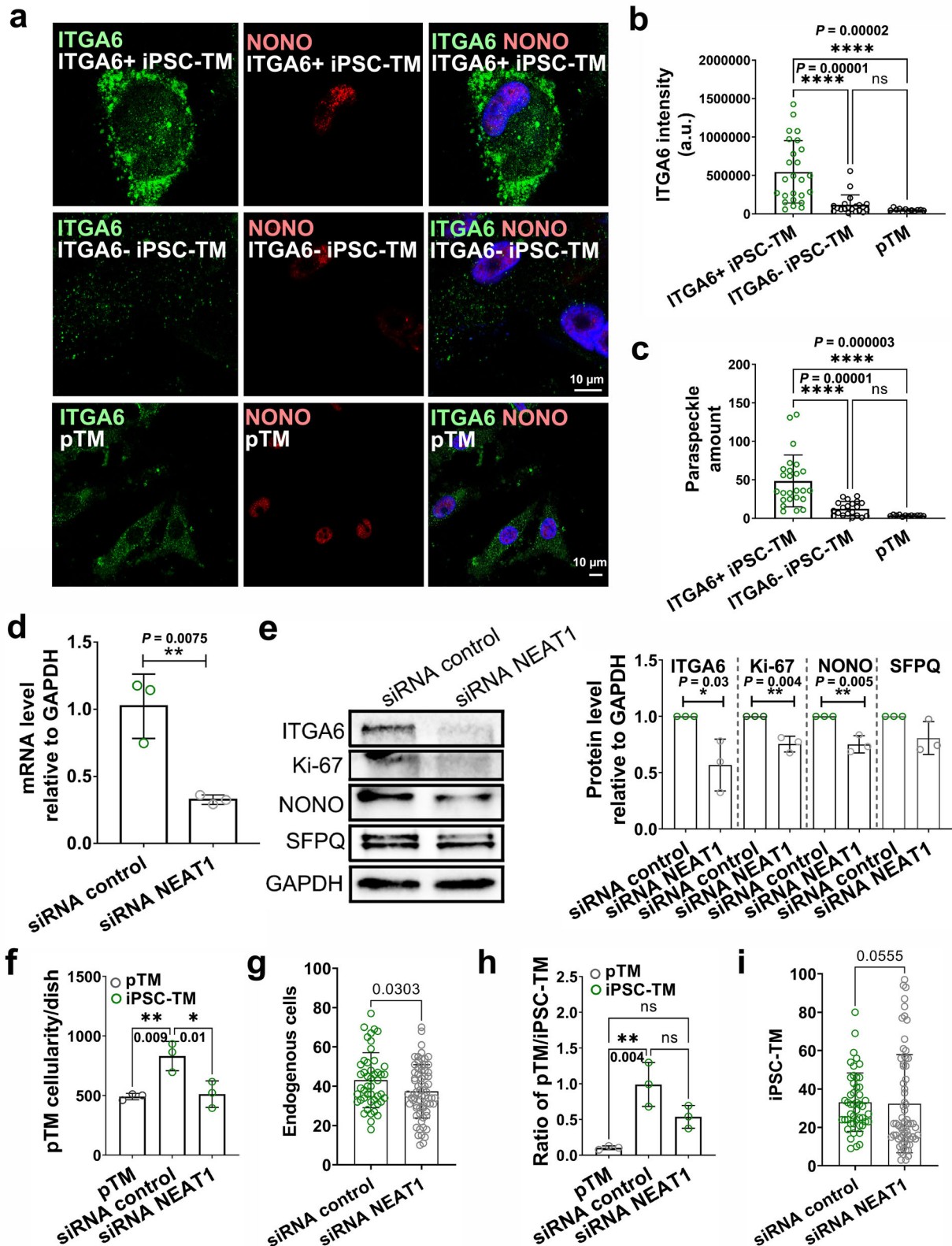

Endo/SC-like cells[17,18]. These findings not only enable us to purify the clusters of interest for further investigation but also enhance our understanding of the distinct functions of different regions in draining AH as well as the regions hosting iPSC-TM.

The ability of iPSC-TM to stimulate endogenous TM cell division has been documented in several glaucoma models, including GC knockout mice[14] and human perfusion organ culture models[10,15]. Since

pTM can be stimulated to proliferate by iPSC-TM but not by additional pTM, we reasoned that this effect may be mediated by an iPSC-TM subtype that is not present among pTM. Indeed, we were able to identify an iPSC-TM-specific cell cluster that is characterized by the expression of ITGA6. These ITGA6+ cells differ from pTM and ITGA6- cells in several aspects, including their cell morphology and elevated expression of the pluripotency markers *SOX2* and *NOTCH1*. As such, it

**Fig. 7 | Role of NEAT1 in the induction of pTM proliferation.**
**a** Immunohistochemical staining of ITGA6 (green) and NONO (red), a paraspeckle-associated nuclear protein, with nuclei counterstained in blue. Scale bars, 10 μm.
**b, c** Quantification of ITGA6[+] intensity (**b**) and paraspeckle abundance (**c**) showing a significant increase in paraspeckles in iPSC-ITGA6[+] cells CM. ****$P < 0.0001$ by one-way ANOVA with Tukey's post-hoc test. In iPSC-ITGA6[+] sample, ITGA6[-] iPSC-TM, and pTM, ITGA6 intensity: $547820 \pm 405961$, $117704 \pm 129287$, and $52989 \pm 15951$; Paraspeckle amount: $49 \pm 34$, $13 \pm 9$, and $4 \pm 0.9$; $n = 25$, 20, and 12 cells/group over 2 independent experiments. **d** RT-PCR results showing reduced NEAT1 expression in iPSC-TM after transfection with NEAT1 siRNA (siRNA control vs. siRNA NEAT1: $1.0 \pm 0.2$ vs. $0.3 \pm 0.03$). iPSC-TM transfected with scrambled siRNA served as a control. **$P < 0.01$ by two-tailed Student's $t$ test. $n = 3$ technical repeats. **e** Left: Western blot results of ITGA6, Ki-67, NONO, SFPQ, and GAPDH (top to bottom) in iPSC-TM transfected with NEAT1siRNA or scrambled control. Right: Quantification of band intensities analyzed using Image Lab software (Bio-Rad) showing significantly reduced expression of ITGA6 (siRNA control vs. siRNA NEAT1: $1.0 \pm 0.0$ vs. $0.6 \pm 0.2$), Ki-67 (siRNA control vs. siRNA NEAT1: $1.0 \pm 0.0$ vs. $0.8 \pm 0.06$), and NONO (siRNA control vs. siRNA NEAT1: $1.0 \pm 0.0$ vs. $0.8 \pm 0.08$) following NEAT1 knockdown. *$P < 0.05$, **$P < 0.01$ by two-tailed Student's $t$ test. $n = 3$ independent experiments. **f** Quantification of pre-stained pTM (red) cellularity, indicating decreased capacity of iPSC-TM CM to stimulate pTM cell proliferation following NEAT1 knockdown. *$P < 0.05$, **$P < 0.01$ by one-way ANOVA with Tukey's post-hoc test. In pTM, siRNA control, and siRNA NEAT1, pTM cellularity/dish: $491.7 \pm 27.0$, $831.7 \pm 122.5$, and $511.3 \pm 111.0$. $n = 3$ technical repeats. **g** Quantification analysis showing endogenous cellularity in the TM and SC regions 48 h after cell transplantation. siRNA control: $43.1 \pm 14.0$ endogenous cell/section ($n = 47$ sections from 9 eyes) vs. siRNA NEAT1: $37.4 \pm 13.5$ endogenous cell/section ($n = 68$ sections from 9 eyes), $P = 0.030$ by two-tailed Student's $t$ test. (**h**) is related to panel (**f**), showing a ratio of pTM cellularity compared to iPSC-TM cellularity in each co-culture sample. **$P < 0.01$ by one-way ANOVA with Tukey's post-hoc test. The ratios of pTM/iPSC-TM: $1.0 \pm 0.3$, $0.5 \pm 0.2$, and $0.1 \pm 0.03$, respectively. $n = 3$ technical repeats. (**i**) is related to (**g**), showing cellularity of STEM121-positive iPSC-TM in the TM and SC regions 48 h after cell transplantation. siRNA control: $33.1 \pm 15.2$ iPSC-TM/section ($n = 48$ sections from 9 eyes) vs. siRNA NEAT1: $32.3 \pm 25.6$ iPSC-TM/section ($n = 67$ sections from 9 eyes). $P = 0.056$ by two-tailed Mann–Whitney test.

is conceivable that ITGA6[+] cells represent iPSC during early stages of differentiation. Here, we demonstrate that ITGA6[+] cells are primarily responsible for inducing pTM proliferation. Preparations of iPSC-TM enriched in ITGA6[+] cells exhibit a significantly stronger pro-proliferative effect than those containing only a small number of these cells. Correspondingly, transplantation of an ITGA6-enriched preparation of iPSC-TM into the eyes of animal models of glaucoma had a much stronger effect on tissue regeneration, including IOP reduction and increased aqueous humor outflow facility, than those with low ITGA6[+] cellularity.

Following transplantation, iPSC-ITGA6[+] cells can be detected not only throughout the entire TM, but also along SC inner wall. The presence of iPSC-ITGA6[+] cells in SC is particularly intriguing since it is becoming increasingly clear that tissues downstream of the TM also contribute to AH outflow resistance[42]. This is largely because endothelial cells form a continuous monolayer on the inner wall of SC. When these cells experience a basal-to-apical pressure drop, numerous pores form. However, in glaucomatous eyes increased cell stiffness impairs the ability to form these pores, significantly hindering AH outflow[43]. In this study, we found that cell therapy not only restored the cellularity of the fibroblast-like cells in the TM but also reversed MYOC[Y437H]-induced endothelial cell loss in SC inner wall. These results strongly suggest that iPSC-ITGA6[+] cells may contribute to reversing the impaired biomechanical behavior of SC cells, thereby helping to control IOP in synergy with the TM.

The histological observations also support our previous finding that the limited presence of retained iPSC-TM after 2 months of transplantation renders them insufficient to sustain the burden of tissue repair in glaucoma (Fig. 5). Instead, in situ cell repopulation emerges as the primary mechanism, predominantly driven by the rejuvenation and proliferation of pTM following contact with iPSC-TM (Fig. 6). Notably, rejuvenation has been shown to be an effective strategy to delay the onset of aging-associated diseases. Epigenetic interventions, such as the regulation of non-coding RNAs, DNA methylation, and histone modification, have demonstrated efficacy in inducing rejuvenation[44]. Given that glaucoma is typically an age-related eye disorder[1], rejuvenation of pTM may become a novel focal point of interest (Fig. 6).

The precise role of ITGA6 in this process remains unclear. We previously demonstrated that iPSC-TM stimulation of TM cell proliferation requires direct cell-cell contacts[30]. Thus, one possible explanation is that integrin α6 simply facilitates the anchoring of transplanted cells to the TM and SC. Xiong et al.[45] reported that α5β1 integrin helps transplanted TM stem cells bind to fibronectin-enriched TM tissue. Integrin α6 and its heterodimers also have the capacity to bind to laminin[46], which is robustly expressed by L-Endo/SC and

fibroblast-like cells. This would make these regions a primary target for ITGA6[+] cells and explain why they tend to localize in these tissues following transplantation. Likewise, it is conceivable that ITGA6 binding to its ECM ligand initiates intracellular signaling cascades that alter the behavior of the ITGA6[+] cell population and its interactions with TM or SC cells[47].

Alternatively, it is possible that ITGA6 does not play any crucial functional roles in the observed processes, but rather represents a biomarker identifying a subpopulation of iPSC-TM that promotes TM regeneration through ITGA6-independent mechanisms. To investigate this hypothesis, we characterized additional DEGs in ITGA6[+] cells. Among the transcripts identified was NEAT1, a lncRNA that is a major constituent of paraspeckles – nuclear structures associated with phase separation and cell proliferation[48–51]. NEAT1 knockdown in ITGA6[+] cells decreases the expression of Ki-67, as well as that of the paraspeckle markers NONO and SFPQ, suggesting reduced paraspeckle abundance and diminished proliferative capacity. As we show herein using in vitro and in vivo functional tests, NEAT1, rather than ITGA6, is critical for iPSC-TM mediated pro-proliferative signaling to pTM.

To the best of our knowledge, this is the first report demonstrating a transcellular effect of NEAT1 expression or paraspeckle formation. Paraspeckles were relatively recently discovered and are typically found within the nucleus of cells[52,53]. They are located in the interchromatin space and are not known to be shared between cells. In this study, we demonstrated menRNA, a tRNA-like product of *NEAT1_2* and only 59 bp in size[54], could be transferred between cells. Once inside a recipient cell, menRNA contributes to paraspeckles formation and modifies the proliferative behavior of the recipient cells. One reason for this observation is that menRNA appears to be involved in the translational regulation of various RNA-binding proteins and may facilitate their recruitment to paraspeckles[55]. However, while paraspeckles are clearly associated with cell proliferation and maturation, the mechanisms through which these functions are carried out remain largely enigmatic. In addition, theories on how the observed transcellular effect may be mediated remain highly speculative and will require further investigation.

Notably, this strategy differs from tissue repair utilizing TM stem cells to replace lost cells[56] or mesenchymal stem cells for their paracrine secretion[57,58]. Those approaches require anchoring and survival of the transplanted material in the damaged tissue, whereas iPSC-ITGA6[+] cells neither exhibit nor require long-term survival. Instead, the primary role of iPSC-ITGA6[+] cells in tissue regeneration appears to lie in their capacity to rapidly, but temporarily, induce proliferation and rejuvenation of endogenous cells[30]. This characteristic reduces the tumorigenicity potential for undesirable long-term effects of

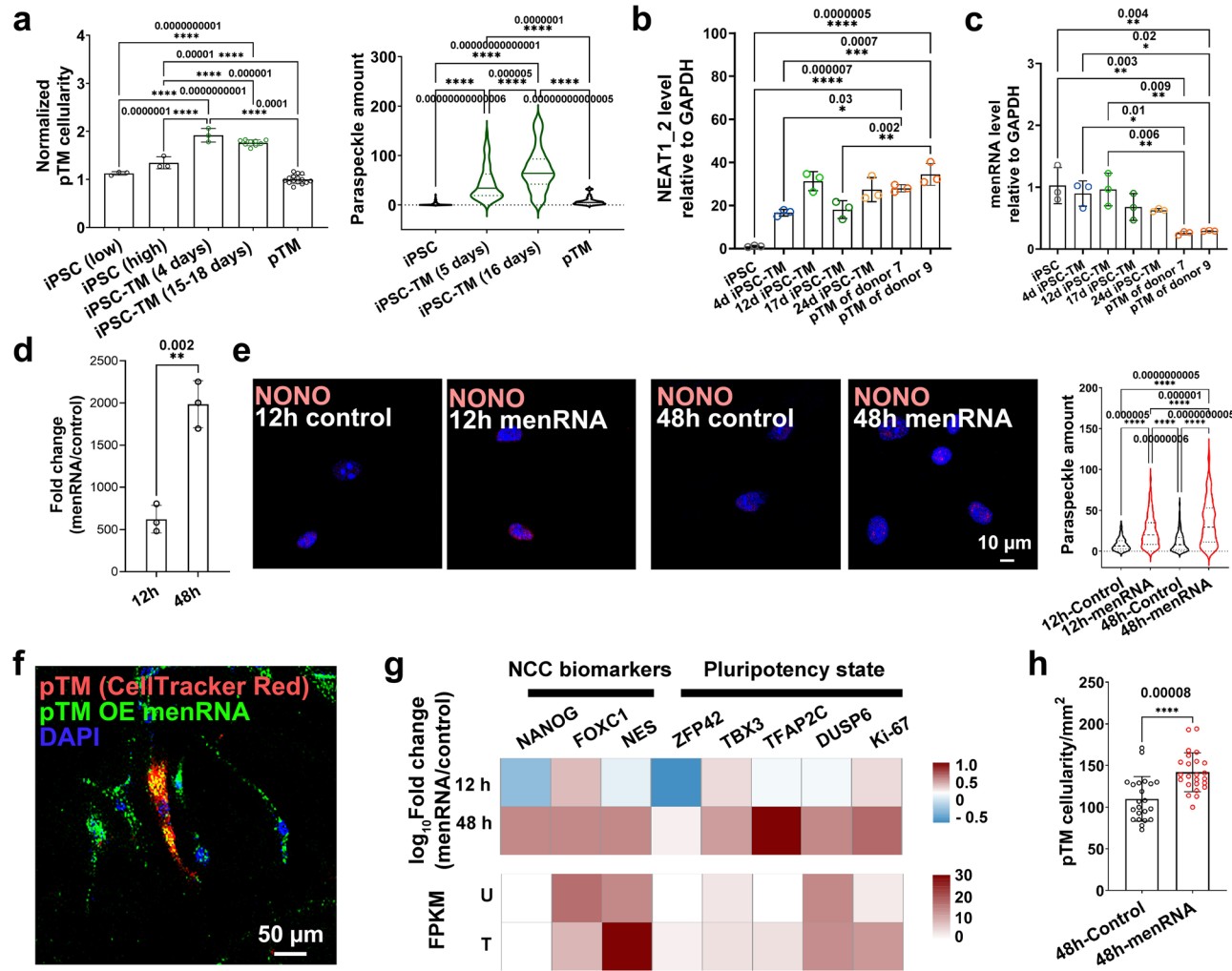

**Fig. 8 | menRNA stimulates TM cell rejuvenation and proliferation through enhancing paraspeckle assembly. a** Left panel: Quantification of pre-stained pTM cellularity following co-culture with iPSCs at low and high densities ($n = 3$ technical repeats), iPSC-TM after differentiation for 4 ($n = 3$ technical repeats) or 15–18 days ($n = 9$ technical repeats over 3 independent experiments), and pTM ($n = 15$ technical repeats over 5 independent experiments). The result indicates that iPSC-TM has a significantly greater pro-proliferative capacity compared to iPSCs and pTM (Supplementary Data 5). Right panel: quantification analysis (Supplementary Data 5) of paraspeckle showing that iPSC-TM (5 days: $n = 65$ cells; 16 days: $n = 51$ cells) has a significantly higher amount than both iPSCs ($n = 73$ cells) and pTM ($n = 29$ cells). RT-PCR results (Supplementary Data 5) showing a higher level of *NEAT1_2* (**b**) but a lower expression of menRNA (**c**) in pTM compared to those in iPSC-TM. $n = 3$. **d** RT-PCR results (Supplementary Data 5) confirming increased expression of menRNA in pTM after menRNA transfection for 12 and 48 h. $n = 3$. **e** Immunohistochemical staining of NONO (red; left panel), along with quantification of paraspeckle amount (right panel; Supplementary Data 5), indicating a significantly higher amount of paraspeckle in pTM after

overexpressing menRNA for 12 and 48 h. Scale bar, 10 μm. $n = 58, 153, 165,$ and 216 cells before or after menRNA treatment for 12 and 48 h. **f** A representative co-culture image of pTM pre-stained with CellTracker Red and pTM overexpressing menRNA pre-labeled with green fluorescence indicates the transfer of menRNA between cells. Scale bar, 50 μm. **g** Upper panel: a Log (Fold change) analysis using RT-PCR results showing induced expressions of neural crest cell (NCC) bio-markers (*NANOG, FOXC1, NES*), stem cell biomarkers (*ZFP42, TBX3, TFAP2C, DUS6*), and *Ki-67* in pTM after overexpressing menRNA. Lower panel: FPKM values of these genes are shown in pTM before (U) and after (T) co-culture with iPSC-TM. **h** Quantification of pTM after menRNA transfection for 48 h revealing menRNA overexpression significantly increased pTM cellularity (Supplementary Data 5). Cells transfected with LipoFiter™ transfection reagent were used as a control. $n = 22$ and 24 regions of pTM before and after menRNA transfection across 3 technical repeats. **P < 0.01, ****P < 0.0001 by two-tailed Student's $t$ test in (**d**) and (**h**), while *P < 0.05, **P < 0.01, ***P < 0.001, ****P < 0.0001 by one-way ANOVA with Tukey's post-hoc test in other panels.

transplantation and presents a distinct advantage for clinical translation of iPSC-based therapy for glaucoma. Consistent with our concern about the tumorigenicity of iPSC-based therapy, Bahranifard et al. recently reported that iPSC-TM generated using conditioned medium for eight weeks shows lower efficacy but higher tumorigenicity when compared to mesenchymal stem cells[59]. Our results indicate that prolonged differentiation reduces the repair capability of iPSC-TM (Fig. 8), suggesting that the duration of differentiation should be considered to generate powerful iPSC-TM in future studies. Additionally, the differing differentiation efficacies of iPSCs obtained from different species highlight the necessity of examining the

tumorigenicity of iPSC-derived cells from humans or non-human primates for the clinical translation of iPSC-based therapies[12,25,60].

In conclusion, our data suggest that the ITGA6⁺ subpopulation of iPSC-TM is crucial for the functional regeneration of conventional aqueous humor outflow structures in the eye. Robust expression of ITGA6 may only serve as a biomarker for this subtype of iPSC-TM cells, but ITGA6 could also facilitate anchoring these cells to the TM and SC inner walls. Instead, the function of iPSC-ITGA6⁺ cells depends on the abundance of paraspeckles. Both transplantation of ITGA6⁺ cells and strategies that enhance paraspeckle assembly emerge as promising strategies to regenerate TM function.

## Methods

### Animals

2-month-old C57BL/6 mice were purchased from Beijing Vital River Laboratory Animal Technology Co., Ltd. (Beijing, China). Transgenic mice expressing human myocilin$^{Y437H}$ (Tg-MYOC$^{Y437H}$) were a kind gift of Professor Val C. Sheffield (University of Iowa)[61]. Copy numbers of mutant myocilin in Tg-MYOC$^{Y437H}$ mice were determined by PCR using primers targeting human Tyr437His Myocilin (Forward: 5′-CGTGCCTAATGGGAGGTCTAT-3′; Reverse: 5′-CTGGTCCAAGGT-CAATTGGT-3′). 10-mon-old Tg-MYOC$^{Y437H}$ mice with copy number above one and at least three IOP measurements greater than 19.2 mmHg were selected for cell therapy. Mice were housed under standard conditions with a 12-h/12-h day/night cycle at $23 \pm 2\,°C$ temperature and $50 \pm 5\%$ humidity. All experiments were conducted according to the ARVO Statement for the Use of Animals in Ophthalmic and Vision Research and the laboratory animal care and use guidelines of Qingdao University Medical Center and Beijing Tongren Hospital.

### Conditioned medium-based differentiation

pTM cells from donors 5 to 8 were cultured in human complete medium, and the cell culture medium was collected after 7 days. The collected conditioned media (CM) from the cell cultures were pooled. After sterilization via filtration through a mixed cellulose ester membrane with $0.2\,\mu m$ pores (Millipore, Danvers, MA), the pooled CM were utilized for the differentiation of both human iPSCs (hiPSCs) and mouse iPSCs. Notably, all donors involved in this study were listed in Supplementary Table 1. hiPSCs were subjected to one-month differentiation, and the resulting cells were designated as iPSC-TM CM and collected for scRNA-seq, bulk RNA-seq, and cell transplantation studies.

### Xeno/feeder-free differentiation with recombinant cytokines

Human iPSCs at 5-10% confluency were differentiated by a xeno- and feeder-free approach with recombinant cytokines (RC)[25]. This method employs two stages: During the first stage (lasting 7 days) the cell culture medium is composed of MEM-α (Gibco), KnockOut serum replacement (10%; Gibco), TGF-β1 (2 ng/ml; Sino Biological, Beijing, China), NGF-β (100 ng/ml; Sino Biological), Erythropoietin (2 U/ml; Sino Biological). During the second stage (lasting 14 days) the medium contains two additional cytokines, PGF2α (10 μM; Sigma-Aldrich) and EGF (10 ng/ml; Sino Biological). The cell culture medium was changed every other day. Differentiated hiPSCs, designated iPSC-TM RC, were collected after completing both stages for scRNA-seq.

### Single-cell RNA sequencing (scRNA-seq)

scRNA-seq was carried out using the 3′ Library and Gel Bead Kit V2 (10× Genomics, 120237, San Francisco, CA) and the Chromium Single Cell A Chip Kit (10× Genomics, 120236). Single cells were suspended in 1× Dulbecco's phosphate-buffered saline (Thermo, 14190144) containing 0.04% BSA (wt/vol; Sigma). The cell suspensions were analyzed by CountStar (ALIT Life Science, Shanghai, China), and suspensions containing 300-600 living cells per microliter were then loaded onto the Chromium single-cell controller (10 × Genomics) to generate single-cell gel beads in the emulsion (GEMs). Captured cells were lysed, and the released RNA was barcoded through reverse transcription in individual GEMs. Reverse transcription was performed on S1000TM Touch Thermal Cycler (Bio-Rad, Hercules, CA) at 53 °C for 45 min, followed by 85 °C for 5 min, and held at 4 °C. The cDNA was generated, amplified, and quality assessed using an Agilent 4200 (Agilent Technologies, Santa Clara, CA). Single-cell RNA-seq libraries were constructed using Single Cell 3′ Library Gel Bead Kit V2 (10× Genomics). The libraries were sequenced using an Illumina Novaseq6000 sequencer (Illumina, San Diego, CA) with a sequencing depth of at least 100,000 reads per cell with a pair-end 150 bp (PE150) reading strategy. Single-cell RNA sequencing was performed in CapitalBio Technology Co., Ltd. (Beijing, China).

### Clustering analysis of scRNA-seq

Raw sequencing data of primary TM and iPSC-derived TM cells were processed with Cell Ranger pipelines (10× Genomics) and aligned to the human genome reference GRCh38. The following criteria were applied as quality control using Seurat R package[62]: cells that had fewer than 1000 UMI counts or more than 20% of mitochondrial genes were removed from further analysis. Genes that were expressed by fewer than 3 cells were also excluded. A total of 2795 primary TM cells, 2322 iPSC-TM cells with conditioned medium, and 2327 iPSC-TM cells with recombinant cytokines were kept for the downstream analyses. Following normalization, identification of variable features, and data scaling, cluster analysis of three scRNA-seq datasets was first performed individually. The top 30 principal components (PCs) were used to generate clustering with Uniform Manifold Approximation and Projection (UMAP). To identify the marker genes of each cluster in the primary TM dataset, differential expression gene (DEG) analysis was performed using Seurat with a minimum of 25 percent of cells that express the target gene and a log2 fold change threshold of 0.25.

Data integration using Seurat R package was performed to analyze the datasets of human conventional outflow tissues[63] and the de novo datasets of TM cells, respectively. Briefly, 2000 features were selected to identify the anchors for each dataset. The top 30 PCs at a resolution of 0.1 to 0.3 were used to generate UMAP clustering for both datasets.

### Calculation of module scores

The top 500 DEGs from non-immune cell clusters in datasets of human conventional outflow tissues were used to calculate the module scores of human primary TM[64]. DEGs that were not identified in our de novo scRNA-seq datasets of TM cells were excluded. The module scores were determined by using AddModuleScore function in Seurat[64]. The feature scores of individual clusters were visualized using heatmaps.

### Analysis of cell-cell interactions

Ligand-receptor interactions between human outflow tissues and primary TM or iPSC-TM cells were inferred using CellPhoneDB with the v4 database. The clusters selected for analysis included TM1, TM2, fibroblast, JCT, vascular endothelium and lymphatic endothelium/SC in human outflow tissues. The inferred interactions are considered significant when the $p$ value < 0.05, and the significant means of ligand-receptor pairs involved in integrins were plotted to compare primary TM or iPSC-TM.

### Magnetic purification

One million differentiated human iPSC-TM cells were incubated with 100 μl of 20 % (wt/vol) solution of FITC anti-human ITGA6 (Miltenyi Biotec, 130-097-245) for 10 min at 4 °C, centrifuged at 350 $g$, and resuspended in 100 μL 1×PBS buffer (Gibco) containing Anti-FITC Micro Beads (Miltenyi Biotec, 130-048-701), 0.5% bovine serum albumin (BSA; Sigma-Aldrich), and 2 mM ethylene diamine tetraacetic acid (EDTA; Sigma-Aldrich). ITGA6$^+$ cells were labeled and purified through LS columns (Miltenyi Biotec,130-042-401) placed in a magnetic field (Miltenyi Biotec, 130-042-501). The remaining non-magnetic, ITGA6-negative cells were collected after passing through the LS columns and purified with LD columns (Miltenyi Biotec,130-042-901).

### Flow cytometry analysis

To determine the ratio of ITGA6$^+$ cells in the samples, 30,000 purified cells were rinsed with 1×PBS (Gibco) and suspended in 1×PBS buffer containing 1 % FBS (Gibco). After incubation with FITC anti-human ITGA6 antibody (Miltenyi Biotec; 1:11) for 30 min at 4 °C, cells were rinsed and resuspended with 500 μl buffer for flow cytometry analysis. Cells with FSC and SSC intensities below approximately 10% of the peak signals are classified as debris and excluded. Cells with FL1-H greater than $10^1$ are identified as fluorescence-positive cells, and the fluorescence-positive cell ratio was analyzed by BD FACSCalibur

(Becton Dickinson). Unstained cells were used as a negative control. The Voltage and Amp Gain of the Forward scatter (FSC) were E00 and 1.00, respectively, and that of the side scatter (SSC) was 340 and 1.00. The voltage of fluorescence2 (FL1) was 381.

## Co-culture system

pTM cells were infected by recombinant lentivirus carrying EGFP multiplicity of infection (MOI) of 5 for 26 h at 37 °C, recovered in Biopsy medium for 6 days or stained with CellTracker Red CMTPX (5 μM for 30 min at 37 °C; Invitrogen, Carlsbad, CA, United States, C34552), and collected for the co-culture experiment. 3,000 pre-stained pTM cells were seeded on a borosilicate glass-bottom cell culture dish with 15 mm diameter (NEST, Wuxi, China). 6000 ITGA6[+/−] iPSC-TM cells were collected and seeded in the same dish. Co-culture with unstained pTM was used as a control. LPC (150 μM; Avanti POLAR LIPIDS, AL, United States, 855475P) was freshly prepared from the stock solution (15 mM) and used to block the potential hemifusion in the co-culture system. After co-culture for 48 h, cells were fixed in 4% paraformaldehyde (Thermo, Waltham, MA) for 20 min at room temperature. Cell nuclei were stained with 4, 6-diamidino-2-phenylindole (DAPI; Santa Cruz, Dallas, TX). Cells were imaged by confocal microscopy (Nikon, Tokyo, Japan). Quantification of cells with fluorescence used ×4 magnification images in a double-blinded fashion.

## PLGA-SPIO-Cypate nanoparticles labeling

The preparation of PLGA-SPIO-Cypate nanoparticles was carried out as described earlier[28]. Cells were maintained in the culture medium until reaching at least 70% confluency and incubated with the nanoparticles (50 μg/mL) for 10 h. Cells labeled with these nanoparticles have superparamagnetic properties without any change in both cell viability and characterizations, which can be exploited to improve the delivery accuracy towards the TM. Labeled cells were imaged by microscopy (Nikon, Japan).

## IOP measurement

Mice were anesthetized with 2.5% isoflurane and 80% (vol/vol) oxygen (RWD Life Science Co., Ltd, Shenzhen, China) for 5 min. IOP was measured by rebound tonometry (TonoLab, Helsinki, Finland)[12]. Measurements were taken between 9:00 and 12:00 am, and data represent the average of 3–5 measurements.

## Outflow facility measurement

The AH outflow facility was measured by a syringe-pump method[12]. The eyes were enucleated and cannulated with 30-gauge ½-inch length sterile needles (Becton Dickenson, Franklin Lakes, NJ) connected to a 100 μl Hamilton syringe (Hamilton, Reno, NV) mounted on a computer-controlled pump (AL-1000, World Precision Instruments, Sarasota, Florida), and continuously infused with 0.9% saline. The pressure was recorded by a flow-through pressure sensor (IcuMedical, San Clemente, CA), and the infusion rate was monitored by Hemolab software (Stauss Scientific, Iowa, IA; http://www.haraldstauss.com/HaraldStaussScientific/). Mouse eyes were perfused using a standard protocol: stabilization at 8 mmHg for 30 min followed by perfusion at six sequential steps at 6, 8, 10, 12, 14, and 16 mmHg. Flow rates (Q; nl/min) at the last 4 min of each step were averaged and used to generate the Q–P plot. The power law model fitted data in the Q–P plot. As described[65], the facility at 8 mmHg, designed $C_r$ (nl/min/mmHg), was calculated using MATLAB to indicate the capacity of TM in controlling AH outflow.

## Cellularity analysis

For quantification of TM and SC cellularity, images at ×20 or ×10 magnifications were taken of mouse tissues. Cell nuclei stained with DAPI (Santa Cruz) were counted for TM and SC cellularity analyses. MAB1281

and STEM121 staining can specifically label human cells, which has been employed to distinguish human iPSC-TM cells and other endogenous cells in animals. In each image, cells located in Schlemm's canal inner wall region were recognized as SC cells. Cells located in the region between the pigmented ciliary body and Schlemm's canal, excluding SC cells, were annotated as TM cells. Each group contained 10 to 17 eyes. Six cryosections from each eye were analyzed.

## Bulk RNA sequencing and bioinformatic analyses

RNA sequencing was carried out in human iPSCs, conditioned medium-derived TM-like cells (iPSC-ITGA6 +/ITGA6[−] cells CM), and human pTM cells (pTM) from donors 6 and 9, treated with or without iPSC-derived ITGA6[+] cells, on the HiSeq 2500 platform (CapitalBio Technology Co., Ltd.; Beijing, China). Total RNA was extracted using Trizol (Tiangen, Beijing) and assessed using Agilent 2100 BioAnalyzer (Agilent Technologies; Santa Clara, CA, USA) and Qubit Fluorometer (Thermo). A stranded mRNA library was prepared for sequencing mRNA and generating stranded mRNA information. The fragments per kilobase per million mapped reads (FPKM) value, reflecting the relative expression of a transcript, was compiled from the Cufflinks output. Genes with a $P$ value ≤ 0.01 and expression ratio ≥ 2 or expression ratio ≤ 0.5 were recognized as significantly differentially expressed genes (DEGs) between the two samples.

Total genes and DEGs were utilized to generate clustering diagrams using Cluster3.0 with the hierarchical method. The uncentered correlation and average linkage parameters were chosen to calculate gene distance and sample distance.

Gene Ontology (GO) enrichment and disease enrichment were analyzed using the command-line program KOBAS 2.0. The number of genes involved in the GO term set was counted, and $P$ values were calculated using a hypergeometric distribution.

Gene Set Enrichment Analysis was performed on all genes detected in all cells of one sample using GSEA software version 2.2.2.4, which utilizes predefined gene sets from the Molecular Signatures Database (MSigDB v6.2). Gene expression data was calculated by the mean UMI count of the gene in one cluster and the rest of the clusters, respectively. The criteria for selection of gene sets from the collection ranged from a minimum to a maximum of 0 and 500 genes, respectively.

## Statistical analysis

For two-group comparison, two-tailed Student's t-test, Mann-Whitney test, and normality and lognormality test were used for statistical analysis of CLANs formation, RT-PCR results, outflow facility data, ITGA6 intensity, and paraspeckle amount. For multiple group comparisons, one-way ANOVA with Tukey's post-hoc test was performed for statistical analysis of the cellularity data, and two-way ANOVA with Tukey's post-hoc test was applied for the statistical analysis of IOP results. ANOVA tables, including the the degrees of freedom and $F$-values, were provided in Supplementary Data 4. All tests were performed in GraphPad Prism 7. Data are expressed as the mean ± SD. $P$ values < 0.05 were considered to be statistically significant.

## Reporting summary

Further information on research design is available in the Nature Portfolio Reporting Summary linked to this article.

## Data availability

Raw data files have been uploaded to the National Library of Medicine of the National Center for Biotechnology Information with BioProject IDs: PRJNA948656 (scRNA-seq) and PRJNA1098785 (RNA-seq). Source data are provided with this paper. All raw and processed data in this study are available in Mendeley Data (https://doi.org/10.17632/55zfpf5t73.1). Source data are provided with this paper.

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

## Acknowledgements

The authors thank Prof. Val C. Sheffield at the University of Iowa for providing the Tg-MYOC^Y437H mice. This study was supported by the National Key Research and Development Program (2022YEF0132500), Taishan Scholar Youth Expert Program (tsqn202103055), Qingdao Key Technology and Industrialization Project (23-1-4-xxgg-16-nsh), Natural Science Foundation of Shandong Province (ZR2022YQ72), National Marine Pilot Laboratory of the Ministry of Science and Technology (10-02), Ophthalmology Joint Project of Qingdao University, Beijing Municipal Public Welfare Development and Reform Pilot Project for Medical Research Institutes (PWD&RPP-MRI, JYY2023-6), Beijing Natural Science Foundation (L212036), Beijing Municipal Institute of Public Medical Research Development and Reform Pilot Project (2018-2), National Key Research and Development Program (2018YFA0109500), Project of Qingdao Haier Biotech Co., Ltd., and Project of the Qingdao Science and Technology Plan Park Cultivation Program (25-1-1-yqpy-19-qy).

## Author contributions

Conceptualization: W.Z., M.H.K., N.W. Methodology: P.F., C.Y., S.W., X.Z., C.X., Z.Y., S.L., B.Z., Z.C., J.T. Investigation: C.Y., S.L., W.X., S.C., B.X., J.K., W.W., Y.W., G.X., X.Z., X.X., X.G., J.Z., Q.C. Funding acquisition: W.Z., N.W., Q.C., S.L. Supervision: W.Z., M.H.K., N.W. Writing—original draft: W.Z., C.Y., P.F; review & editing: M.H.K., N.W.

## Competing interests

Qilong Cao and Zhixin Yuan are affiliated with Qingdao Haier Biotech Co., Ltd., Shaoshuai Liang and Bingqiang Zhang are affiliated with Qingdao Restore Biotechnology Co., Ltd. The other authors declare no conflict of interest in this work. We have a granted patent related to this work.

## Additional information

[1]Department of Pharmacology, School of Pharmacy, Qingdao University, Qingdao, China. [2]Department of Ophthalmology, Duke University School of Medicine, Durham, NC, USA. [3]College of Marine Life Sciences, Ocean University of China, Qingdao, China. [4]Henan Academy of Innovations in Medical Science, Zhengzhou, China. [5]Beijing Institute of Ophthalmology, Beijing Tongren Eye Center, Beijing Tongren Hospital, Capital Medical University, Beijing Ophthalmology & Visual Sciences Key Laboratory, Beijing, China. [6]Beijing Institute of Brain Disorders, Collaborative Innovation Center for Brain Disorders, Capital Medical University, Beijing, China. [7]Department of Immunology, Duke University School of Medicine, Durham, NC, USA. [8]Qingdao Restore Biotechnology Co., Ltd., Qingdao, China. [9]Qingdao Haier Biotech Co., Ltd., Qingdao, China. [10]Biomedical center, Qingdao University, Qingdao, China. [11]Key Laboratory of Bio-Fibers and Eco-Textiles, Qingdao University, Qingdao, China. [12]Zhejiang Key Laboratory of Multiomics and Molecular Enzymology, Yangtze Delta Region Institute of Tsinghua University, Zhejiang, China. [13]Department of Inspection, Qingdao University, Qingdao, China. [14]Department of Ophthalmology and Visual Sciences, University of Iowa, Iowa City, IA, USA. [15]Center for the Prevention and Treatment of Visual Loss, Iowa City Veterans Affairs Medical Center, Iowa City, IA, USA. [16]Beijing Key Laboratory of Fundamental Research on Biomechanics in Clinical Application, Capital Medical University, Beijing, China. [17]These authors contributed equally: Pengchao Feng, Chen Yu, Xiaoyan Zhang, Bin Xu, Shen Wu. ✉e-mail: qd.wh@163.com; wningli@vip.163.com; wzhu@qdu.edu.cn

