## [Transparent Peer Review file · Nature Communications]

iPSC-derived alpha6 integrin-positive cells restore aqueous humor outflow in glaucoma eyes

Corresponding Author: Dr Wei Zhu

Version 0:

Reviewer comments:

Reviewer #1

(Remarks to the Author)

Zhu and colleagues have submitted an interesting work that extends their research on the therapeutic potential of iPSC-derived trabecular meshwork-like cells in glaucoma. In previous work, they have shown that cell therapy using iPSC-derived TM-like cells in mice can lower intraocular pressure and induce recellularization of the trabecular meshwork in MYOC-induced models of ocular hypertension. Here, they use single cell RNA sequencing to better characterize these cells before transplantation and identify potential molecules and pathways that might mediate the therapeutic effect. While the study is rigorous and interesting, novelty and impact is reduced by incomplete characterization of the cells, which makes it difficult to understand their true phenotype, how these iPSC-derived "TM cells" are related to in vivo trabecular meshwork cells, and how they mediate recovery of the aqueous outflow pathway.

Major questions:

- 1: The authors have identified a subset of cells present in their iPSC trabecular meshwork cell cultures that express ITGA6 and have the ability to trigger recellularization of the endogenous TM in mouse models of glaucoma. However, as described in detail throughout the manuscript, these cells are not present in primary TM cell cultures, and it is unclear if these cells are "iPSC-derived trabecular meshwork" or are a non-TM cell type that arose in the iPSC differentiation process. Additional characterization of these cells is needed to determine their true identity and if an analogous population occurs in vivo. Ascertain the actual identity of this cell type is critical to understanding how they may function in relation to TM repair.
- 2: Beyond their magnetic bead approach, have the authors considered altering their culture and/or differentiation conditions to further enrich for ITGA6+ cells or potentiate their formation? As these cells are supplying a stimulus that is not present in primary TM cells and appear to have a distinct transcriptional profile, optimizing culture conditions to generate maximally differentiated TM cells may not be the most effective method for obtaining outflow-targeted cell therapy candidates.
- 3: Authors should perform knockdown of ITGA6 in iPSC-TM cells to determine if this is the functional effector or merely a biomarker of the cells mediating protective effect.
- 4: Additional characterization of the potential role(s) for NEAT1 is required. Was expression of ITGA6 or any of the DEGs observed in ITGA6+ vs - iPSC-TMs altered in NEAT1 knockdown cells? Was proliferation of NEAT1 knockdown ITGA6+ iPSC-TM cells reduced during their 48-hour culture period with HTMs?
- 5: It is unclear why authors focused on NEAT1, a cell-autonomous mediator of cell proliferation, when it appears that iPSC-TM cells themselves are not repopulating the outflow tissues but rather encouraging endogenous cells to do so. Do iPSC-TM cells proliferate once embedded within the TM?
- 6: Authors have previously described a key role for Cx43 expression in iPSC-TM-mediated TM regeneration. Was Cx43 expression different in ITGA6+ and - cell populations? Was it altered in NEAT1 knockdown cells?
- 7: Authors state that only a limited number of transplanted cells remain after 45 days (figure 4 F-G). Is this due to transplanted cells dying, or is implantation efficiency low initially? Data from a shorter timepoint should be included.

Minor questions

- 1: In figure 1 A-C, it is unclear if the figures exist in the same UMAP space.
- 2: Based on the UMAP projection shown in Figure S2 B, it appears that ITGA6+ cells are located in cell clusters CM1 and RC2. Figure 1E would be clearer if this information was available without referring to the supplemental information.
- 3: The trajectory analysis presented in figure 2 A is not informative and should be removed. A more useful trajectory analysis might help to position the iPSC-TM cells within the context of the previously published datasets of outflow tissues.

4: On lines 142, it is unclear how cluster pTM5 was identified as Schlemm's canal. None of the marker genes identified for this cluster in figure S3 are unique to endothelial cells or Schlemm's canal.
5: In figure 4 D and E, all statistical comparisons described in the manuscript should be included on the plots.
6: Figure 3 C would be clearer if the X axis timeline was continuous or more clearly indicated the full time elapsed between IOP measurement at day 25 and post transplantation day 10. → Were cells transplanted on day 25?
7: In figure 4 C, labels obscure the most important section of the images. Arrows should be used instead. In addition, at the provided magnification it is difficult to appreciate the changes in cellularity. While authors can easily differentiate cell types for counting nuclei, in the format of a small figure panel specific staining of the cell types quantified would make this figure easier to follow.

Reviewer #2

(Remarks to the Author)

- 1) this is a strong and generally well-developed study of an important topic that will advance the field.
- 2) The authors have identified a subset of differentiated iPSCs that are more effective when transplanted into glaucomatous anterior segments in restoring intraocular pressure control.
- 3) In general, the methods and data analysis are sound and support the conclusion.
- 4) line 51-54 - wording is confusing, elevated IOP is the primary addressable risk factor instead of making age, which is not addressable anyway, seem more important
- 5) line 90-91 is a little overstated
- 6) a number of the figure captions are too small to read, especially the superscripts
- 7) line 156 - were your cells confluent then as integrins change at confluence
- 8) line 186 - reference your paper on this and maybe ref 61, since they initially developed this approach - it is always good to give credit
- 9) line 191 - 200 - these are really small differences to put so much weight on?
- 10) lines 294-301 - your discussion of TM cells as epithelial-like goes against everything in the outflow pathway literature and you should probably totally rework that - you will lose credibility with these kind of wrong analogies that don't tell us anything anyway
- 11) line 317, ref 46 is an unusual choice for that
- 12) line 320-322 - actually, SC cells are not thought to be particularly depleted in glaucoma and you don't have enough evidence to go off into that conjecture of SCE cell stimulation by your iPSCs. you could soften this a bit as you are way out on thin ice - your data lead there, but you are extending it a bit far. it is potentially important, so do talk about it, just be a bit less certain
- 13) line 332-334 - if the iPSCs were initiating ECM remodeling, they would probably hang around and do that instead of triggering TM cell proliferation - soften this fairly unlikely conjecture
- 14) line 388 5 to 8 what? clarify
- 15) line 391 - clarify "hiPSCs of U1" for non stem cell readers
- 16) line 452 - reference for cellPhoneDB v4 database
- 17) there are a couple places where you need to think about multitest corrections or at least tell us why not
- 18) figure 1 B & C - the contrast on the bar headers is so bad it is difficult to read the headers
- 19) it is a little difficult to see where the cluster in fig 1F comes from - could you explain this more clearly in the text and or in the caption - this is a very important point and how you chose it is a bit vague
- 20) the heatmap in fig 1E is a little strange - there is only a little similarity between individuals and the individual variation is huge - which is normally - but with this much variability, how did you pick what to use? please help the reader figure out what this figure tells you and how you got anything useful out of it
- 21) if you could explain parts of fig 3 a little better it would help readers
- 22) fig 4A the writing on the left side is small and with the colors very low contrast making it very hard to read - can you bold the words
- 23) fig 4C - is there color beyond blue there? are you just counting nuclei? explain what this shows more clearly
- 24) fig 5a the superscript + and - are too small to see - same for throughout this fig - maybe just put them as a full sized - and + behind the word - otherwise the figures are ok and size is appropriate
- 25) the contrast of the green and red lettering in the figures b, e, h, are very hard to read against black - can you lighten the letters up and still keep color? it is also a little hard to see the staining for the same reason

aside from these minor concerns - nice and important paper

Reviewer #3

(Remarks to the Author)

In this manuscript, the authors conducted single-cell RNA sequencing to characterize the molecular mechanisms underlying TM regeneration. They performed clustering analysis and identified a group of alpha6 integrin-positive (ITGA6+) iPSC-TM with a distinct transcription pattern that was not observed among primary TM cells. These ITGA6+ iPSC-TM stimulated cell proliferation of pTM and were capable of restoring aqueous humor outflow in glaucoma. Furthermore, ITGA6+ iPSC-TM repopulated the TM and reversed damage to the inner wall of Schlemm's canal. This study provided new insights into iPSC-derived cell therapy for treatments for POAG and could benefit the cell therapy area.

Nonetheless, the mechanisms of why ITGA6+ iPSC-TM functioned better than pTM are unclear and need more thorough investigation. The role of alpha6 integrin in regulating cell-cell interactions needs more studies. My other concern is that the transplanted cells are a cocktail of iPSC-TM but not ITGA6+ iPSC-TM only, which makes it hard to conclude the better treatment is due to the ITGA6+ iPSC-TM.

For the magnetic nanoparticle labeling, evidence of the cellular changes especially molecular changes upon treatment with magnetic nanoparticles is necessary. The distribution of the cells in the TM should be provided. Also, I recommend a control group with MNP-labeled transplanted pTM.

Version 1:

Reviewer comments:

Reviewer #1

(Remarks to the Author)

The authors have submitted a revised manuscript that incorporates additional experiments guided by reviewer's comments. However, the new data has reinforced my central concern with the paper: That ITGA6+ iPSC-derived cells are not "TM cells" but are instead a fibroblast-like less differentiated cell type arising during the preparation of iPSC-TM cells. This needs to be clarified throughout the manuscript, possibly by renaming the cells to reflect their actual identity (e.g. "iPSC-ITGA6+ cells" or similar). Furthermore, additional work is needed to determine why these cells, and not ITGA6- iPSC-TM cells, are capable of driving TM regeneration. Lacking that information, the present manuscript offers limited insights over author's previous work.

Additional comments:

1: The new findings using NEAT1 knockdown cells suggest that NEAT1 is an important regulator of iPSC-TM-ITGA6+ cell proliferation, consistent with the known role of NEAT1 in other cells. This raises the likelihood that the reduced effect of NEAT1 knockdown cells on proliferation of cultured pTM cells and in vivo TM cells after transplantation is caused by reduced proliferation (and thus numbers) of iPSC-TM-ITGA6+ cells and not a direct signaling role for NEAT1. Indeed, authors report that NEAT1 knockdown is associated with reduced numbers of iPSC-TM present 48 hours after transplantation. Additional experiments are needed to clarify the role of NEAT1 and determine if its role extends beyond regulating proliferation of iPSC-TM cells.

2: The author's goal of improving the trajectory analysis in figure 2 A in future studies (response to reviewers page 10) is admirable, but the present version should be removed from the paper. A more useful approach for showing the relationship between these cell types would be to use a PCA, as the authors have done in the past with very similar data: <https://doi.org/10.1038/s41598-020-59941-0>. As the cell types under comparison are the end products of parallel culture or differentiation protocols and no biological trajectory exists between them, the current analysis is inappropriate.

3: The new axis labeling in figure 3 C has greatly improved readability of that figure. However, as authors have retained the previous timeline labeling in figure 3 D, it has become confusing. Figure 3 D should be updated so that the X axis timeline is consistent with 3C as they describe the same data. In addition, the timepoint "before" should be labeled more descriptively to make clear which timepoint these values represent. Is it a true baseline (e.g. day 0) or immediately before cell transplantation (e.g. after glaucoma induction)?

4: In the newly-revised section on lines 191-193, authors state "...demonstrated a 44.4% decrease 24 days after injection which was significantly higher...(Fig 3 C and D; P = 0.0017)." In figure 3 D, the legend describes this statistical analysis as a 2-way ANOVA. Please provide additional information regarding this analysis. Was the 24-day P value derived from a posttest? Are the P values reported in the figure derived from the anova itself, or are these from a posttest? If these results are derived from posttesting, please make this clear and also provide the Df and F values for the underlying 2-way anova. These data should also be provided for all other panels using ANOVA in combination with posttesting.

5: Reviewer #2 raised an important point regarding statistical correction for multiple comparisons, which was not satisfactorily addressed in the revised manuscript. Regardless of the "planned" nature of the statistical comparisons reported, additional comparisons will alter the family-wise error rate and thus correction is required according to the number of comparisons performed.

6: On line 202, authors refer to figure 3 G instead of Figure 3 E.

7: Throughout the manuscript it is difficult to determine if the cells used in mouse experiments are produced with the conditioned media or recombinant cytokine approach. This should be clarified.

8: The association of -pTM5 with Schlemm's canal remains unconvincing. While ITGA8 has previously been identified in primary cells exhibiting SC-like characteristics, it is also expressed in other cell types of the angle. In addition, none of the DE genes shown in figure S3A are characteristic of Schlemm's canal, with RGS5, COL6A1 and COL6A2 especially being expressed by trabecular meshwork, smooth muscle and pericytes but not Schlemm's canal endothelium.

9: In figure 4D, while the higher magnification panels help the reader, please remove the label "TM". The arrows adequately indicate the TM, while the text unnecessarily obscures the region of interest.

Reviewer #2

(Remarks to the Author)

This reviewer is now satisfied that my concerns and suggestions have been addressed, and I now approve this revision.

Reviewer #3

(Remarks to the Author)

The authors have addressed my concerns in the revisions. Further investigations have been conducted to reveal the mechanisms of why ITGA6+ iPSC-TM functioned better than pTM. The authors confirmed the critical role of NEAT1 in iPSC-TM and incorporated the results to Figs. 5G-K, S8, S9, and S10A-B.

Version 3:

Reviewer comments:

Reviewer #1

(Remarks to the Author)

Thank you for the opportunity of reviewing this revised manuscript, which addresses the concerns I had with the previous submission and nicely characterizes the identity of Itga6+ iPSC cells.

Could authors discuss the following two further questions, raised by these interesting new findings:

1: Since this manuscript was first submitted in 2023, a study in eLife was published in 2025 that shares an author with the present paper (<https://doi.org/10.7554/eLife.103256.3.sa0>). This paper reported that mesenchymal stem cells were more effective at IOP-lowering than unsorted iPSC-TM cells prepared using the same conditioned media approach used here. As the new data indicates that ITGA6+ cells represent a less differentiated cell state than other iPSC-TMs, could these cells be behaving more similarly to the MSCs reported in the eLife paper?

2: Related to point 1, the previous study by Zhu et al (<https://doi.org/10.1073/pnas.1604153113>) and the eLife paper above both described tumor formation following treatment with iPSC-TM cells—a limitation that Zhu et al solved elegantly by negative selection of SSEA1+ pluripotent cells. If ITGA6+ cells are more highly pluripotent than other iPSC-TMs was tumor formation observed? How did it compare with the ITGA6-neg cells? Were these cells negatively selected for SSEA1+ cells before injection?

August 2023

We are very grateful for the reviewers' thoughtful comments. In response we have substantially revised our manuscript, provided additional data, and have incorporated all requested changes in this revision. Substantiative changes or additions to the manuscript have been marked in red font.

Our responses to specific concerns of the reviewers (in blue font) are listed below (in black font).

REVIEWER COMMENTS

Reviewer #1 (Remarks to the Author):

Zhu and colleagues have submitted an interesting work that extends their research on the therapeutic potential of iPSC-derived trabecular meshwork-like cells in glaucoma. In previous work, they have shown that cell therapy using IPSC-derived TM-like cells in mice can lower intraocular pressure and induce recellularization of the trabecular meshwork in MYOC-induced models of ocular hypertension. Here, they use single cell RNA sequencing to better characterize these cells before transplantation and identify potential molecules and pathways that might mediate the therapeutic effect. While the study is rigorous and interesting, novelty and impact is reduced by incomplete characterization of the cells, which makes it difficult to understand their true phenotype, how these IPSC-derived "TM cells" are related to in vivo trabecular meshwork cells, and how they mediate recovery of the aqueous outflow pathway.

Major questions:

1: The authors have identified a subset of cells present in their iPSC trabecular meshwork cell cultures that express ITGA6 and have the ability to trigger recellularization of the endogenous TM in mouse models of glaucoma. However, as described in detail throughout the manuscript, these cells are not present in primary TM cell cultures, and it is unclear if these cells are "iPSC-derived trabecular meshwork" or are a non-TM cell type that arose in the iPSC differentiation process. Additional characterization of these cells is needed to determine their true identity and if an analogous population occurs in vivo. Ascertaining the actual identity of this cell type is critical to understanding how they may function in relation to TM repair.

Response: Thank you for your suggestion. We greatly appreciate your input. Currently, we have an ongoing project in our lab aimed at gaining a deeper understanding of the identity of the ITGA6⁺ cluster and how cells within this cluster differentiated from iPSCs.

To investigate this, we have collected iPSC-TM cells at various stages of differentiation (0, 3, 5, 11, and 25 days) and performed single-cell RNA sequencing using 10x Genomics, similar to what we did in this study. As shown in Figure R1, we observed robust expression of ITGA6 and NEAT1, two distinct biomarkers of the ITGA6⁺ cluster, in some intermediate cells at 3, 5, 11, and 25 days. Interestingly, iPSCs and pTM cells did not express these two biomarkers simultaneously. Additionally, we noticed that ITGA6⁺ cells were exclusively found in the outer layer of Schlemm's canal *in vivo*, while the inner layer of SC and the TM, the most critical tissues controlling AH outflow, lacked ITGA6 expression (Fig. S9A). Consequently, we concluded that ITGA6⁺ iPSC-TM represents a distinct cell type only existing at the intermediate state of differentiation.

Figure R1. A UMAP plot showing normalized expressions of *ITGA6* (top panels) and *NEAT1* (bottom panels).

To further characterize *ITGA6*⁺ and *ITGA6*⁻ iPSC-TM, we compared them to annotated clusters of interest using module scores. Figure 4B illustrates that pTM was similar to TM1 and TM2, while *ITGA6*⁻ iPSC-TM showed greater similarity to JCT cells. In contrast, *ITGA6*⁺ iPSC-TM exhibited a distinct cell type identity, resembling fibroblasts and endothelial cells (Fig. 4B). Intriguingly, we found that *in situ* fibroblasts and SC endothelial cells were the major cell types expressing LAMC1, the specific ligand for alpha6 integrin (Fig. 4C). These findings led us to hypothesize that the transplanted *ITGA6*⁺ iPSC-TM might first anchor to regions abundant in fibroblasts and endothelial cells and subsequently influence the behaviors of neighboring cells. These valuable results have been included in the revised manuscript (lines 220 and 811) and have contributed to enriching our understanding of how *ITGA6*-positive cells function in tissue repair.

Revision to the manuscript:

Line 220:

Unlike pTM and *ITGA6*⁻ iPSC-TM, *ITGA6*⁺ iPSC-TM was characterized as a distinct cell type based on its similarity to fibroblasts and endothelial cells (Fig. 4B). Among the ligand-expressing cell types, we found fibroblasts and SCs in the outflow tissues are the primary sources of cell types that express LAMC1 (Fig. 4C).

Line 811:

Figure 4B

Figure 4. Regeneration of TM and SC by *ITGA6*⁺ iPSC-TM. (B) Heatmap showing the normalized module scores of the clusters identified in *ITGA6*⁺ iPSC-TM, *ITGA6*⁻ iPSC-TM, and

pTM.

2: Beyond their magnetic bead approach, have the authors considered altering their culture and/or differentiation conditions to further enrich for ITGA6⁺ cells or potentiate their formation? As these cells are supplying a stimulus that is not present in primary TM cells and appear to have a distinct transcriptional profile, optimizing culture conditions to generate maximally differentiated TM cells may not be the most effective method for obtaining outflow-targeted cell therapy candidates.

Response: The reviewer has raised an excellent point, and we wholeheartedly agree with their suggestion. The newly obtained scRNAseq data indeed provides us a valuable resource to develop a more effective approach for generating more functional ITGA6⁺ cells from iPSCs. Since transcriptional factors (TFs) have been demonstrated to be influential in controlling cell fate, we may select the critical TFs responsible for cell fate specification of ITGA6⁺ cells and apply them to differentiate iPSCs. This targeted approach has the potential to yield more reliable and functional ITGA6⁺ cells.

However, acknowledge that even if we successfully identify the critical TFs, several challenges remain for the clinical translation of iPSC-based therapy. One such challenge is the method of delivering these TFs into iPSCs. While virus-based approaches have been efficient in transducing TFs into cells, they may not be suitable for clinical applications due to safety concerns. Therefore, we must explore alternative delivery methods, such as electrotransformation or nanoparticle-based delivery, which could offer more feasible and safer options for clinical use. Furthermore, before these iPSC-derived cells can be utilized, it is crucial to rigorously test their ability to stimulate endogenous TM proliferation and effectively lower intraocular pressures (IOPs). Validating their functionality is essential to ensure their potential therapeutic benefits.

Despite these difficulties and challenges, we firmly believe that pursuing this practical approach for repairing the damaged TM is our responsibility and the next critical goal of our research. We extend our heartfelt gratitude to the reviewer for their valuable insights and encouragement. Their input has further motivated us to enhance our research efforts towards achieving our objective of developing a viable and effective treatment approach using iPSC-derived ITGA6⁺ cells.

3: Authors should perform knockdown of ITGA6 in iPSC-TM cells to determine if this is the functional effector or merely a biomarker of the cells mediating protective effect.

Response: Thanks for your suggestion. In response to your recommendation, we conducted a series of experiments to investigate the effects of siRNA ITGA6 treatment on iPSC-TM cells.

Initially, we applied siRNA ITGA6 to reduce the expression of ITGA6 in iPSC-TM. The results indicated a significant reduction in ITGA6 expression (siRNA control vs. siRNA ITGA6: 1.00 vs. 0.25; 75% knockdown; $P < 0.01$). This reduction in ITGA6 expression did not appear to have any significant influence on NEAT1 expression (siRNA control vs. siRNA ITGA6: 1.00 vs. 1.31; $P = 0.45$; Fig. S10A). Next, proceeded with a co-culture experiment, where we co-cultured HTM cells pre-labeled with CellTracker with 61.0% ITGA6⁺ iPSC-TM cells that were pre-treated with siRNA ITGA6.

The results, as depicted in Figure S10B, indicated that siRNA ITGA6 treatment impaired the

effect of iPSC-TM on HTM proliferation (siRNA control vs. siRNA ITGA6: 831.8 vs. 664.5 cells/dish; $P < 0.05$; Fig. S10B). However, when we compared these results with the fully diminished capacity of iPSC-TM to stimulate cell proliferation following 71.9% NEAT1 knockdown (siRNA control vs. siRNA NEAT1: 831.7 vs. 511.3 cells/dish; $P < 0.05$; Fig. 5L), we concluded that ITGA6 knockdown only partially damaged iPSC-TM function.

These significant findings have been carefully summarized and included in the revised manuscript (lines 271) and the revised supplementary materials (lines 160 and 290). We believe that these new results add further depth and understanding to our research on the functional role of ITGA6 in iPSC-TM cells. Once again, we sincerely appreciate your valuable suggestion, which has contributed to the enhancement of our study.

Revision to the manuscript:

Line 271:

Compared to the critical effect of NEAT1 (Fig. 5), ITGA6 knockdown only affected part of iPSC-TM's functions (Fig. S10 A-B).

Revision to the supplementary materials:

Line 160:

NEAT1/ITGA6 siRNA and scrambled siRNA oligonucleotides (Table S5) were prepared (NEAT1: Shanghai GenePharma Co., Ltd, China; ITGA6: sc-43130, SantaCruz Biotechnology, Inc., Texas, U.S.A) and transfected into iPSC-TM cells using LipoFiter™ transfection reagents (Invitrogen).

Line 290:

Fig. S10. Influence of ITGA6/NEAT1 on iPSC-TM's function. (A). RT-PCR results showing the expressions of *ITGA6* and *NEAT1* in iPSC-TM after treatment with scrambled siRNA (siRNA control) or siRNA *ITGA6*. Typical results from $n=3$ are shown. ** $P < 0.01$ by two-tailed t-test. (B). Left panel: representative images of HTM (red) after co-culturing for 48 hours with either HTM or iPSC-TM pre-treated with siRNA. Scale bars, 100 μm . Right panel: quantification of pre-stained HTM cellularity indicating an impaired capacity of iPSC-TM after *ITGA6* knockdown in stimulating cell proliferation. Typical results from $n=3$ are shown. * $P < 0.05$, **** $P < 0.0001$ by one-way ANOVA.

4: Additional characterization of the potential role(s) for *NEAT1* is required. Was expression of *ITGA6* or any of the DEGs observed in *ITGA6*⁺ vs - iPSC-TMs altered in *NEAT1* knockdown cells? Was proliferation of *NEAT1* knockdown *ITGA6*⁺ iPSC-TM cells reduced during their 48-hour culture period with HTMs?

Response: As per your suggestion, we proceeded with the down-regulation of NEAT1 expression in iPSC-TM using siRNA (Fig. 5H). To understand the underlying molecular changes resulting from NEAT1 knockdown, we evaluated the expression levels of ITGA6, Ki-67 (a representative biomarker for cell proliferation), and the paraspeckle-related structural proteins, NONO and SFPQ, via Western blot. As anticipated, NEAT1 knockdown led to a decrease in the expression levels of both NONO and SFPQ (Fig. 5I). However, what came as an unexpected finding was that the expression levels of ITGA6 and Ki-67 were also significantly reduced in iPSC-TM after NEAT1 knockdown (Fig. 5I).

Furthermore, we examined the correlation between NEAT1 and iPSC-TM cellularity locating in the TM region after transplantation, along with the correlation with endogenous cells at a proliferative state. Figure 5J showed a lower amount of iPSC-TM at the beginning of transplantation after NEAT1 knockdown (before vs. after: 39.1 vs. 32.3 iPSC-TM/section, $P = 0.12$, $n=57-85$ sections of 6 eyes in each group). Before NEAT1 knockdown, there was a strong correlation observed ($r_s = 0.22$ between STEM121⁺/Ki-67⁺ iPSC-TM and endogenous Ki-67⁺ cells, $P = 0.0015$; Fig. 5J-K and S8). However, after NEAT1 knockdown, this correlation was notably diminished ($r_s = 0.07$, $P = 0.051$; Fig. 5J-K and S8). Additionally, taking into account our previous result indicating the complete loss of iPSC-TM's capacity to stimulate neighboring cell proliferation after NEAT1 knockdown (Fig. 5L), we have concluded that the molecular mechanism underlying iPSC-based regeneration primarily relies on the robust NEAT1 expression in iPSC-TM. These findings have been added to the revised manuscript (lines 258, 261, and 828) and the revised supplementary materials (lines 264 and 276).

Revision to the manuscript:

Line 258:

In addition, ITGA6, NONO, and NEAT1 were expressed relatively lower in HTM cells than in iPSC-TM cells (Fig. 5E-G).

Line 261:

As shown in Fig. 5H-I, NEAT1 knockdown not only impaired the expressions of NONO and SFPQ, the major components in paraspeckles but also significantly decreased the expressions of ITGA6 and Ki-67, a typical proliferative biomarker. Although the transplanted iPSC-TM gradually diminished over time (Fig. 4), NEAT1 is associated with the amount of iPSC-TM at the beginning of transplantation (before vs. after NEAT1 knockdown: 39.1 vs. 32.3 iPSC-TM/section, $P = 0.12$, $n=57-85$ sections of 6 eyes in each group; Fig. 5J and S8) and iPSC-TM-induced endogenous cell proliferation in the TM (before vs. after NEAT1 knockdown: $r_s = 0.22$, $P = 0.0015$ vs. $r_s = 0.07$, $P = 0.051$; Fig. 5K, S8 and S9).

Line 828:

Figure 5. Participation of NEAT1 in induction of pTM proliferation. (G) RT-PCR results indicating a significantly lower expression of NEAT1 in pTM than in iPSC-TM. *** $P < 0.001$ by two-tailed t-test. Typical results from three technical repeats are shown. (H) RT-PCR results showing a lower expression of NEAT1 in iPSC-TM after transfection with siRNA NEAT1. iPSC-TM transfected with scrambled siRNA served as a control. ** $P < 0.01$ by two-tailed t-test. Typical results from three technical repeats are shown. (I) Left panel: Western blot results of ITGA6, Ki-67, NONO, SFPQ, and GAPDH (from top to bottom) in iPSC-TM transfected with siRNA NEAT1. iPSC-TM transfected with scrambled siRNA served as a control. Right panel: Quantification of band intensities analyzed using Image Lab software (Bio-Rad) showing significantly lower expression of ITGA6, Ki-67, or NONO in iPSC-TM after treatment with siRNA NEAT1. * $P < 0.05$, ** $P < 0.01$ by two-tailed t-test. Typical results from three technical repeats are shown. (J) A violin plot showing cellularity of STEM121-positive iPSC-TM in the TM and SC regions after cell transplantation for 48 hours, with the solid line representing the mean value and the dashed lines indicating associated 95% confidence limits. siRNA control: 39.1 iPSC-TM/section vs. 32.3 iPSC-TM/section, $P = 0.12$ by two-tailed t-test. (K) STEM121-negative/Ki-67-positive TM cells (indicating *in situ* proliferative cells) cross-plotted against STEM121-positive iPSC-TM pre-treated with siRNA control (left panel) or siRNA NEAT1 (right panel). r_s and P by the Person correlation coefficient are shown. Results showing a smaller r_s after treatment with siRNA NEAT1.

Revision to the supplementary materials:

Line 264:

Figure S8. Human iPSC-TM in mouse iridocorneal angle at the beginning of transplantation. (A) Immunohistochemical staining of STEM121 (green) and LAMC1 (red) in the iridocorneal tissues of C57BL/6 mice after receiving 50,000 human iPSC-TM pre-treated with siRNA control or siRNA NEAT1 for 48 hours. (B) Immunohistochemical staining of STEM121 (green) and Ki-67 (red) in the iridocorneal tissues of the above mice. Nuclei were labeled with DAPI (blue). Experiments were technically repeated three times using 57-85 sections of 6 eyes in each group. Scale bars, 100 μm . TM: trabecular meshwork; SC: Schlemm's canal.

Line 276:

Figure S9. Mouse iPSC-TM in mouse iridocorneal angle at the beginning of transplantation.

(B). After transplantation for 24 or 48 hours, dsRed-negative/Ki-67-positive TM cells cross-plotted against dsRed-positive iPSC-TM (top panels) or dsRed-positive/Ki-67-positive cells (bottom panels). r_s and P by the Person correlation coefficient are shown.

5: It is unclear why authors focused on NEAT1, a cell-autonomous mediator of cell proliferation, when it appears that iPSC-TM cells themselves are not repopulating the outflow tissues but rather encouraging endogenous cells to do so. Do iPSC-TM cells proliferate once embedded within the TM?

Response: It is reasonable to consider the implications of iPSC-TM's robust expression of NEAT1 and its potential capacity for cell proliferation, which could potentially enable it to replace the damaged TM after transplantation. However, our initial study (PMID: 27274060) revealed an interesting and unexpected finding that long-term regeneration was primarily attributed to endogenous cell proliferation rather than replacement by the transplanted cells. Consistent with this finding, we also found that iPSC-TM gradually diminished over the course of transplantation (2 months; Fig. 4H and J). However, as you pointed out, we did observe that iPSC-TM demonstrated a proliferation capacity at the beginning of transplantation, which is highly associated with the presence of endogenous proliferative cells (Fig. 5J-K, S8 and S9).

Upon combining these findings, we have come to the conclusion that although the niche of conventional outflow tissue may not be suitable for the long-term survival of iPSC-TM, the transient stimulation of iPSC-TM at the beginning of transplantation plays a vital role in facilitating the long-term regeneration of the TM. These findings have been added to the revised manuscript (line 264).

Revision to the manuscript:

Line 264:

Although the transplanted iPSC-TM gradually diminished over time (Fig. 4), NEAT1 is associated with the amount of iPSC-TM at the beginning of transplantation (before vs. after NEAT1 knockdown: 39.1 vs. 32.3 iPSC-TM/section, $P = 0.12$, $n=57-85$ sections of 6 eyes in each group; Fig. 5J and S8) and iPSC-TM-induced endogenous cell proliferation in the TM (before vs. after NEAT1 knockdown: $r_s = 0.22$, $P = 0.0015$ vs. $r_s = 0.07$, $P = 0.051$; Fig. 5K, S8 and S9).

6: Authors have previously described a key role for Cx43 expression in iPSC-TM-mediated TM regeneration. Was Cx43 expression different in ITGA6+ and – cell populations? Was it altered in NEAT1 knockdown cells?

Response: As per your suggestion, we first collected three samples, ITGA6⁺ iPSC-TM (61.0%), ITGA6⁺ iPSC-TM pre-treated with siRNA ITGA6, and ITGA6⁻ iPSC-TM and subsequently determined the expressions of ITGA6 and Cx43 using Western blot. Our results showed that in comparison to ITGA6⁺ iPSC-TM, both siRNA-treated iPSC-TM and ITGA6⁻ iPSC-TM exhibited a relatively lower expression of ITGA6 (Fig. S10C). Quantification analysis further confirmed that Cx43 expression was significantly lower in ITGA6⁻ iPSC-TM than in ITGA6⁺ iPSC-TM ($P < 0.001$; Fig. S10C). In addition, ITGA6 knockdown did not have any influence on Cx43 expression (Fig. S10C). In contrast, NEAT1 knockdown significantly reduced Cx43 expression ($P < 0.05$; Fig.

S10D), further implicating the crucial role of NEAT1 in iPSC-TM. These findings have been added to the revised manuscript (lines 354 and 373) and the revised supplementary materials (line 290).

Revision to the manuscript:

Line 354:

Our results clearly showed that NEAT1 knockdown decreases the expressions of ITGA6, Ki-67, NONO, SFPQ, and Connexin 43 in iPSC-TM, impairs the capacity of iPSC-TM in anchoring the LAMC1-positive regions of conventional outflow tissue, and weakens iPSC-TM in stimulating endogenous cell proliferation.

Line 373:

It also remains insufficiently explained how NEAT1 expression in iPSC-TM influences pTM cell proliferation and whether this effect is tied to other genes.

Revision to the supplementary materials:

Line 290:

Fig. S10. Influence of ITGA6/NEAT1 on iPSC-TM's function. (C). Left panel: Western blot detection of Connexin 43, ITGA6, and GAPDH (from top to bottom) in ITGA6⁺ iPSC-TM, ITGA6⁺ iPSC-TM transfected with siRNA ITGA6, and ITGA6⁻ iPSC-TM. Right panel: Quantification of band intensities showing a significantly lower expression of Connexin 43 in ITGA6⁻ iPSC-TM than in ITGA6⁺ iPSC-TM. (D). Left panel: Western blot detection of Connexin 43 and GAPDH in iPSC-TM transfected with siRNA NEAT1. iPSC-TM transfected with scrambled siRNA was used as a control. Right panel: Quantification of band intensities showing a significantly lower expression of Connexin 43 in iPSC-TM after treatment with siRNA NEAT1. *** $P < 0.001$, * $P < 0.05$ by two-tailed t-test. Typical results from $n=3$ technical repeats are shown.

7: Authors state that only a limited number of transplanted cells remain after 45 days (figure 4 F-G). Is this due to transplanted cells dying, or is implantation efficiency low initially? Data from a shorter timepoint should be included.

Response: As suggested, we quantified the cellularities of human/mouse iPSC-TM in the mouse iridocorneal angle after transplantation for 24 or 48 hours (Fig. 5: 48 hours: 39.1 human iPSC-TM cells/section, $n=85$; Fig. S9: 24 hours: 6.3 mouse iPSC-TM cells/section, $n=12$; 48 hours: 21.5 mouse iPSC-TM cells/section, $n=14$). These data indicated that the diminished iPSC-TM in the iridocorneal region over time is not due to the low implantation efficiency. These findings have been added to the revised manuscript (line 264).

Revision to the manuscript:

Line 264:

Although the transplanted iPSC-TM gradually diminished over time (Fig. 4), NEAT1 is associated with the amount of iPSC-TM at the beginning of transplantation (before vs. after NEAT1 knockdown: 39.1 vs. 32.3 iPSC-TM/section, $P = 0.12$, $n=57-85$ sections of 6 eyes in each group; Fig. 5J and S8) and iPSC-TM-induced endogenous cell proliferation in the TM (before vs. after NEAT1 knockdown: $r_s = 0.22$, $P = 0.0015$ vs. $r_s = 0.07$, $P = 0.051$; Fig. 5K, S8 and S9)

Minor questions

1: In figure 1 A-C, it is unclear if the figures exist in the same UMAP space.

Response: These panels were not shown in the same UMAP space. It has been describe in the legend of Figure 1 (line 747).

2: Based on the UMAP projection shown in Figure S2 B, it appears that ITGA6+ cells are located in cell clusters CM1 and RC2. Figure 1E would be clearer if this information was available without referring to the supplemental information.

Response: As suggested, data showing the similarities of ITGA6+ iPSC-TM and the clusters of interest has been added to the revised manuscript (Figure 4B).

3: The trajectory analysis presented in figure 2 A is not informative and should be removed. A more useful trajectory analysis might help to position the IPSC-TM cells within the context of the previously published datasets of outflow tissues.

Response: Thank you for your suggestion. We acknowledge that due to the limited overlap between iPSC-TM and pTM, performing a comprehensive trajectory analysis solely with these samples. However, we are committed to addressing this limitation and improving the trajectory analysis by incorporating our latest data. It is one goal for our subsequent study.

4: On lines 142, it is unclear how cluster pTM5 was identified as Schlemm's canal. None of the marker genes identified for this cluster in figure S3 are unique to endothelial cells or Schlemm's canal.

Response: pTM5 has been characterized as a cluster with a relatively robust expression of *ITGA8* (Fig. S3). We have previously demonstrated that *ITGA8*-positive cells exhibit several characteristics similar to Schlemm's canal endothelial cells. These features include the robust expressions of SC biomarkers, a low capacity of glucocorticoid-induced myocilin expression, and a strong ability of cell contractility (PMID: 33961857). These previous findings have been described in the revised manuscript to strengthen our conclusion (lines 130 and 138).

Revision to the manuscript:

Line 130:

Compared to *ITGA8*-negative cells, *ITGA8*-positive cells exhibit several characteristics similar to Schlemm's canal endothelial cells, such as the robust expressions of SC biomarkers, a low capacity of glucocorticoid-induced myocilin expression, and a strong ability of cell contractility.

Line 138:

Together, these data demonstrate the heterogeneity of cultured human TM cells and their relationship to TM cells, fibroblasts, and Schlemm's canal endothelial cells *in situ*.

5: In figure 4 D and E, all statistical comparisons described in the manuscript should be included on the plots.

Response: As suggested, P values have been added in Figure 4 (line 806).

6: Figure 3 C would be clearer if the X axis timeline was continuous or more clearly indicated the full time elapsed between IOP measurement at day 25 and post transplantation day 10. Were cells transplanted on day 25?

Response: The transplantation occurred on day 25, which has been re-marked in Figure 3C. As suggested, a continuous timeline was shown, and the figure legend has been updated in the revised manuscript (line 784). Thanks.

Revision to the manuscript:

Line 784:

Figure 3. Role of ITGA6⁺ iPSC-TM in restoring TM function in glaucoma mice. (C) IOPs in different groups of mice: naïve mice (light gray; n=10), naïve mice following intracameral injection of Ad5-EGFP (dark gray; WT-EGFP; n=8), glaucoma mice receiving injections of Ad5-MYOC^{Y437H}-EGFP and PBS treatment (orange; n=8); glaucoma mice receiving injections of Ad5-MYOC^{Y437H}-EGFP and iPSC-TM containing 22.8% ITGA6⁺ cells (aquamarine; n=8), and glaucoma mice receiving injections of Ad5-MYOC^{Y437H}-EGFP and iPSC-TM containing 40.6% ITGA6⁺ cells (blue; n=8). The blue-shaded background indicates the period before iPSC-TM transplantation.

7: In figure 4 C, labels obscure the most important section of the images. Arrows should be used instead. In addition, at the provided magnification it is difficult to appreciate the changes in cellularity. While authors can easily differentiate cell types for counting nuclei, in the format of a small figure panel specific staining of the cell types quantified would make this figure easier to follow.

Response: As per your recommendation, we have made the necessary updates to Figure 4D. We have now added arrows to clearly indicate the TM and SC regions in the iridocorneal angle. The distinct structures of the conventional outflow tissue, as shown in the schematic illustration of Figure 4D, will aid in easily distinguishing cells located on the TM beams or beneath the SC inner wall. Additionally, in response to your concern, we have included images at a higher magnification in Figure 4D.

Revision to the manuscript:

Line 806:

Figure 4. Regeneration of TM and SC by ITGA6⁺ iPSC-TM. (D) Immunohistochemical staining of nuclei in the iridocorneal regions and a schematic illustration showing the anatomical structure of the TM and SC. Tissues at a higher magnification are shown at the bottom of each image.

Reviewer #2 (Remarks to the Author):

Response: Thank you for your thoughtful and encouraging comments. Changes have been updated in the revised manuscript.

1) this is a strong and generally well-developed study of an important topic that will advance the field.

2) The authors have identified a subset of differentiated iPSCs that are more effective when transplanted into glaucomatous anterior segments in restoring intraocular pressure control.

3) In general, the methods and data analysis are sound and support the conclusion.

4) line 51-54 - wording is confusing, elevated IOP is the primary addressable risk factor instead of making age, which is not addressable anyway, seem more important

Response: Thanks. It has been revised (line 49).

Revision to the manuscript:

Lines 49:

Glaucoma is the leading cause of irreversible blindness worldwide (1). Elevated intraocular pressure (IOP) due to the imbalance of aqueous humor (AH) production and outflow is the primary risk factor for developing glaucoma (2).

5) line 90-91 is a little overstated

Response: The statement has been modified (line 85).

Revision to the manuscript:

Lines 85:

These findings facilitate a better understanding of both the TM's and SC's physiological function in controlling AH drainage.

6) a number of the figure captions are too small to read, especially the superscripts

Response: The captions in Figures 4 and 5 have been modified.

7)line 156 - were your cell confluent then as integrins change at confluence

Response: In response to the query raised by Reviewer #1 regarding the function of ITGA6 in iPSC-TM, we conducted experiments involving siRNA knockdown of ITGA6 in iPSC-TM cells. We carefully observed the cell morphology and confluence following the treatments. As shown in Figure R2, changes in ITGA6 expression (Fig. S10) resulting from the siRNA treatment had no significant impact on cell confluence or cell morphology in the culture.

Figure R2. Representative images of iPSC-TM transfected with siRNA control or siRNA ITGA6. Scale bars, 100 μ m.

8) line 186 - reference your paper on this and maybe ref 61, since they initially developed this approach - it is always good to give credit

Response: Thanks. The study from Prof. Borrás's group has been cited here (lines 183 and 650).

Revision to the manuscript:

Line 183:

In our first experiment we induced IOP elevation through injections of Ad5-MYOC^{Y437H}-EGFP (34, 35).

Line 650:

34. T. Borrás, The effects of myocilin expression on functionally relevant trabecular meshwork genes: a mini-review. *J Ocul Pharmacol Ther* 30, 202-212 (2014).

35. S. Sui et al., iPSC-Derived Trabecular Meshwork Cells Stimulate Endogenous TM Cell Division Through Gap Junction in a Mouse Model of Glaucoma. *Invest Ophthalmol Vis Sci* 62, 28 (2021).

9)line 191 - 200 - these are really small differences to put so much weight on?

Response: This paragraph has been rewritten (line 188).

Revision to the manuscript:

Line 188:

Consistent with previous reports, injecting 8×10^7 PFU Ad5-MYOC^{Y437H}-EGFP (n=8) significantly increased IOP when compared to eyes that received Ad5-EGFP (Day 25: 17.8 mmHg vs. 14.1 mmHg, $P = 0.0001$, n = 8) or PBS (Day 25: 17.8 mmHg vs. 14.6 mmHg, $P = 0.0002$, n = 10). Eyes that received preparations containing 40.6% ITGA6⁺ iPSC-TM demonstrated a 44.4% decrease in IOP at day 24 after transplantation, which was significantly higher than those received preparations containing 22.8% ITGA6⁺ iPSC-TM (Fig. 3C and D; $P = 0.0017$). After day 24, fractions containing 40.6% and 22.8% ITGA6⁺ iPSC-TM led to a similar decrease in IOP, which continued until the end of the experiment. Importantly, no significant reduction in IOP was observed following the injection of PBS.

10) lines 294-301 - your discussion of tm cells as epithelial-like goes against everything in the

outflow pathway literature and you should probably totally rework that - you will lose credibility with these kind of wrong analogies that don't tell us anything anyway

Response: Thanks. As suggested, we have rewritten this paragraph (line 309).

Revision to the manuscript:

Line 309:

Consistent with the reported clusters in conventional outflow tissue, we also found that the inner layers of the TM mainly consist of pTM1 (TM1-like cells) and pTM2 (fibroblast-like cells), while the outer regions are primarily covered with pTM4 (JCT-like cells) and pTM5 (endothelial cells). Of note, The distinct biomarkers for these clusters have been identified, including *FABP4*, *DCN*, *PDGFRA*, and *BMP5* for TM1-like cells, *ANGPTL7*, *CHI3L1*, *RSPO4*, and *FMOD* for JCT-like cells, and *POSTN*, *TFF3*, *MMRN1*, and *FLT4* for L-Endo/SC-like cells (20, 21). These findings not only facilitate us to purify the clusters of interest for further investigation but also enhance our understanding of the distinct functions of different regions in draining AH. For example, previous studies have revealed that cells located on the inner uveal and corneoscleral meshwork possess a unique capacity to scavenge debris in the AH (36). Based on the positive role of PDGF in phagocytic activity (37), the robust expression of *PDGFRA* in TM1-like cells may explain this phenomenon. Moreover, the exclusive expression of *POSTN*, *MMRN1*, and *FLT4* relevant to vasculogenesis and lymphangiogenesis were only expressed in Schlemm's canal, further supporting the notion that SC functions as a blood/lymphatic vessel-like structure (38). Beyond the reported evidence, we found that *ANGPTL7*, a negative regulator for vasculature development, was only expressed in the JCT, suggesting it may be crucial for maintaining the avascular feature of the TM.

11) line 317, ref 46 is an unusual choice for that

Response: It has been replaced by a new reference (line 670).

Revision to the manuscript:

Lines 670:

46. J. O'Callaghan et al., Matrix metalloproteinase-3 (MMP-3)-mediated gene therapy for glaucoma. *Science advances* 9, eadf6537 (2023).

12) line 320-322 - actually, SC cells are not thought to be particularly depleted in glaucoma and you don't have enough evidence to go off into that conjecture of SCE cell stimulation by your iPSCs. you could soften this a bit as you are way out on thin ice - your data lead there, but you are extending it a bit far. it is potentially important, so do talk about it, just be a bit less certain

Response: Thank you for your suggestion. It has been revised (line 332).

Revision to the manuscript:

Line 332:

As such, the significant improvements observed in the eye's ability to regulate IOP are likely attributed to the functional restoration of both TM and SC (35).

13) line 332-334 - if the iPSCs were initiating ECM remodeling, they would probably hang around and do that instead of triggering TM cell proliferation - soften this fairly unlikely conjecture

Response: It has been corrected.

Revision to the manuscript:

Line 343:

It is also possible that ITGA6⁺ binding to ECM in our glaucoma models generates signals that evoke iPSC-TM to stimulate cell proliferation.

14) line 388 5 to 8 what? clarify

Response: It has been clarified in the revised manuscript (line 402).

Revision to the manuscript:

Line 402:

Notably, all donors used in this study were listed in Table S1.

15) line 391 - clarify "hiPSCs of U1" for non stem cell readers

Response: Three donors donating their renal urethra epithelial cells for reprogramming were named U1, U2, and U3, respectively. Thus “hiPSCs of U1” refers to human iPSCs reprogrammed from the renal urethra epithelial cells of donor U1. It has been clarified in the revised manuscript (line 399).

Revision to the manuscript:

Line 399:

The conditioned media (CM) obtained from the cell cultures were pooled. After sterilization by filtering through a mixed cellulose ester membrane with 0.2 μm pores (Millipore, Danvers, MA), the pooled CM were utilized for the differentiation of both human iPSCs (hiPSCs) and mouse iPSCs. Notably, all donors involved in this study were listed in Table S1. hiPSCs of donor U1 were subjected to one-month differentiation, and the resulting cells were designated iPSC-TM CM and collected for scRNA-seq analysis.

16) line 452 - reference for cellPhoneDB v4 database

Response: It has been added (line 214).

Revision to the manuscript:

Line 214:

We first evaluated the ligand-receptor interactions between iPSC-TM cells and the aqueous outflow tissues using CellPhoneDB (<https://www.cellphonedb.org/>).

17) there are a couple places where you need to think about multitesting corrections or at least tell us why not

Response: Thanks. We greatly appreciate your input. In this study, we performed planned

comparisons between two groups of interest, such as 22.8% ITGA6⁺ iPSC-TM vs. 40.6% ITGA6⁺ iPSC-TM, PBS vs. 22.8% ITGA6⁺ iPSC-TM, or PBS vs. 40.6% ITGA6⁺ by two-way ANOVA in Figure 4D. Based on the principle of multitesting correction (the Bonferroni correction), the significant level is still 0.05. The method for the statistical analysis has been updated in the revised manuscript (line 538).

Revision to the manuscript:

Line 538:

Of note, planned comparisons between two groups of interests were performed in this study, and p values < 0.05 were considered to be statistically significant.

18) figure 1 B & C - the contrast on the bar headers is so bad it is difficult to read the headers

Response: It has been updated (line 742).

19) it is a little difficult to see where the cluster in fig 1F comes from - could you explain this more clearly in the text and or in the caption - this is a very important point and how you chose it is a bit vague

Response: As shown in Figures 1F and S2B, cluster 1 exclusively presents in iPSC-TM RC, while cluster 3 is only detectable in iPSC-TM CM. Given that both iPSC-TM CM and iPSC-TM RC demonstrated the ability to stimulate cell proliferation, we excluded these distinct clusters from the following analysis. Instead, we focused on the remaining clusters, as indicated within the red frame in Figure 1F. Upon analyzing these specific clusters, we made an intriguing observation: the orange subclusters were exclusively present in iPSC-TM CM and RC, but absent in pTM (Fig. 1H). As suggested, we have now included this finding in the revised manuscript (line 144).

Revision to the manuscript:

Line 144:

We first conducted an integrated analysis of iPSC-TM and pTM cells with low resolution (Fig. 1F-G, Fig. S2B) and excluded the distinct cluster compositions in either iPSC-TM CM or RC (cluster 1 in iPSC-TM RC; cluster 3 in iPSC-TM CM). After focusing on the shared cluster in the red frame of Figure 1F, we observed a great difference between iPSC-TM (CM and RC) and pTM at high-resolution settings.

20) the heatmap in fig 1E is a little strange - there is only a little similarity between individuals and the individual variation is huge - which is normally - but with this much variability, how did you pick what to use? please help the reader figure out what this figure tells you and how you got anything useful out of it

Response: In our study, we utilized a 2D static *in vitro* culture system without mechanical signals, which indeed differs from the complex *in vivo* niche present in conventional outflow tissue. As a result, it is reasonable to expect differences between the behavior of our TM cells cultured *in vitro* and the *in situ* cells of the conventional outflow tissue. This phenomenon is well-documented in the literature, where TM cell behavior can be significantly influenced by its culture environment (PMID: 18789927). While most investigators tend to favor primary cultures of isolated TM cells

as an *in vitro* TM model for examining TM physiology, our results indicated that at least during the early stages of the culture, our *in vitro* TM model demonstrates feasibility in mirroring certain aspects of *in situ* TM. This similarity is evident through the comparison of TM1 and JCT, providing one advantage of our approach. Additionally, another notable advantage is the revelation of the utterly different transcriptome of iPSC-TM (both CM and RC) compared to pTM. This differential gene expression explains why iPSC-TM, rather than pTM, possesses the unique function of stimulating cell proliferation. It has been clarified in the revised manuscript (lines 120 and 294).

Revision to the manuscript:

Line 120:

We observed that pTM clusters exhibited a higher similarity to TM1, TM2, and JCT cells in human tissues compared to iPSC-TM cells, whereas iPCS-TM CM and iPCS-TM RC showed fewer transcriptional similarities (Fig. 1E). Of note, our samples still displayed distinctions compared to the *in situ* tissues.

Line 294:

In addition, there were other detectable differences between pTM and *in situ* cells, yet these differences are reasonable and expected, given that TM cell behaviors can be significantly influenced by its culture environment³⁶. Despite these dissimilarities, our results still indicate that cells isolated from the TM and grown *in vitro* retain several TM subtypes, at least during the early stages of the culture.

21) if you could explain parts of fig 3 a little better it would help readers

Response: The figure legend has been updated (line 790).

Revision to the manuscript:

Line 790:

Figure 3. Role of ITGA6⁺ iPSC-TM in restoring TM function in glaucoma mice. (D) The percentage of IOP decrease in each group of mice. Eyes that received 40.6% ITGA6⁺ iPSC-TM exhibited a more significant decrease in IOP than those that received 22.8% ITGA6⁺ iPSC-TM ($P = 0.0014$). $**P < 0.01$, $****P < 0.0001$ by two-way ANOVA. (E-F) Changes in IOP (dIOP, left, $*P < 0.05$ by two-way ANOVA) and outflow facility (C_r , right, $*P < 0.05$ by two-tailed t-test) in Tg-MYOC^{Y437H} receiving iPSC-TM CM at ITGA6⁺ ratios of either 3.9% (orange; n=5) or 49.7% (green; n=8). Significant reductions in IOP and noticeable increases in C_r were observed in Tg-MYOC^{Y437H} mice receiving 49.7% ITGA6⁺ cells compared to those receiving 3.9% ITGA6⁺ cells ($P = 0.012$). No significant changes in either IOP after pTM treatment. Each data point represents an individual dIOP or C_r . Scale bars, 100 μ m.

22) fig 4A the writing on the left side is small and with the colors very low contrast making it very hard to read - can you bold the words

Response: It has been changed (line 806).

23) fig 4C - is there color beyond blue there? are you just counting nuclei? explain what this

shows more clearly

Response: We have made the necessary updates to Figure 4D. We have now added arrows to clearly indicate the TM and SC regions in the iridocorneal angle. The distinct structures of the conventional outflow tissue, as shown in the schematic illustration of Figure 4D, will aid in easily distinguishing cells located on the TM beams or beneath the SC inner wall. Additionally, in response to your concern, we have included images at a higher magnification in Figure 4D.

Revision to the manuscript:

Line 806:

Figure 4. Regeneration of TM and SC by ITGA6⁺ iPSC-TM. (D) Immunohistochemical staining of nuclei in the iridocorneal regions and a schematic illustration showing the anatomical structure of the TM and SC. Tissues at a higher magnification are shown at the bottom of each image.

24) *fig 5a the superscript + and - are too small to see - same for throughout this fig - maybe just put them as a full sized - and + behind the word - otherwise the figures are ok and size is appropriate*

Response: It has been changed (line 828).

25) *the contrast of the green and red lettering in the figures b, e, h, are very hard to read against black - can you lighten the letters up and still keep color? it is also a little hard to see the staining for the same reason*

Response: We have modified the contrast of the letters as suggested, hopefully visualizing better.

aside from these minor concerns - nice and important paper

Reviewer #3 (Remarks to the Author):

In this manuscript, the authors conducted single-cell RNA sequencing to characterize the molecular mechanisms underlying TM regeneration. They performed clustering analysis and identified a group of alpha6 integrin-positive (ITGA6+) iPSC-TM with a distinct transcription pattern that was not observed among primary TM cells. These ITGA6+ iPSC-TM stimulated cell proliferation of pTM and were capable of restoring aqueous humor outflow in glaucoma. Furthermore, ITGA6+ iPSC-TM repopulated the TM and reversed damage to the inner wall of Schlemm's canal. This study provided new insights into iPSC-derived cell therapy for treatments for POAG and could benefit the cell therapy area.

Nonetheless, the mechanisms of why ITGA6+ iPSC-TM functioned better than pTM are unclear and need more thorough investigation.

Response: Thanks for your suggestions. DEGs in ITGA6⁺ iPSC-TM (Fig. 5A) facilitated us to narrow down the possibilities of the molecular mechanism underlying iPSC-based regeneration. We showed different amounts of paraspeckles in pTM and iPSC-TM, implicating that NEAT1 may function more importantly (Fig. 5B-F). In response to your suggestion, we have conducted further investigations to confirm the critical role of NEAT1 in iPSC-TM (Figs. 5G-K, S8, S9, and S10A-B).

We first detected the expression levels of NEAT1 in pTM and iPSC-TM by RT-PCR. The results in Fig. 5G revealed a significantly lower expression of NEAT1 in pTM than in iPSC-TM, consistent with the observation of the difference in paraspeckle abundance between these cells (Fig. 5C and 5F). Next, we observed that NEAT1 knockdown not only led to lower expressions of NONO and SFPQ but also significantly reduced the expressions of ITGA6 and Ki-67, a representative biomarker of cell proliferation, in iPSC-TM (Fig. 5H-I). Furthermore, the correlation between iPSC-TM cells and endogenous cell proliferation were impaired after NEAT1 knockdown (Fig. 5J-K, S8 and S9). Moreover, in line with our previous findings, we observed that the stimulatory effect of iPSC-TM on cell proliferation nearly disappeared with a 71.9% downregulation of NEAT1, bringing it back to a similar level as pTM (Fig. 5L). Based on these comprehensive results, we can conclude that the higher expression of NEAT1 in iPSC-TM compared to pTM cells is a primary reason for the enhanced role of iPSC-TM in tissue regeneration. These results have been added in the revised manuscript (lines 258, 261 and 828) and the revised supplementary materials (line 264 and 276).

Revision to the manuscript:

Line 258:

In addition, ITGA6, NONO, and NEAT1 were expressed relatively lower in HTM cells than in iPSC-TM cells (Fig. 5E-G).

Line 261:

As shown in Fig. 5H-I, NEAT1 knockdown not only impaired the expressions of NONO and SFPQ, the major components in paraspeckles but also significantly decreased the expressions of

ITGA6 and Ki-67, a typical proliferative biomarker. Although the transplanted iPSC-TM gradually diminished over time (Fig. 4), NEAT1 is associated with the amount of iPSC-TM at the beginning of transplantation (before vs. after NEAT1 knockdown: 39.1 vs. 32.3 iPSC-TM/section, $P = 0.12$, $n=57-85$ sections of 6 eyes in each group; Fig. 5J and S8) and iPSC-TM-induced endogenous cell proliferation in the TM (before vs. after NEAT1 knockdown: $r_s = 0.22$, $P = 0.0015$ vs. $r_s = 0.07$, $P = 0.051$; Fig. 5K, S8 and S9).

Line 828:

Figure 5. Participation of NEAT1 in induction of pTM proliferation. (G) RT-PCR results indicating a significantly lower expression of NEAT1 in pTM than in iPSC-TM. *** $P < 0.001$ by two-tailed t-test. Typical results from three technical repeats are shown. (H) RT-PCR results showing a lower expression of NEAT1 in iPSC-TM after transfection with siRNA NEAT1. iPSC-TM transfected with scrambled siRNA served as a control. ** $P < 0.01$ by two-tailed t-test. Typical results from three technical repeats are shown. (I) Left panel: Western blot results of ITGA6, Ki-67, NONO, SFPQ, and GAPDH (from top to bottom) in iPSC-TM transfected with siRNA NEAT1. iPSC-TM transfected with scrambled siRNA served as a control. Right panel: Quantification of band intensities analyzed using Image Lab software (Bio-Rad) showing significantly lower expression of ITGA6, Ki-67, or NONO in iPSC-TM after treatment with siRNA NEAT1. * $P < 0.05$, ** $P < 0.01$ by two-tailed t-test. Typical results from three technical repeats are shown. (J) A violin plot showing cellularity of STEM121-positive iPSC-TM in the TM and SC regions after cell transplantation for 48 hours, with the solid line representing the mean value and the dashed lines indicating associated 95% confidence limits. siRNA control: 39.1 iPSC-TM/section vs. 32.3 iPSC-TM/section, $P = 0.12$ by two-tailed t-test. (K) STEM121-negative/Ki-67-positive TM cells (indicating *in situ* proliferative cells) cross-plotted against STEM121-positive iPSC-TM pre-treated with siRNA control (left panel) or siRNA NEAT1 (right panel). r_s and P by the Person correlation coefficient are shown. Results showing a smaller r_s after treatment with siRNA NEAT1.

Revision to the supplementary materials:

Line 264:

Figure S8. Human iPSC-TM in mouse iridocorneal angle at the beginning of transplantation. (A) Immunohistochemical staining of STEM121 (green) and LAMC1 (red) in the iridocorneal tissues of C57BL/6 mice after receiving 50,000 human iPSC-TM pre-treated with siRNA control or siRNA NEAT1 for 48 hours. (B) Immunohistochemical staining of STEM121 (green) and Ki-67 (red) in the iridocorneal tissues of the above mice. Nuclei were labeled with DAPI (blue). Experiments were technically repeated three times using 57-85 sections of 6 eyes in each group. Scale bars, 100 μm. TM: trabecular meshwork; SC: Schlemm's canal.

Line 276:

Figure S9. Mouse iPSC-TM in mouse iridocorneal angle at the beginning of transplantation.

(B). After transplantation for 24 or 48 hours, dsRed-negative/Ki-67-positive TM cells cross-plotted against dsRed-positive iPSC-TM (top panels) or dsRed-positive/Ki-67-positive cells (bottom panels). r_s and P by the Person correlation coefficient are shown.

The role of alpha6 integrin in regulating cell-cell interactions needs more studies.

Response: In our previous study, we extensively investigated the critical role of Connexin 43 (Cx43) in transducing proliferative signals from iPSC-TM to primary TM cells. Building upon this knowledge, we focused on Cx43 as the target and sought to determine whether its functionality could be affected by alpha6 integrin. To achieve this, we collected three samples, 61.0% ITGA6⁺ iPSC-TM cells transfected with or without siRNA ITGA6 and ITGA6⁻ iPSC-TM. Subsequently, we analyzed Cx43 expression through the Western blotting analysis (Fig. S10C). The quantification analysis revealed a significantly lower expression of Cx43 in ITGA6⁻ iPSC-TM compared to ITGA6⁺ iPSC-TM ($P < 0.001$; Fig. S10C). However, ITGA6 knockdown in iPSC-TM did not influence Cx43 expression ($P = 0.78$; Fig. S10C). In addition, we found that NEAT1 knockdown in iPSC-TM significantly lowered Cx43 expression ($P < 0.05$; Fig. S10D). These findings further indicated that NEAT1 is a critical factor responsible for cell-cell interactions between iPSC-TM and TM, rather than alpha6 integrin. We have modified the descriptions in the revised manuscript (line 354) and the revised supplementary materials (line 290).

Revision to the manuscript:

Line 354:

Our results clearly showed that NEAT1 knockdown decreases the expressions of ITGA6, Ki-67, NONO, SFPQ, and Connexin 43 in iPSC-TM, impairs the capacity of iPSC-TM in anchoring the LAMC1-positive regions of conventional outflow tissue, and weakens iPSC-TM in stimulating endogenous cell proliferation.

Revision to the supplementary materials:

Line 290:

Figure S10. Influence of ITGA6/NEAT1 on iPSC-TM's function. (C). Left panel: Western blot detection of Connexin 43, ITGA6, and GAPDH (from top to bottom) in ITGA6⁺ iPSC-TM, ITGA6⁺ iPSC-TM transfected with siRNA ITGA6, and ITGA6⁻ iPSC-TM. Right panel: Quantification of band intensities showing a significantly lower expression of Connexin 43 in ITGA6⁻ iPSC-TM than in ITGA6⁺ iPSC-TM. (D). Left panel: Western blot detection of Connexin 43 and GAPDH in iPSC-TM transfected with siRNA NEAT1. iPSC-TM transfected with

scrambled siRNA was used as a control. Right panel: Quantification of band intensities showing a significantly lower expression of Connexin 43 in iPSC-TM after treatment with siRNA NEAT1. *** $P < 0.001$, * $P < 0.05$ by two-tailed t-test. Typical results from $n=3$ technical repeats are shown.

My other concern is that the transplanted cells are a cocktail of iPSC-TM but not ITGA6+ iPSC-TM only, which makes it hard to conclude the better treatment is due to the ITGA6+ iPSC-TM.

Response: Based on our current data, it is evident that alpha6 integrin mainly serves as a biomarker for a functional subpopulation of iPSC-TM cells. The key factor driving the regenerative capacity of this specific subpopulation is the robust expression of NEAT1 (Fig. 2F-G and 5A), which plays a critical role in influencing two important aspects:

i) Cell targeting

As previously suggested by Reviewer #1, ITGA6+ iPSC-TM was further characterized as a distinct cell type compared to ITGA6- iPSC-TM based on its similarity to fibroblasts and endothelial cells (Fig. 4B). More intriguingly, we found that *in situ* fibroblasts and SC endothelial cells predominantly express LAMC1, the specific ligand for alpha6 integrin receptor (Fig. 4C). Consequently, these specific sites, responsible for AH outflow and IOP homeostasis, may become the primary target for anchoring of ITGA6+ iPSC-TM. Our subsequent data verified this hypothesis, as we observed a highly significant association between iPSC-TM (STEM121+ cells) at the beginning of transplantation with the LAMC1 expression (Fig. S8), which could be regulated by NEAT1 (Fig. 5J).

ii) Stimulating cell proliferation

Although the transplanted iPSC-TM gradually diminished over time (Fig. 4), we observed a strong correlation between the number of dividing iPSC-TM cells in the TM and SC at the beginning of transplantation and endogenous cell proliferation (Fig. 5K and S9). Similarly, NEAT1 knockdown not only impaired the proliferative capacity of iPSC-TM itself (Fig. 5I) but also significantly abrogated the ability of iPSC-TM to stimulate neighboring cell proliferation *in vivo* (Fig. 5K, S8 and S9) and *in vitro* (Fig. 5L) settings. Another contributing factor to this phenomenon may be the weakened cell-cell interactions between iPSC-TM and pTM due to NEAT1 knockdown (Fig. S10).

In summary, the primary distinction between ITGA6+ iPSC-TM and ITGA6- iPSC-TM in tissue regeneration lies in the differential expression levels of NEAT1.

For the magnetic nanoparticle labeling, evidence of the cellular changes especially molecular changes upon treatment with magnetic nanoparticles is necessary. The distribution of the cells in the TM should be provided. Also, I recommend a control group with MNP-labeled transplanted pTM.

Response: Thank you for your suggestions. The biocompatibility and distribution of MNP-labeled iPSC-TM were thoroughly examined in our previous study (PMID: 35345785). Immunohistochemical staining and Western blot analysis indicated that the MNP labeling did not influence cell behaviors of iPSC-TM, including the robust expressions of TM biomarkers, dexamethasone-inducible myocilin expression, and the capacity to stimulate cell proliferation. Subsequent *in vivo* evaluation revealed that the use of these nanoparticles significantly improved

the delivery accuracy of the transplanted iPSC-TM towards the conventional outflow tissue in live animals.

To address the reviewer's concern, a control group was incorporated in the study. Similar to the labeling and transplantation procedure for iPSC-TM, the same amount of pTM was labeled with the magnetic NPs and intracamerally transplanted into the eyes of 10-month-old Tg-MYOC^{Y437H} mice. In this control group, IOP was closely monitored for eight weeks after cell transplantation. As shown in Fig. 3E, no significant change in IOP was observed in the group of mice (dIOP: from -1.8 mmHg to -0.8 mmHg over time). These results have been added to the revised manuscript (lines 198, 205 and 784) and the revised supplementary materials (line 152).

Revision to the manuscript:

Line 198:

Here, 10 months old animals with IOPs over 19.2 mmHg were chosen and received NP-labeled pTM or iPSC-TM injections containing either 3.9% or 49.7% ITGA6⁺ cells.

Line 205:

Of note, no change in IOP was found in pTM-treated eyes of Tg-MYOC^{Y437H} mice (dIOP: from -1.8 mmHg to -0.8 mmHg after transplantation for 8 weeks).

Line 784:

Figure 3. Role of ITGA6⁺ iPSC-TM in restoring TM function in glaucoma mice. (E-F) Change in IOP (dIOP, left, **P*<0.05 by two-way ANOVA) and outflow facility (*C_r*, right, **P*<0.05 by two-tailed t-test) in Tg-MYOC^{Y437H} receiving iPSC-TM CM at ITGA6⁺ ratios of either 3.9 % (orange; n=5) or 49.7 % (green; n=8). A significant decrease in IOP and a noticeable increase in *C_r* in Tg-MYOC^{Y437H} mice receiving 49.7% ITGA6⁺ cells compared to those receiving 3.9% ITGA6⁺ cells (*P* = 0.012). No change in either IOP after pTM treatment. Each data point shows an individual dIOP or *C_r*. Scale bars, 100 μm.

Revision to the supplementary materials:

Line 152:

50,000 pTM or iPSC-TM cells labeled with PLGA-SPIO-Cypate nanoparticles were resuspended in 3 μ l 1 \times PBS (Gibco) and injected into mouse anterior chamber.

June 2024

We sincerely appreciate the constructive comments provided by the reviewers, which have significantly improved the quality of this study. In response, we have incorporated bulk RNA-seq findings into the revised manuscript (highlighted in red font) and elucidated the response of *in situ* TM to iPSC-ITGA6⁺ cells and its role in tissue regeneration. Our responses to specific concerns raised by the reviewers (in blue font) are outlined below (in black font).

REVIEWER COMMENTS

Reviewer #1 (Remarks to the Author):

The authors have submitted a revised manuscript that incorporates additional experiments guided by reviewer's comments. However, the new data has reinforced my central concern with the paper: That ITGA6⁺ iPSC-derived cells are not "TM cells" but are instead a fibroblast-like less differentiated cell type arising during the preparation of iPSC-TM cells. This needs to be clarified throughout the manuscript, possibly by renaming the cells to reflect their actual identity (e.g. "iPSC-ITGA6⁺ cells" or similar). Furthermore, additional work is needed to determine why these cells, and not ITGA6⁻ iPSC-TM cells, are capable of driving TM regeneration. Lacking that information, the present manuscript offers limited insights over author's previous work.

Response: We sincerely appreciate the reviewer's suggestions for strengthening the conclusion of this study. We completely agree that ITGA6⁺ iPSC-TM are not TM cells. Following reviewer's suggestion, we now refer to these cells as iPSC-ITGA6⁺ cells throughout the revised manuscript.

To further define these cells, we conducted bulk RNA sequencing (bulk RNA-seq) using iPSCs, iPSC-ITGA6⁺ cells, iPSC-ITGA6⁻ cells, and primary TM cells treated with or without iPSC-ITGA6⁺ cells. These data allow comprehensive characterization of iPSC-ITGA6⁺ cells with much higher sequencing depth as was available from scRNA-seq. We also sought to uncover the transcriptomic changes in primary TM cells upon exposure to iPSC-ITGA6⁺ cells. These newly added results (please refer to "Revisions in the manuscript" below) in conjunction with previous scRNA-seq data, have led to two new findings:

- i) Molecular characteristics of iPSC-ITGA6⁺ cells, which are crucial effectors of iPSC-based therapy for the TM, indicate that these cells display elevated levels of several markers of pluripotency when compared to iPSC-ITGA6⁻ cells (Fig. 4).
- ii) pTM exposed to iPSC-ITGA6⁺ express cellular markers of proliferation and rejuvenation, suggesting that this is the regenerative mechanism underlying iPSC-based therapy for the conventional outflow pathway (Fig. 7).

The submitted manuscript has undergone substantial revision to both improve readability, to present these new data, and to incorporate findings into the conclusions of the study. Below we highlight those changes related to the presentation of new data and our interpretation of these.

Revisions in the manuscript

Line 209

Molecular Characterization of iPSC-ITGA6⁺ cells

In order to further characterize the molecular profile of iPSC-ITGA6⁺ cells we specifically examined global transcriptional differences between ITGA6⁺ cells (33.0% ITGA6⁺ cells), ITGA6⁻ cells (14.0% ITGA6⁺ cells), and pTM and confirmed these with RT-PCR (Fig. 4A). Our results indicate significantly higher expressions of TM biomarkers, including *AQP1* and *ABCG2* in ITGA6⁺ cells when compared to ITGA6⁻ cells. However, we also observed elevated expression of the pluripotency markers *ABCG2*, *SOX2*, *NANOG*, and *NOTCH1* in iPSC-ITGA6⁺ cells, whereas *Nes* and *Pou5F1* (Oct4) were expressed at lower levels. These data suggest that iPSC-ITGA6⁺ cells retain a higher degree of pluripotency and may thus represent a less defined stage during the induction of the TM cellular phenotype. Consistent with this notion gene ontology analysis also indicates an enrichment of genes in iPSC-ITGA6⁺ cells representing developmental processes of numerous tissues, including muscle, lung, glia, and neurons (Fig. 4B). However, this cluster also is enriched in genes involved in cell differentiation, adhesion, and proliferation (Fig. 4B and S9) as well as those that are associated more directly with TM function. These include insulin signaling, which appears to play a role in the maintenance of IOP^{37,38}, and pathways regulating TGFβ production, a crucial factor mediating the production of extracellular matrix constituents by TM cells^{39,40}. Further, pathway analysis of DE genes identified *IL1A*, *SERPINE1*, *VAV3*, *COL8A1*, and *TYRP1* which are all implicated in the development of POAG (Fig. 4C-D).

Line 294

Transcriptomic changes in pTM after co-culture

It is conceivable that the observed effect of iPSC-TM on pTM proliferation is the result of cell fusion between both cell types. However, exposure to lysophosphatidylcholine (LPC), a potent inhibitor for hemifusion and subsequent cell fusion⁴¹, failed to impede the iPSC-TM's ability to stimulate pTM proliferation (Fig. 7A). Moreover, LPC treatment did not alter the expression ratio of the sex-determining region Y gene (*SRY*), a specific Y chromosome gene, to *ACTB* in female pTM co-cultured with male iPSC-TM (Fig. 7A). These results aligned closely with our previous observations *in vivo* and argue against cell fusion as a crucial mechanism¹⁵.

To further investigate possible mechanisms involved in iPSC-ITGA6⁺ cell mediated pTM proliferation we conducted bulk RNA-seq analysis of iPSCs, iPSC-ITGA6⁺ cells (33.0% ITGA6⁺ cells), ITGA6⁻ iPSC-TM (14.0% ITGA6⁺ cells), and pTM from donors 6 and 9 following co-culture (treated) with iPSC-ITGA6⁺ cells (Fig. 7B). As expected, all cell types exhibited distinct transcriptomic profiles in accordance with the distinct morphologic appearance of the cells (Fig. 7C). Hierarchical clustering of the data resulted in the formation of three main clusters: one comprised of pTM, the second comprised of ITGA6⁻ iPSC-TM and co-cultured pTM, and the third containing iPSC and iPSC-ITGA6⁺ (Fig. 7D). These findings indicate that iPSC-ITGA6⁺ cells retain some similarity to iPSC and consequently may represent an intermediate step during stem cell differentiation towards pTM. On the other hand, co-culture of pTM alters their expression pattern to resemble that of iPSC-ITGA6⁻ cells, suggesting partial reprogramming or rejuvenation of pTM⁴². To gather further support for this hypothesis, we examined expression changes of individual genes associated with cellular rejuvenation in our RNAseq data and confirmed these by RT-PCR (Fig. 7E). Increased expression in co-cultured pTM was observed for the stem cell biomarkers *NES*, *NOTCH1*, *POU5F1*, *SOX2*, and *NANOG* as well as the proliferation markers *MKI67* and *NEAT1*. Expression of *AQP1*, a biomarker for TM cells also increased but that of

MGP was reduced, suggesting that this protein may be expressed less robustly in developing TM cells. Finally, pathway analysis of our data indicates an enrichment of DEGs related to cell fate and developmental processes in co-cultured pTM (Fig. 7F-G, S13, and Table S8), further supporting the notion that exposure to iPSC-ITGA6⁺ causes cellular rejuvenation in pTM.

Line 355

While morphologically similar to other iPSC-derived TM cells, the expression pattern of ITGA6⁺ cells differs from that of pTM and ITGA6⁻ cells in several aspects, including elevated expression of the pluripotency markers *SOX2* and *NOTCH1*. As such it is conceivable that ITGA6⁺ cells represent iPSC during early stages of differentiation. ITGA6⁺ cells also differ from other iPSC-TM cell types functionally and appear to be primarily responsible for inducing pTM proliferation.

Line 380

The histological observations also support our previous finding that the limited presence of retained iPSC-TM after 2 months of transplantation renders them insufficient to sustain the burden of tissue repair in glaucoma (Fig. 5). Instead, *in situ* cell repopulation emerges as the primary mechanism, predominantly driven by the rejuvenation and proliferation of pTM following contact with iPSC-TM (Fig. 7). Notably, rejuvenation has been proven to be an effective strategy to delay the onset of aging-associated diseases, and epigenetic interventions, such as the regulation of non-coding RNAs, DNA methylation, and histone modification, have demonstrated efficacy in inducing rejuvenation⁴⁷. Given that glaucoma is a typical age-related eye disorder¹, lncRNA-based rejuvenation of pTM has become a focal point of renewed interest (Fig. 7).

Line 410

Recent findings from several laboratories have suggested that it is possible to reverse age and cell stress-induced aging phenotypes in mature cells and tissues^{55,56}. The analysis of gene expression patterns in pTM after exposure to iPSC-ITGA6⁺ indicates a shift toward expression patterns exhibited by iPSC-TM, including robust expression of stem cell biomarkers, particularly those associated with neural crest cells. These findings introduce the intriguing possibility that the regenerative effect mediated by iPSC-ITGA6⁺ is enabled by cellular rejuvenation of pTM.

Notably, regeneration of TM cells differs from tissue repair utilizing stem cells. TM stem cells serve to replace lost cells which requires anchoring and survival in the damaged tissue⁵⁷, whereas iPSC-ITGA6⁺ cells do neither exhibit nor require long-term survival. Instead, the main role of iPSC-ITGA6⁺ cells in tissue regeneration appears to lie in their capacity to induce proliferation of endogenous cells temporarily³⁶. This characteristic presents a distinct advantage for clinical translation of stem cell-based therapy and *in situ* regeneration may emerge as a novel strategy for regenerating conventional outflow tissues.

Line 876

Figure 4. Characterization of iPSC-ITGA6⁺ cells. (A) FPKM values (upper panel) and RT-PCR results (lower panel) indicating higher expressions of *AQP1*, a biomarker for the TM, and several stem cell biomarkers, including *ABCG2*, *SOX2*, *NOTCH1*, and *NANOG*, in iPSC-ITGA6⁺ cells (orange) compared to ITGA6⁻ cells (blue) and pTM (grey). Experiments were repeated three times.

** $P < 0.01$, *** $P < 0.001$, **** $P < 0.0001$ by one-way ANOVA with Tukey's post-hoc test. 6⁺: iPSC-ITGA6⁺ cells; 6⁻: iPSC-ITGA6⁻ cells. (B) Gene ontology (GO) analysis showing enrichment of cell differentiation, proliferation, and some metabolic processes in iPSC-ITGA6⁺ cells. (C). Disease ontology (DO) analysis indicating five DEGs (red), *IL1A*, *SERPINE1*, *VAV3*, *COL8A1*, and *TYRP1*, when compared iPSC-ITGA6⁺ cells to ITGA6⁻ cells, which are highly associated with the pathogenesis of open-angle glaucoma (red). (D). FPKM values (up panel) and RT-PCR results (lower panel) showing the expressions of these five DEGs in iPSC-ITGA6⁺ cells (orange) and ITGA6⁻ cells (blue). Experiments were repeated three times. * $P < 0.05$, ** $P < 0.01$, *** $P < 0.001$ by two-tailed Student's t-test.

Line 948

Figure 7. Transcriptomic changes in co-cultured pTM. (A) The capacity of iPSC-ITGA6⁺ cells to stimulate pTM proliferation (left panel) and the relative expression of *SRY* to *ACTB* in the co-cultured sample (right panel) after LPC treatment. In the right panel, pure iPSCs of U1 (male) and pTM of donor 6 (female) were used as positive and negative controls, respectively. The mixture contains two types of cells, iPSC-TM (male) and pTM (female), mixed at a ratio of 2:1. Co-cultured samples treated with or without LPC (150 μ M) for 48 hours are shown. Experiments were repeated three times. * $P < 0.05$, ** $P < 0.01$, **** $P < 0.0001$ by one-way ANOVA. (B) Principal component analysis (PCA) showing different transcriptomic profiles of iPSCs, iPSC-ITGA6^{+/-} cells, and pTM from donors 6 and 9 with or without co-culture for 7 days. Each group of samples contains two technique repeats. (C) Presentative morphology of iPSCs, iPSC-ITGA6^{+/-} cells, and pTM from donor 6. Cells at a higher magnification are shown in each image. Scale bars, 100 μ m. (D) Hierarchical clustering of DEGs in the above samples, consistent with the results in panel B. (E) FPKM values (up panel) and RT-PCR results (lower panel) showing higher expressions of biomarkers for stem cells (*NES*, *NOTCH1*, *POU5F1*, *SOX2*, and *NANOG*), cell proliferation (*MKI-67* and *NEAT1*), and TM biomarkers (*MGP* and *AQP1*) in pTM after co-culture. Experiments were repeated three times. * $P < 0.05$, ** $P < 0.01$, *** $P < 0.001$, **** $P < 0.0001$ by two-tailed Student's t-test. U: untreated pTM; T: treated pTM. (F) Volcano plot displaying DEGs in the co-cultured pTM from donor 6, some of which are involved in cell fate determination and proliferative regulation (Table S8). (G) Biological process enriched in pTM from donor 6, including neuron system development, neural precursor proliferation, integrin-mediated cell adhesion, and cell junction assembly.

Additional comments:

1: The new findings using NEAT1 knockdown cells suggest that NEAT1 is an important regulator of iPSC-TM-ITGA6⁺ cell proliferation, consistent with the known role of NEAT1 in other cells. This raises the likelihood that the reduced effect of NEAT1 knockdown cells on proliferation of cultured pTM cells and in vivo TM cells after transplantation is caused by reduced proliferation (and thus numbers) of iPSC-TM-ITGA6⁺ cells and not a direct signaling role for NEAT1. Indeed, authors report that NEAT1 knockdown is associated with reduced numbers of iPSC-TM present 48 hours after transplantation. Additional experiments are needed to clarify the role of NEAT1 and determine if its role extends beyond regulating proliferation of iPSC-TM cells.

Response: Reviewer raises a good point. However, NEAT1 knockdown resulted in only a 17.4% decrease in the of implanted iPSC-TM 48 hours post-transplantation ($P = 0.12$; Fig. 5I), whereas

NEAT1 knockdown almost completely inhibited the ability of iPSC-TM to stimulate TM proliferation (94.2% decrease, $P = 0.012$; Fig. 5K). Thus it is more plausible that NEAT1 expression in iPSC-TM, rather than the quantity of implanted iPSC-TM, plays a critical role in regulating TM cell proliferation. Remarkably, the opposite results were obtained in co-culture experiments utilizing siRNA targeting of *ITGA6*. In those experiments, decrease in *ITGA6* expression (Fig. S10B) led to a pronounced decrease in the implanted iPSC-TM population (36.4%, $P < 0.001$; Fig. R1, for review only), while only modestly affecting iPSC-TM's capacity to stimulate pTM proliferation (20.1% reduction; Fig. S10D). A more detailed description of these findings is now included in the revised manuscript (line 282).

Figure R1 (for review purposes only)

Figure R1. Normalized iPSC-TM and pTM cellularity. (A) Normalized iPSC-TM in cell transplantation study (*in vivo*) and co-culture experiments (*in vitro*), related to Figures 5I, 5K, and S10D, respectively. (B) Normalized pTM in co-culture experiments, related to Figures 5K and S10D, respectively. * $P < 0.05$, *** $P < 0.001$ by two-tailed Student's t-test.

Revisions in the manuscript

Line 282:

Since this reduction could be due to failure of NEAT1 deficient iPSC-TM to implant and survive in the eye following transplantation, we determined the abundance of these cells in the TM following knockdown. However, a significant effect was not observed (siRNA control vs. siRNA NEAT1: 39.1 vs. 32.3 iPSC-TM/section, $P = 0.12$, $n=49-69$ sections of 3 eyes in each group; Fig. 6I and S11A).

However, loss of NEAT1 significantly decreased the ability of iPSC-TM to stimulate pTM proliferation both *in vivo* (siRNA control vs. siRNA NEAT1: $P = 0.046$ when endogenous cells cross-plotted against iPSC-TM; $P = 0.044$ when endogenous cells cross-plotted against Ki-67⁺ iPSC-TM, Fig. 5J and S11B) and *in vitro* (94.2% decrease in pTM proliferation, $P < 0.01$, Fig. 5K).

2: The author's goal of improving the trajectory analysis in figure 2 A in future studies (response to reviewers page 10) is admirable, but the present version should be removed from the paper. A more useful approach for showing the relationship between these cell types would be to use a PCA, as the authors have done in the past with very similar data: <https://doi.org/10.1038/s41598-020-59941-0>. As the cell types under comparison are the end products of parallel culture or differentiation protocols and no biological trajectory exists between them, the current analysis is inappropriate.

Response: We appreciate the reviewer's valuable input and have excluded the trajectory analysis and introduced a new PCA analysis in the revised manuscript, indicating distinct transcriptomic features of iPSCs, iPSC-ITGA6⁺ cells, iPSC-ITGA6⁻ cells, and pTM (Fig. 7).

3: The new axis labeling in figure 3 C has greatly improved readability of that figure. However, as authors have retained the previous timeline labeling in figure 3 D, it has become confusing. Figure 3 D should be updated so that the X axis timeline is consistent with 3C as they describe the same data. In addition, the timepoint "before" should be labeled more descriptively to make clear which timepoint these values represent. Is it a true baseline (e.g. day 0) or immediately before cell transplantation (e.g. after glaucoma induction)?

Response: Based on the reviewer's suggestion, the X-axis and background of Fig. 3D have been revised accordingly. To clarify the reviewer's concern, it is important to note that dIOPs after cell transplantation were calculated based on the IOP measurements taken on day 25. We have provided clarification regarding this in the legend of Fig. 3D.

4: In the newly-revised section on lines 191-193, authors state "...demonstrated a 44.4% decrease 24 days after injection which was significantly higher...(Fig 3 C and D; P = 0.0017)." In figure 3 D, the legend describes this statistical analysis as a 2-way ANOVA. Please provide additional information regarding this analysis. Was the 24-day P value derived from a posttest? Are the P values reported in the figure derived from the anova itself, or are these from a posttest? If these results are derived from posttesting, please make this clear and also provide the Df and F values for the underlying 2-way anova. These data should also be provided for all other panels using ANOVA in combination with posttesting.

Response: All p-values used in one-way or two-way ANOVA were derived from Tukey's post-hoc tests. All statistical analyses of ANOVA were performed in two steps: (1) calculate the p-values of ANOVA itself to determine if there is a significant difference. (2) If p-values of ANOVA are significant (< 0.05), Tukey's post-hoc tests are performed to determine where the difference is for multiple comparisons. We provided all ANOVA tables showing the degree of freedom and f values in the supplemental Table S9. Specifically, the value of 0.0017 is the adjusted P value obtained when multiple comparisons were made.

5: Reviewer #2 raised an important point regarding statistical correction for multiple comparisons, which was not satisfactorily addressed in the revised manuscript. Regardless of the "planned" nature of the statistical comparisons reported, additional comparisons will alter the family-wise error rate and thus correction is required according to the number of comparisons performed.

Response: We appreciate the reviewer's correction and explanation and apologize for not being

clear about p-values in ANOVA with multiple comparisons. In fact, the p-values we showed are derived from Tukey's post-hoc test, but not directly from main ANOVA. As we explained in Question #4, Tukey's post-hoc tests were only performed when the p-values of ANOVA were significant. As the procedure of Tukey's maintains the alpha at the specified level (alpha = 0.05 in our case), our p-values shown have adjusted for multiple comparisons. Based on the reviewer's suggestion, we have provided clarification regarding this in the revised manuscript (lines 606).

Revisions in the manuscript

Line 606:

For multiple group comparisons, one-way ANOVA with Tukey's post-hoc test was performed for statistical analysis of the cellularity data, and two-way ANOVA with Tukey's post-hoc test was applied for the statistical analysis of IOP results. ANOVA tables, including the degree of freedom and F-values, were provided in Table S9.

6: On line 202, authors refer to figure 3 G instead of Figure 3 E.

Response: It has been corrected. Thanks.

7: Throughout the manuscript it is difficult to determine if the cells used in mouse experiments are produced with the conditioned media or recombinant cytokine approach. This should be clarified.

Response: As suggested, we have clarified it in the revised manuscript (lines 455).

Revisions in the manuscript

Line 455:

hiPSCs were subjected to one-month differentiation, and the resulting cells were designated as iPSC-TM CM and collected for scRNA-seq, bulk RNA-seq, and cell transplantation studies.

8: The association of \neg pTM5 with Schlemm's canal remains unconvincing. While ITGA8 has previously been identified in primary cells exhibiting SC-like characteristics, it is also expressed in other cell types of the angle. In addition, none of the DE genes shown in figure S3A are characteristic of Schlemm's canal, with RGS5, COL6A1 and COL6A2 especially being expressed by trabecular meshwork, smooth muscle and pericytes but not Schlemm's canal endothelium.

Response: The reviewer has raised a great point! We agree that in the absence of typical SC biomarkers, such as *Prox1*, *Ccl21a*, *Kdr*, *Plvap*, and *Ihh*, the definition of cluster pTM5 as SC endothelial cells based solely on the localization of ITGA8 was overly speculative. We have modified our description of pTM5 in the revised manuscript (line 137).

Revisions in the manuscript

Line 137:

Characterization of pTM5 remains challenging as its typical DEGs cannot be readily classified. These include *ITGA8*, primarily expressed in the inner wall of Schlemm's canal³², *RGS5*, a biomarker of pericytes, and *COL6A1/2*, robustly expressed in trabecular meshwork and fibroblasts. However, it is possible that pTM5 represents an artifact of *in vitro* cell culture, such as senescent pTM cells or those in the early stages of de-differentiating.

9: *In figure 4D, while the higher magnification panels help the reader, please remove the label “TM”. The arrows adequately indicate the TM, while the text unnecessarily obscures the region of interest.*

Response: The labels have been removed as suggested.

July 2025

We sincerely appreciate the constructive comments provided by the reviewers, which have significantly improved the quality of this study. In response, we have incorporated bulk RNA-seq findings into the revised manuscript and further elucidated the characteristics of iPSC-ITGA6⁺ cells. The sequencing data also illustrate how primary TM cells respond after interacting with iPSC-ITGA6⁺ cells. Furthermore, we have explained how NEAT1 expression or paraspeckle formation in iPSC-ITGA6⁺ cells can influence neighboring TM cells. Our responses to the specific concerns raised by the reviewers are outlined below, with their comments in blue font and our replies in black font.

REVIEWER COMMENTS

Reviewer #1 (Remarks to the Author):

The authors have submitted a revised manuscript that incorporates additional experiments guided by reviewer's comments. However, the new data has reinforced my central concern with the paper: That ITGA6⁺ iPSC-derived cells are not "TM cells" but are instead a fibroblast-like less differentiated cell type arising during the preparation of iPSC-TM cells. This needs to be clarified throughout the manuscript, possibly by renaming the cells to reflect their actual identity (e.g. "iPSC-ITGA6⁺ cells" or similar). Furthermore, additional work is needed to determine why these cells, and not ITGA6⁻ iPSC-TM cells, are capable of driving TM regeneration. Lacking that information, the present manuscript offers limited insights over author's previous work.

Response: We sincerely appreciate the reviewer's suggestions for enhancing the conclusion of this study.

Given the high sequencing depth of bulk RNA sequencing, we conducted this analysis to better define the characteristics of iPSC-ITGA6⁺ cells and to further investigate how primary TM cells respond to them. Our bulk RNA-seq data, morphological observations, and RT-PCR results indicate that iPSC-ITGA6⁺ cells retain a higher degree of pluripotency and represent a less defined stage during the induction of the TM cellular phenotype (Fig. 4). In accordance with the reviewer's recommendation, we have renamed these cells as iPSC-ITGA6⁺ cells throughout the revised manuscript. Furthermore, gene expression analysis of primary TM cells co-cultured with iPSC-ITGA6⁺ cells revealed that their interaction is characterized by the proliferation and rejuvenation of endogenous pTM cells (Fig. 6). These new findings have been included in the revised manuscript (Lines 211-231 and 267-289).

Since cell fusion is not a critical mechanism (Fig. 6), we agree with the reviewer that further investigation is needed to fully understand the regenerative mechanisms of iPSC-ITGA6⁺ cells. As scRNA-seq data have shown that the repair process of iPSC-ITGA6⁺ cells primarily relies on this long non-coding RNA, nuclear paraspeckle assembly transcript 1 (NEAT1), and the assembly of paraspeckles (Fig. 7), we next investigated how NEAT1 influences iPSC-TM function (Fig. 8). Our new data show that NEAT1 expression is abundant in primary TM cells, similar to the levels

found in iPSC-ITGA6⁺ cells. However, the quantity of paraspeckles in primary TM cells is extremely low, nearly negligible. Through investigating the pattern of paraspeckle assembly during TM development, we discovered that MEN β -associated RNA (menRNA), a tRNA-like transcript of NEAT1, plays a vital role in facilitating the assembly of paraspeckles (Fig. 8). To our knowledge, this is the first report demonstrating that menRNA is involved in paraspeckle assembly. Furthermore, the transfer of menRNA between cells offers new insight into how NEAT1 expression or paraspeckle formation in iPSC-ITGA6⁺ cells can exert transcellular effects on neighboring pTM cells (Fig. 8). Additionally, overexpressing menRNA in primary TM cells directly promote cell rejuvenation and proliferation, suggesting that strategies aimed at enhancing paraspeckle assembly could be promising approaches for regenerating TM function (Fig. 8). In summary, we have clarified the regenerative mechanism of iPSC-ITGA6⁺ cells and incorporated these new experimental data in the revised manuscript (Lines 331-352).

Additional comments:

1: The new findings using NEAT1 knockdown cells suggest that NEAT1 is an important regulator of iPSC-TM-ITGA6⁺ cell proliferation, consistent with the known role of NEAT1 in other cells. This raises the likelihood that the reduced effect of NEAT1 knockdown cells on proliferation of cultured pTM cells and in vivo TM cells after transplantation is caused by reduced proliferation (and thus numbers) of iPSC-TM-ITGA6⁺ cells and not a direct signaling role for NEAT1. Indeed, authors report that NEAT1 knockdown is associated with reduced numbers of iPSC-TM present 48 hours after transplantation. Additional experiments are needed to clarify the role of NEAT1 and determine if its role extends beyond regulating proliferation of iPSC-TM cells.

Response: The reviewer raised an excellent point! In response, we have evaluated the cellularity of iPSC-TM in our co-culture experiments (Fig. 7H and S9D) and compared it to the corresponding cellularity of pTM. This ratio indicates the regenerative capability of iPSC-TM, while minimizing the influence of iPSC-TM cellularity itself. As shown in Fig. 7H, iPSC-TM cells treated with siRNA targeting NEAT1 lost their ability to stimulate the proliferation of pTM, while iPSC-TM cells treated with scramble siRNA retained this capacity. This finding is consistent with our previous observation based solely on pTM cellularity (Fig. 7F), which further confirms the important role of NEAT1.

Interestingly, we observed that iPSC-TM maintained its pro-proliferative ability even after losing ITGA6 expression (Fig. S9D). The primary reason for the decreased pTM cellularity in our earlier observation is attributed to the reduced population of iPSC-TM due to ITGA6 deficiency. This finding further supports our hypothesis that ITGA6 primarily functions in facilitating iPSC-TM anchoring to target tissues rather than in sustaining the regenerative function of iPSC-TM. For clarity, we have summarized these findings in Fig. R1 (for review purposes only), and a more detailed description has been included in the revised manuscript (Fig. 7H and S9D; Lines 320-324).

Figure R1

Figure R1. Ratios of pTM cellularity compared to the corresponding iPSC-TM in each co-culture sample, related to Fig. 7H and S9D. * $P < 0.05$ by one-way ANOVA.

2: The author's goal of improving the trajectory analysis in figure 2 A in future studies (response to reviewers page 10) is admirable, but the present version should be removed from the paper. A more useful approach for showing the relationship between these cell types would be to use a PCA, as the authors have done in the past with very similar data: <https://doi.org/10.1038/s41598-020-59941-0>. As the cell types under comparison are the end products of parallel culture or differentiation protocols and no biological trajectory exists between them, the current analysis is inappropriate.

Response: As suggested, we have excluded the trajectory analysis and introduced a new PCA analysis using the transcriptomic profiles of iPSCs, iPSC-ITGA6⁺ cells, iPSC-ITGA6⁻ cells, and pTM co-cultured with or without iPSC-ITGA6⁺ cells (Figs. 4 and 6). Hierarchical clustering resulted in the formation of three main clusters: one comprising pTM, the second comprising ITGA6⁻ iPSC-TM and co-cultured pTM, and the third containing iPSC and iPSC-ITGA6⁺ cells (Fig. 6B). Pathway analyses revealed that iPSC-ITGA6⁺ cells retain a higher degree of pluripotency (Fig. 4C) and suggested a partial reprogramming or rejuvenation of co-cultured pTM (Fig. 6E). Due to these PCA results, we can be more confident in our study of NEAT1 and have discovered a new function of menRNA. We appreciate the reviewer's valuable input. These results have been added in the revised manuscript (Figs. 4 and 6).

3: The new axis labeling in figure 3 C has greatly improved readability of that figure. However, as authors have retained the previous timeline labeling in figure 3 D, it has become confusing. Figure 3 D should be updated so that the X axis timeline is consistent with 3C as they describe the same data. In addition, the timepoint "before" should be labeled more descriptively to make clear which timepoint these values represent. Is it a true baseline (e.g. day 0) or immediately before cell transplantation (e.g. after glaucoma induction)?

Response: In response to the reviewer's suggestion, we have revised the X-axis and background of Fig. 3D to align with Fig. 3C. To address the reviewer's concern, it is important to note that diOPs after cell transplantation were calculated based on the IOP measurements taken on day 25. We have provided clarification regarding this in the legend of Fig. 3D.

4: In the newly-revised section on lines 191-193, authors state "...demonstrated a 44.4% decrease

24 days after injection which was significantly higher...(Fig 3 C and D; P = 0.0017).” In figure 3 D, the legend describes this statistical analysis as a 2-way ANOVA. Please provide additional information regarding this analysis. Was the 24-day P value derived from a posttest? Are the P values reported in the figure derived from the anova itself, or are these from a posttest? If these results are derived from posttesting, please make this clear and also provide the Df and F values for the underlying 2-way anova. These data should also be provided for all other panels using ANOVA in combination with posttesting.

Response: All p-values used in one-way or two-way ANOVA were derived from Tukey’s post-hoc tests. All statistical analyses of ANOVA were performed in two steps: (1) calculate the p-values of ANOVA itself to determine if there is a significant difference. (2) If p-values of ANOVA are significant (< 0.05), Tukey’s post-hoc tests are performed to determine where the difference is for multiple comparisons. We provided all ANOVA tables showing the degrees of freedom and F values in the supplemental Table S9. Specifically, the value of 0.0017 is the adjusted P value obtained when multiple comparisons were made.

5: Reviewer #2 raised an important point regarding statistical correction for multiple comparisons, which was not satisfactorily addressed in the revised manuscript. Regardless of the “planned” nature of the statistical comparisons reported, additional comparisons will alter the family-wise error rate and thus correction is required according to the number of comparisons performed.

Response: We appreciate the reviewer’s correction and explanation, and we apologize for our previous mistake. In fact, the p-values we showed are derived from Tukey’s post-hoc test, but not directly from the main ANOVA. As we explained in Question #4, Tukey’s post-hoc tests were only performed when the p-values of ANOVA were significant. As the procedure of Tukey’s maintains the alpha at the specified level (alpha = 0.05 in our case), our p-values shown have been adjusted for multiple comparisons. Based on the reviewer’s suggestion, we have provided clarification regarding this matter in the revised manuscript (Please see Table S9 and Supplemental Materials and Methods, Lines 361-370).

6: On line 202, authors refer to figure 3 G instead of Figure 3 E.

Response: It has been corrected. Thanks.

7: Throughout the manuscript it is difficult to determine if the cells used in mouse experiments are produced with the conditioned media or recombinant cytokine approach. This should be clarified.

Response: As suggested, we have clarified it in the revised manuscript (Supplemental Materials and Methods, Lines 68-70 and 79-81).

8: The association of \neg pTM5 with Schlemm’s canal remains unconvincing. While ITGA8 has previously been identified in primary cells exhibiting SC-like characteristics, it is also expressed in other cell types of the angle. In addition, none of the DE genes shown in figure S3A are characteristic of Schlemm’s canal, with RGS5, COL6A1 and COL6A2 especially being expressed by trabecular meshwork, smooth muscle and pericytes but not Schlemm’s canal endothelium.

Response: We agree that in the absence of typical SC biomarkers, such as *Prox1*, *Ccl21a*, *Kdr*, *Plvap*, and *Ihh*, the definition of cluster pTM5 as SC endothelial cells based solely on the

localization of ITGA8 was overly speculative. We have modified our description of pTM5 in the revised manuscript (Lines 137-141).

Revisions in the revised manuscript:

Characterization of pTM5 remains challenging as its typical DEGs cannot be readily classified. These include *ITGA8*, primarily expressed in the inner wall of Schlemm's canal [26], *RGS5*, a biomarker of pericytes, and *COL6A1/2*, robustly expressed in TM and fibroblasts. However, it is possible that pTM5 represents an artifact of *in vitro* cell culture, such as senescent pTM cells or those in the early stages of de-differentiation.

9: In figure 4D, while the higher magnification panels help the reader, please remove the label "TM". The arrows adequately indicate the TM, while the text unnecessarily obscures the region of interest.

Response: The labels have been removed as suggested. Thanks.